# Neural Rendering Model: Joint Generation and Prediction for Semi-Supervised Learning

## Abstract

Unsupervised and semi-supervised learning are important problems that are especially challenging with complex data like natural images. Progress on these problems would accelerate if we had access to appropriate generative models under which to pose the associated inference tasks. Inspired by the success of Convolutional Neural Networks (CNNs) for supervised prediction in images, we design the Neural Rendering Model (NRM), a new hierarchical probabilistic generative model whose inference calculations correspond to those in a CNN. The NRM introduces a small set of latent variables at each level of the model and enforces dependencies among all the latent variables via a conjugate prior distribution. The conjugate prior yields a new regularizer for learning based on the paths rendered in the generative model for training CNNs–the Rendering Path Normalization (RPN). We demonstrate that this regularizer improves generalization both in theory and in practice. Likelihood estimation in the NRM yields the new Max-Min cross entropy training loss, which suggests a new deep network architecture–the Max-Min network–which exceeds or matches the state-of-art for semi-supervised and supervised learning on SVHN, CIFAR10, and CIFAR100.

*Keywords:* neural nets, generative models, semi-supervised learning, cross-entropy

## 1 Introduction

Unsupervised and semi-supervised learning have still lagged behind compared to performance leaps we have seen in supervised learning over the last five years. This is partly due to a lack of good generative models that can capture all latent variations in complex domains such as natural images and provide useful structures that help learning. When it comes to probabilistic generative models, it is hard to design good priors for the latent variables that drive the generation.

Instead, recent approaches avoid the explicit design of image priors. For instance, the Generative Adversarial Networks (GANs) use implicit feedback from an additional discriminator that distinguishes real from fake images (Goodfellow et al., 2014). Using such feedback helps GANs to generate visually realistic images, but it is not clear if this is the most effective form of feedback for predictive tasks. Moreover, due to separation of generation and discrimination in GANs, there are typically more parameters to train, and this might make it harder to obtain gains for semi-supervised learning in the low (labeled) sample setting.

We propose an alternative approach to GANs by designing a class of probabilistic generative models, such that inference in those models also has good performance on predictive tasks. This approach is well-suited for semi-supervised learning since it eliminates the need for a separate prediction network. Specifically, we answer the following question: what generative processes output Convolutional Neural Networks (CNNs) when inference is carried out? This is natural to ask since CNNs are state-of-the-art (SOTA) predictive models for images, and intuitively, such powerful predictive models should capture some essence of image generation. However, standard CNNs are not directly reversible and likely do not have all the information for generation since they are trained for predictive tasks such as image classification. We can instead invert the irreversible operations in CNNs, e.g., the rectified linear units (ReLUs) and spatial pooling, by assigning auxiliary latent variables to account for uncertainty in the CNN's inversion process due to the information loss.

**Contribution 1 – Neural Rendering Model:** We develop the Neural Rendering Model (NRM) whose bottom-up inference corresponds to a CNN architecture of choice (see Figure 1a). The "reverse"

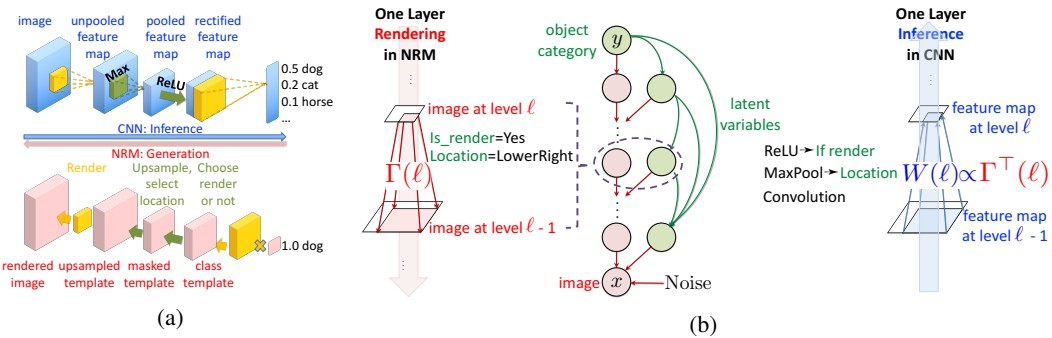

Figure 1: (a) The Neural Rendering Model (NRM) captures latent variations in the data and yields CNNs as its inference. Costs for training CNNs are derived from likelihood estimations in NRM. (b) Graphical model depiction of NRM. Latent variables in NRM depend on each other. Object category $y$ decides the class template, and then new latent variables are incorporated at each layer to render intermediate images (red) with finer details. Finally, pixel noise is added to render the image $x$.

top-down process of image generation is through coarse-to-fine rendering, which progressively increases the resolution of the rendered image. This is intuitive since the reverse process of bottom-up inference reduces the resolution (and dimension) through operations such as spatial pooling. We also introduce structured stochasticity in the rendering process through a small set of discrete latent variables, which capture the uncertainty in reversing the CNN feed-forward process. The rendering in NRM follows a product of linear transformations, which can be considered as the transpose of the inference process in CNNs. In particular, the rendering weights in NRM are proportional to the transposes of the filters in CNNs. Furthermore, the bias terms in the ReLU units at each layer (after the convolution operator) make the latent variables in different network layers dependent (when the bias terms are non-zero). This design of image prior has an interesting interpretation from a predictive-coding perspective in neuroscience: the dependency between latent variables can be considered as a form of backward connections that captures prior knowledge from coarser levels in NRM and helps adjust the estimation at the finer levels (Rao & Ballard, 1999; Friston, 2018).

NRM is a likelihood-based framework, where unsupervised learning can be derived by maximizing the expected complete-data log-likelihood of the model while supervised learning is done through optimizing the class-conditional log-likelihood. Semi-supervised learning unifies both log-likelihoods into an objective cost for learning from both labeled and unlabeled data. The NRM prior has the desirable property of being a conjugate prior, which makes learning in NRM computationally efficient.

Interestingly, we derive the popular *cross-entropy loss* used to train CNNs for supervised learning *as an upper bound of the NRM's negative class-conditional log-likelihood*. This new interpretation of cross-entropy allow us to develop better losses for training CNNs. An example is the *Max-Min cross-entropy* discussed in Contribution 2 and Section 3.

**Contribution 2 – New regularization, loss function, architecture and generalization bounds:** The joint nature of generation, inference, and learning in NRM allows us to develop new training procedures for semi-supervised and supervised learning, as well as new theoretical (statistical) guarantees for learning. In particular, for training, we derive a new form of regularization termed as the *Rendering Path Normalization* (RPN) from the NRM's conjugate prior. A rendering path is a set of latent variable values in NRM. Unlike the path-wise regularizer in (Neyshabur et al., 2015), RPN uses information from a generative model to penalizes the number of the possible rendering paths, encouraging the network to be compact in terms of representing the image. It also helps enforce the dependency among different layers in NRM during training and improves classification performance.

We provide a new theoretical bound based on NRM. In particular, we prove that NRM is statistically consistent and derive a generalization bound of NRM for (semi-)supervised learning tasks. Our generalization bound is proportional to the number of active rendering paths that generate close-to-real images. This suggests that RPN regularization may help in generalization since RPN enforces the dependencies among latent variables in NRM and, therefore, reduces the number of active rendering paths. We observe that RPN helps improve generalization in our experiments.

*Max-Min cross-entropy and network*: We propose the new *Max-Min cross-entropy* loss function for learning, based on negative class-conditional log-likelihood in NRM. It combines the traditional cross-entropy with another loss, which we term as the Min cross-entropy. While the traditional (Max) cross-entropy maximizes the probability of correct labels, the Min cross-entropy minimizes

the probability of incorrect labels. We show that the Max-Min cross-entropy is also an upper bound to the negative conditional log-likelihood of NRM, just like the cross-entropy loss. The Max-Min cross-entropy is realized through a new CNN architecture, namely the *Max-Min network*, which is a CNN with an additional branch sharing weights with the original CNN but containing minimum pooling (MinPool) operator and negative rectified linear units (NReLUs), i.e., $\min(\cdot, 0)$ (see Figure 4). Although the Max-Min network is derived from NRM, it is a meta-architecture that can be applied independently on any CNN architecture. We show empirically that Max-Min networks and cross-entropy help improve the SOTA on object classification for supervised and semi-supervised learning.

**Contribution 3 – State-of-the-art empirical results for semi-supervised and supervised learning:** We show strong results for semi-supervised learning over CIFAR10, CIFAR100 and SVHN benchmarks in comparison with SOTA methods that use and do not use consistency regularization. Consistency regularization, such as those used in Temporal Ensembling (Laine & Aila, 2017) and Mean Teacher (Tarvainen & Valpola, 2017), enforces the networks to learn representation invariant to realistic perturbations of the data. NRM alone outperforms most SOTA methods which do not use consistency regularization (Salimans et al., 2016; Dumoulin et al., 2017) in most settings. Max-Min cross-entropy then helps improves NRM's semi-supervised learning results significantly. When combining the NRM, Max-Min cross-entropy, and Mean Teacher, we achieve SOTA results or very close to those on CIFAR10, CIFAR100, and SVHN (see Table 2, 3, and 4). Interestingly, compared to the other competitors, our method is consistently good, achieving either best or second best results in all experiments. Furthermore, Max-Min cross-entropy also helps supervised learning. Using the Max-Min cross-entropy, we achieve SOTA result for supervised learning on CIFAR10 (2.30% test error). Similarly, Max-Min cross-entropy helps improve supervised training on ImageNet.

Despite good classification results, there is a caveat that NRM may not generate good looking images since that objective is not "baked" into its training. NRM is primarily aimed at improving semi-supervised and supervised learning through better regularization. Potentially, an adversarial loss can be added to NRM to improve visual characteristics of the image.

**Related Work:** In addition to GANs, other recently developed deep generative models include the Variational Autoencoders (VAE) (Kingma & Welling, 2013) and the Deep Generative Networks (Kingma et al., 2014). Unlike these models, which replace complicated or intractable inference by CNNs, NRM derives CNNs as its inference. This advantage allows us to develop better learning algorithms for CNNs with statistical guarantees, as being discussed in Section 2.3 and 2.4. Recent works including the Bidirectional GANs (Donahue et al., 2017) and the Adversarially Learned Inference model (Dumoulin et al., 2017) try to make the discriminators and generators in GANs reversible of each other, thereby providing an alternative way to invert CNNs. These approaches, nevertheless, still employ a separate network to bypass the irreversible operators in CNNs. NRM is also close in spirit to the Deep Rendering Model (DRM) (Patel et al., 2016) and the Multi-layered Convolutional-Sparse-Coding Model (ML-CSC) (Papyan et al., 2018) but markedly different. Compared to NRM, DRM and ML-CSC have several limitations. In particular, latent variables in DRM are assumed to be independent, which is rather unrealistic. This lack of dependency causes the missing of the bias terms in the ReLUs of the CNN derived from DRM. Furthermore, the cross-entropy loss used in training CNNs for supervised learning tasks is not captured naturally by DRM and ML-CSC. Due to these limitations, model consistency and generalization bounds are not derived for DRM and ML-CSC.

**Notation:** To facilitate the presentation, the NRM's notations are deferred to Table 5 in Appendix A.

## 2 THE NEURAL RENDERING MODEL

### 2.1 GENERATIVE MODEL

The NRM attempts to invert the CNNs as its inference by employing the structure of its latent variables. The dependencies among latent variables in the model are implicitly captured by their conjugate prior distribution. More precisely, NRM can be defined as follows:

**Definition 2.1.** [**Neural Rendering Model (NRM)**] NRM is a deep generative model in which the latent variables $z(\ell) = \{t(\ell), s(\ell)\}_{\ell=1}^{L}$ at different layers $\ell$ are dependent. Let $x$ be the input image and $y \in \{1, \ldots, K\}$ be the target variable, e.g. object category. Generation in NRM takes the form

$$\pi_{z|y} \triangleq \text{Softmax}\left(\frac{1}{\sigma^2}\eta(y, z)\right); \; z|y \sim \text{Cat}(\pi_{z|y}) \tag{1}$$

$$h(y, z; 0) \triangleq \Lambda(z; 1)\Lambda(z; 2)\cdots\Lambda(z; L)\mu(y); \; x|z, y \sim \mathcal{N}(h(0), \sigma^2\mathbf{1}_{D(0)}), \tag{2}$$

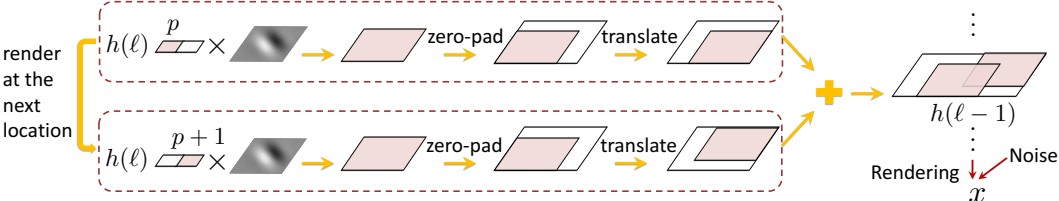

Figure 3: Rendering from level $\ell$ to level $\ell - 1$ in the NRM. At each pixel $p$ in the intermediate image $h(\ell)$, NRM renders iff the template selects latent variable $s(\ell, p) = 1$. If rendering, the template $\Gamma(\ell, p)$ is multiplied by the pixel value $h(\ell, p)$. Then the matrix $B(\ell, p)$ zero-pads the result to the size of the intermediate image at level $\ell - 1$. Next, the translation matrix $T(\ell, p)$ locally translates the rendered image to location specified by the latent variable $t(\ell, p)$. The same rendering repeats at other pixels of $h(\ell)$. NRM adds all rendered images to obtain the intermediate image $h(\ell - 1)$.

where $\eta(y, z) \triangleq \sum_{\ell=1}^{L} \langle b(t; \ell), s(\ell) \odot h(\ell) \rangle$ and Softmax $(\eta) \triangleq \exp(\eta) / \sum_{\eta} \exp(\eta)$.

The generation process in the NRM can be summarized in the following steps (details are in Algorithm 1 in Appendix A): 1) Sample the latent variables $z$ from a categorical distribution whose prior is $\pi_{z|y}$. 2) Render its coarsest image, the object template $h(L) = \mu(y)$ of class $y$. 3) Incorporate a set of of latent variations $z(\ell)$ into $h(y; \ell)$ at each layer $\ell$ via a linear transformation $\Lambda(z; \ell)$ to render the finer image $h(y, z; \ell - 1)$. 4) Add Gaussian pixel noise into $h(y, z; 0)$ to render the final image $x$. In Eqn.2, if $\Lambda(z; \ell)$ is a linear transformation, it will yield many parameters to learn. As a result, we would like to introduce new structures into NRM so that CNNs can be derived as NRM's inference. This way, NRM will be informed by the prior knowledge of natural images captured by CNNs. In our attempt to invert CNNs, we constrain

Layer 4 Layer 3 Layer 2 Layer 1 Layer 0 Original

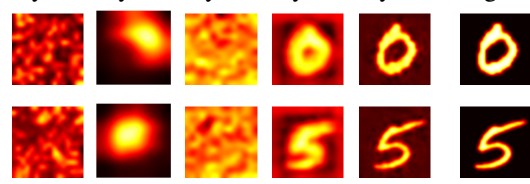

Figure 2: Reconstructed images at each layer in a 5-layer NRM trained on MNIST with 50K labeled data. Original images are in the rightmost column. Early layers in the rendering such as layer 4 and 3 capture coarse-scale features of the image while later layers such as layer 2 and 1 capture finer-scale features. From layer 2, we begin to see the gist of the rendered digits.

the latent variables $z(\ell)$ at layer $\ell$ in NRM to a set of template selecting latent variables $s(\ell)$ and local translation latent variables $t(\ell)$. As been shown later in Section 2.2, during inference of NRM, the ReLU non- linearity at layer $\ell$ "inverts" $s(\ell)$ to find if particular features are in the image or not. Similarly, the MaxPool operator "inverts" $t(\ell)$ to locate where particular features, if exist, are in the image. Both $s(\ell)$ and $t(\ell)$ are vectors indexed by $p(\ell) \in \mathcal{P}(\ell) \equiv \{\text{pixels in level } \ell\}$. The rendering process from layer $\ell$ to layer $\ell - 1$ is given by:

$$h(\ell - 1) \triangleq \Lambda(\ell)h(\ell) = \sum_{p \in \mathcal{P}(\ell)} s(\ell, p)T(t; \ell, p)B(\ell, p)\Gamma(\ell, p)h(\ell, p). \qquad (3)$$

The rendering process in Eqn.3 is illustrated in Figure 3. At each pixel $p$ in the intermediate image $h(\ell)$ at layer $\ell$, NRM decides to use that pixel to render or not according to the value of the template selecting latent variable $s(p; \ell)$ at that pixel location. If $s(p; \ell) = 1$, then NRM renders; otherwise, it does not. If rendering, the pixel value $h(\ell, p)$ is used to scale the local rendering template $\Gamma(\ell, p)$, which has the same number of feature maps as the next rendered image $h(\ell - 1)$, but is of smaller size, e.g., $3 \times 3$ or $5 \times 5$. Next, the padding matrix n $B(\ell, p)$ pads the resultant patch to the size of the image $h(\ell - 1)$ with zeros, and the translation matrix $T(t; \ell, p)$ translates the result to a local location. NRM then keeps rendering at the next pixel location $p + 1$ following the same process. All rendered images are added to form the final rendered image $h(\ell - 1)$ at layer $\ell - 1$. Note that we can constrain NRM by enforcing all pixels in the same feature maps of $h(\ell)$ share the same rendering template. This constraint helps to yield convolutions in CNNs during the inference of NRM, and the rendering template in NRM now corresponds to the convolution filters in CNNs. Reconstructed images at each layer of a 5-layered NRM trained on MNIST are visualized in Figure 2. The network is trained using the semi-supervised learning framework discussed in Section 2.3.

While $s(\ell)$ and $t(\ell)$ can be independent, we further constrain the model by *enforcing the dependency among $s(\ell)$ and $t(\ell)$ at different layers in NRM*. This constraint is motivated from realistic rendering of natural objects: *different parts of a natural object are dependent on each other*. NRM captures such dependency in natural objects by imposing more structures into the joint prior of latent variables at all layer in the model. The form of the join prior $\pi_{z|y}$ in Eqn. 1 might look mysterious at first, but NRM parametrizes $\pi_{z|y}$ in this particular way so that $\pi_{z|y}$ is the conjugate prior of the model likelihood as being proven in Appendix C.11. The conjugate form of $\pi_{z|y}$ allows efficient inference in the NRM. Parameters $b(t;\ell)$ of the conjugate prior $\pi_{z|y}$ *will become the bias terms after the CNN's convolutions* as will be shown in Theorem 2.2. When training in an unsupervised setup, the conjugate prior results in the RPN regularization as shown in Theorem 2.3(b). This RPN regularization helps enforce the dependencies among latent variables and increases the likelihood of latent configuration presents in the data during training.

Table 1: Correspondence between NRM and CNN.

| NRM (Generation) | CNN (Inference) |
|---|---|
| Rendering templates $\Gamma(\ell)$ | Transpose of weights $W(\ell)$ |
| Class templates $\mu(y)$ | Softmax weights |
| Parameters $b(t;\ell)$ of the conjugate prior $\pi_{y|x}$ | Bias terms $b(\ell)$ after each convolution |
| Intermediate image $h(\ell) = h(y, z; \ell)$ of size $D(\ell)$ in layer $\ell$ | Feature maps in layer $\ell$ in CNNs |
| Latent variables $z(\ell, p) = \{s(\ell, p), t(\ell, p)\}$ at pixel $p$, layer $\ell$ | States of ReLUs & MaxPools |
| Max over template selecting variable $s(\ell, p)$ | ReLU |
| Max over local translation variable $t(\ell, p)$ | MaxPool |
| Zero-padding matrices $B(\ell, p)$ in layer $\ell$ | Downsampling in MaxPool |
| Conditional log-likelihood | Cross-entropy loss |
| Expected complete-data log-likelihood | Reconstruction loss |

**NRM with skip connections:** We derive ResNet and DenseNet by adding skip connections into the rendering matrices $\Lambda(\ell)$ of NRM (see Appendix B).

## 2.2 INFERENCE

We show that calculations in the bottom-up inference in NRM corresponds to a CNN (see Figure 1b) and, therefore, is tractable and efficient. The impact of this correspondence goes beyond a reverse-engineering effort. First, it provides probabilistic semantics for components in CNNs, justifying their usage, and providing an opportunity to employ probabilistic inference methods in the context of CNNs. Second, such a correspondence offers a flexible framework to design CNNs. Instead of directly engineering CNNs for new tasks and datasets, we can modify NRM to incorporate our knowledge of the tasks and datasets into the model and perform JMAP inference to achieve a new CNN architecture. The following theorem establishes the NRM-CNN correspondence:

**Theorem 2.2.** *The JMAP inference of latent variable z in NRM is the feedforward step in CNNs. Particularly, we have:*

$$\max_z \{p(z|x, y)\} = \max_z \frac{1}{\sigma^2} \{\langle h(y, z; 0), x\rangle + \eta(y, z)\} + const \geq \frac{1}{\sigma^2}\langle \mu(y), \psi(L)\rangle + const \quad (4)$$

*where $\psi(L)$ is computed recursively. In particular, $\psi(0) = x$ and*

$$\psi(\ell) = \text{MaxPool}(\text{ReLu}(\text{Conv}(\Gamma^\top(\ell), \psi(\ell - 1))) + b(\ell)). \quad (5)$$

*The equality holds in Eqn. 4 when the parameters $\theta$ in NRM satisfy the non-negativity assumption that the intermediate rendered image $h(\ell) \geq 0$, $\forall \ell = 1, 2, \ldots, L$.*

There are four key results in Theorem 2.2. First, ReLU non-linearities in CNNs find the optimal value for the template selecting latent variables $s(\ell)$ at each layer $\ell$ in NRM, detecting if particular features exist in the image or not. Second, MaxPool operators in CNNs find the optimal value for the local translation latent variables $t(\ell)$ at each layer $\ell$ in NRM, locating where features are rendered in the image. Third, bias terms after each convolution in CNNs are from the prior knowledge of latent variables in the model. Those bias terms update the posterior estimation of latent variables from data using the knowledge encoded in the prior distribution of those latent variables. Fourth, convolutions in CNNs result from reversing the local rendering operator using template $\Gamma(\ell)$ in NRM. Instead of rendering as in NRM, convolutions in CNNs perform template matching. The convolution weights $W(\ell)$ in CNNs are proportional to the transposes of the rendering templates $\Gamma(\ell)$. The proofs for these correspondences and derivations for leaky ReLU and batch normalization are in Appendix C.

## 2.3 LEARNING

Learning in NRM can be posed as likelihood estimation problems in which we find the optimal values for parameters in NRM to optimize the appropriate likelihood functions. The optimization can be done by gradient-based methods (Robbins & Monro, 1985). The following theorem derives the learning objectives for NRM, and more details and proofs is provided in Appendix B and C.

**Theorem 2.3.** *For any $n \geq 1$, let $x_1, \ldots, x_n$ be i.i.d. samples from the NRM. Assume that the final rendered template $h(y, z; 0)$ is normalized such that its norm is constant. The following holds:*
*(a) Cross-entropy loss for training CNNs with labeled data*

$$\max_{(z_i)_{i=1}^n, \theta} \frac{1}{n} \sum_{i=1}^n \log p(y_i | x_i, z_i; \theta) \geq \max_\theta \frac{1}{n} \sum_{i=1}^n \log q(y_i | x_i) = -\min_{\theta \in \mathcal{A}_\gamma} H_{p,q}(y | x) \tag{6}$$

*where $q(y|x)$ is the posterior estimated by CNN, and $H_{p,q}(y|x)$ is the cross-entropy between $q(y|x)$ and the true posterior $p(y|x)$ given by the ground truth.*

*(b) Reconstruction loss with RPN for unsupervised training of CNNs with labeled and unlabeled data*

$$\min_\theta \frac{1}{n} \sum_{i=1}^n \mathbb{E}\left[\log p(x_i, z_i | y_i)\right] \overset{asymp}{\approx} \min_\theta \frac{1}{n} \sum_{i=1}^n \frac{\|x_i - h(y_i, z_i^*; 0)\|^2}{2} + RPN, \text{ when } \sigma \to 0 \tag{7}$$

*where the latent variable $z_i^*$ is estimated by the CNN as described in Theorem 2.2, $h(y_i, z_i^*; 0)$ is the reconstructed image, and the RPN regularization is the negative log prior defined as follows:*

$$RPN = -\frac{1}{n} \sum_{i=1}^n \log p(z_i^* | y_i) = -\frac{1}{n} \sum_{i=1}^n Softmax\left(\eta(y_i, z_i^*)\right). \tag{8}$$

**Cross-Entropy Loss for Training CNNs with Labeled Data:** Theorem 2.3(a) establishes the cross-entropy loss in the context of CNNs as an upper bound of the NRM's negative conditional log-likelihood $L_{\text{sup}} := -\frac{1}{n} \sum_{i=1}^n \log p(y_i | x_i, z_i; \theta)$. In contrast to other derivations of cross-entropy loss via logistic regression, we derives the cross-entropy loss in conjunction with the architecture of CNNs since the estimation of the optimal latent variables $z^*$ is part of the optimization in Eqn. 6. In other word, Theorem 2.3(a) ties feature extraction and learning for classification in CNNs into an end-to-end conditional likelihood estimation problem in NRM. This new interpretation of the cross-entropy loss suggests an interesting direction in which better losses for training CNNs with labeled data for supervised classification tasks can be derived from other upper bounds of $L_{sup}$. The Max-Min cross-entropy in Section 3 is an example.

**Reconstruction Loss with RPN Regularization:** Theorem 2.3(b) suggests that NRM learns without labels by maximizing its expected complete-data log-likelihood. One term in this objective function is the reconstruction loss between the input image $x_i$ and the reconstructed template $h(y_i, z_i^*; 0)$. Another term is the RPN from Eqn. 8. RPN encourages the $(y_i, z_i^*)$ inferred in the bottom-up E-step to have higher prior among all possible values of $(y_i, z_i)$ and, thanks to its structure, enforces the dependencies among latent variables $(s(\ell), t(\ell))$ at different layers in NRM.

**Semi-Supervised Learning in NRM:** Let $x_1, \ldots, x_n$ be i.i.d. samples from NRM and assume that the labels $y_1, \ldots, y_{n_1}$ are unknown for some $0 \leq n_1 \leq n$, NRM determines optimal parameters employed for the semi-supervised classification task via the following model:

$$\min_\theta \left\{ \frac{\alpha_{\text{RC}}}{n} \sum_{i=1}^n \left( \frac{\|x_i - h(y_i, z_i^*; 0)\|^2}{2} + RPN \right) - \frac{\alpha_{\text{CE}}}{n - n_1} \sum_{i=n_1+1}^n \log q_\theta(y_i | x_i) \right\}, \tag{9}$$

where $\alpha_{\text{RC}}$ and $\alpha_{\text{CE}}$ are non-negative weights associated with the reconstruction loss with RPN regularization and the cross-entropy loss, respectively. Again, the optimal latent variables $z_i^*$ is estimated by CNN as in Theorem 2.2. For unlabeled data, $y_i$ is set to the label estimated by CNN.

## 2.4 GENERALIZATION BOUND FOR CLASSIFICATION

Our generalization bound for classification with NRM is proportional to the ratio of the number of active rendering paths and the total number of rendering paths in the trained NRM. A rendering path is a configuration of all latent variables in NRM, and active optimal rendering paths are those among optimal rendering paths $(\hat{y}, \hat{z})$ whose corresponding rendered image is sufficiently close to one of the data point from the input data distribution. Let $L_{\mathcal{A}}$ and $L_{\mathcal{D}}$ denote the population and empirical losses on the data population $\mathcal{A}$ and the training set $\mathcal{D}$ of NRM, respectively. Our key result is summarized below. More details and proofs are deferred to Appendix B and C.

**Theorem 2.4.** *Under the margin-based loss, with high probability, the following result on the generalization bound of the classification framework with optimal solutions from Eqn. 9 holds:*

$$L_{\mathcal{A}} \leq L_{\mathcal{D}} + \overline{\tau}_n |\mathcal{L}|/\sqrt{n}.$$

Here, $\overline{\tau}_n \in (0,1)$ denotes the ratio of active optimal rendering paths among all the optimal rendering paths, $|\mathcal{L}|$ is the total number of rendering paths, and $n$ is the number of training data samples. The dependence of generalization bound on the number of active rendering paths $\overline{\tau}_n|\mathcal{L}|$ helps to justify our modeling assumptions. In particular, NRM helps to reduce the number of active rendering paths thanks to the dependencies among its latent variables, thereby tightening the generalization gap. Nevertheless, there is a limitation regarding the current generalization bound. In particular, the bound involves the number of rendering paths $|\mathcal{L}|$, which is usually large. This is mainly because our bound has not fully taken into account the structure of CNNs, which is the limitation shared among other latest generalization bounds for CNN. It is interesting to explore if techniques in works by Bartlett et al. (2017) and Golowich et al. (2018) can be employed to improve the term $|\mathcal{L}|$ in our bound.

## 3 NEW MAX-MIN CROSS ENTROPY FROM THE NEURAL RENDERING MODEL

In this section, we explore a particular way to derive an alternative to cross-entropy inspired by the results in Theorem 2.3(a). In particular, denoting $z^{\max} \triangleq \arg\max_z \{\langle h(y,z;0), x \rangle + \eta(y,z)\}$ and $z^{\min} \triangleq \arg\min_z \{\langle h(y,z;0), x \rangle + \eta(y,z)\}$, the new cross-entropy $H^{M\&M}$, which is called the *Max-Min cross-entropy*, is the weighted average of the cross-entropy losses from $z^{\max}$ and $z^{\min}$:

$$H^{\text{M\&M}} \triangleq \alpha^{\max} H_{p,q}(y|x, z^{\max}) + \alpha^{\min} H_{p,q}(y|x, z^{\min}) = \alpha^{\max} H_{p,q}^{\max}(y|x) + \alpha^{\min} H_{p,q}^{\min}(y|x).$$

Here the Max cross-entropy $H_{p,q}^{\max}$ and Min cross entropy $H_{p,q}^{\min}$ maximizes the correct target posterior and and minimizes the incorrect target posterior, respectively. Similar to the cross-entropy loss, the Max-Min cross-entropy can also be shown to be an upper bound for the negative conditional log-likelihood $L_{\text{sup}}$ of the NRM and has the same generalization bound derived in Section 2.4. The Max-Min networks in Figure 4

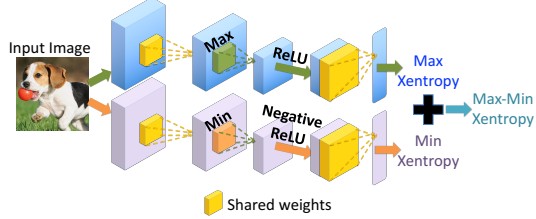

Figure 4: The Max-Min network

realize this new loss. These networks have two CNN-like branches that share weights. The max branch estimates $z^{\max}$ using ReLU and Max-Pooling, and the min branch estimates $z^{\min}$ using the Negative ReLU, i.e., $\min(\cdot, 0)$, and Min-Pooling. The Max-Min networks can be interpreted as a form of knowledge distillation like the Born Again networks (Furlanello et al., 2018) and the Mean Teacher networks. However, instead of a student network learning from a teacher network, in Max-Min networks, two students networks, the Max and the Min networks, cooperate and learn from each other during the training.

## 4 EXPERIMENTS

### 4.1 SEMI-SUPERVISED LEARNING

We show NRM armed with Max-Min cross-entropy and Mean Teacher regularizer achieves SOTA on benchmark datasets. We discuss the experimental results for CIFAR10 and CIFAR100 here. The results for SVHN, training losses, and training details, can be found in the Appendix A & D.

**CIFAR-10:** Table 2 shows comparable results of NRM to SOTA methods. NRM is also better than the best methods that do not use consistency regularization like GAN, Ladder network, and ALI when using only $N_l$=2K and 4K labeled images. NRM outperforms DRM in all settings. Also, among methods in our comparison, NRM achieves the best test accuracy when using all available labeled data ($N_l$=50K). NRM has the advantage over consistency regularization methods like Temporal Ensembling and Mean Teacher when there are enough labeled data because the consistency regularization in those methods tries to match the activations in the network, but does not take into account the available class labels. On the contrary, NRM employs the class labels, if they are available, in its reconstruction loss and RPN regularization as in Eqns. 7 and 8. In all settings, RPN regularizer improves NRM performance. Even though the improvement from RPN is small, it is consistent across the experiments. Furthermore, using Max-Min cross-entropy significantly reduces the test errors. When combining with Mean-Teacher, our Max-Min NRM improves upon Mean-Teacher and consistently achieves either SOTA results or second best results in all settings.

Table 2: Error rate percentage on CIFAR-10 over 3 runs.

| | 1K labels 50K images | 2K labels 50K images | 4K labels 50K images | 50K labels 50K images |
|---|---|---|---|---|
| Adversarial Learned Inference (Dumoulin et al., 2017) | $19.98 \pm 0.89$ | $19.09 \pm 0.44$ | $17.99 \pm 1.62$ | |
| Improved GAN (Salimans et al., 2016) | $21.83 \pm 2.01$ | $19.61 \pm 2.09$ | $18.63 \pm 2.32$ | |
| Ladder Network (Rasmus et al., 2015) | | | $20.40 \pm 0.47$ | |
| Π model (Laine & Aila, 2017) | $27.36 \pm 1.20$ | $18.02 \pm 0.60$ | $13.20 \pm 0.27$ | $6.06 \pm 0.11$ |
| Temporal Ensembling (Laine & Aila, 2017) | | | $12.16 \pm 0.31$ | $5.60 \pm 0.10$ |
| Mean Teacher (Tarvainen & Valpola, 2017) | $21.55 \pm 1.48$ | $15.73 \pm 0.31$ | $12.31 \pm 0.28$ | $5.94 \pm 0.15$ |
| VAT+EntMin (Miyato et al., 2018) | | | **10.55** | |
| DRM (Patel et al., 2016; 2015) | $27.67 \pm 1.86$ | $20.71 \pm 0.30$ | $15.36 \pm 0.34$ | $5.75 \pm 0.24$ |
| Supervised-only | $46.43 \pm 1.21$ | $33.94 \pm 0.73$ | $20.66 \pm 0.57$ | $5.82 \pm 0.15$ |
| NRM without RPN | $24.88 \pm 0.76$ | $18.97 \pm 0.80$ | $14.41 \pm 0.19$ | $5.57 \pm 0.07$ |
| NRM+RPN | $24.48 \pm 0.43$ | $18.62 \pm 0.70$ | $14.18 \pm 0.46$ | $5.35 \pm 0.08$ |
| NRM+RPN+Max-Min | $21.55 \pm 0.46$ | $16.24 \pm 0.17$ | $12.50 \pm 0.35$ | **4.85 ± 0.10** |
| NRM+RPN+Max-Min+Mean Teacher | **19.79 ± 0.74** | **15.11 ± 0.51** | $11.81 \pm 0.13$ | $4.88 \pm 0.09$ |

This consistency in performance is only observed in our method and Mean-Teacher. Also, like with Mean-Teacher, NRM can potentially be combined with other consistency regularization methods, e.g., the Virtual Adversarial Training (VAT) (Miyato et al., 2018), to obtain better results.

**CIFAR-100:** Table 3 shows NRM's comparable results to Π model and Temporal Ensembling, as well as better results than DRM. Same as with CIFAR10, using the RPN regularizer results in a slightly better test accuracy, and NRM achieves better results than Π model and Temporal Ensembling method when using all available labeled data.

Notice that combining with Mean-Teacher just slightly improves NRM's performance when training with 10K labeled data. This is again because consistency regularization methods like Mean-Teacher do not add much advantage when there are enough labeled data. However, NRM+Max-Min still yields

Table 3: Error rate percentage on CIFAR-100 over 3 runs.

| | 10K labels 50K images | 50K labels 50K images |
|---|---|---|
| Π model (Laine & Aila, 2017) | $39.19 \pm 0.36$ | $26.32 \pm 0.04$ |
| Temporal Ensembling (Laine & Aila, 2017) | $38.65 \pm 0.51$ | $26.30 \pm 0.15$ |
| DRM (Patel et al., 2016; 2015) | $41.09 \pm 0.31$ | $27.06 \pm 0.19$ |
| Supervised-only | $44.56 \pm 0.30$ | $26.42 \pm 0.17$ |
| NRM without RPN | $40.70 \pm 1.13$ | $26.27 \pm 0.09$ |
| NRM+RPN | $39.85 \pm 0.46$ | $25.84 \pm 0.10$ |
| NRM+RPN+Mean Teacher | $39.84 \pm 0.32$ | $25.98 \pm 0.35$ |
| NRM+RPN+Max-Min | **37.75 ± 0.66** | **24.38 ± 0.29** |

better test errors and achieves SOTA result in all settings.

## 4.2 SUPERVISED LEARNING WITH MAX-MIN CROSS-ENTROPY

The Max-Min cross-entropy can be applied not only to improve semi-supervised learning on deep models including CNNs but also to enhance their supervised learning performance. In our experiments, we indeed observe Max-Min cross-entropy reduces the test error for supervised object classification on CIFAR10. In particular, using the Max-Min cross-entropy loss on a 29-layer ResNet (Xie et al., 2017) trained with the Shake-Shake regularization (Gastaldi, 2017) and Cutout data augmentation (DeVries & Taylor, 2017), we are able to achieve SOTA test error of 2.30% on CIFAR10, an improvement of 0.26% over the test error of the baseline architecture trained with the traditional cross-entropy loss. While 0.26% improvement seems small, it is a meaningful enhancement given that our baseline architecture (ResNeXt + Shake-Shake + Cutout) is the second best model for supervised learning on CIFAR10. Such small improvement over an already very accurate model is significant in applications in which high accuracy is demanded such as self-driving cars or medical diagnostics. Similarly, we observe Max-Min improves the top-5 test error of the Squeeze-and-Excitation ResNeXt-50 network (Hu et al., 2018) on ImageNet by 0.17% compared to the baseline (7.04% vs. 7.21%). For a fair comparison, we re-train the baseline models and report the scores in the re-implementation.

## 5 CONCLUSIONS

We present the NRM, a general and an effective framework for semi-supervised learning that combines generation and prediction in an end-to-end optimization. Using NRM, we can explain operations used in CNNs and develop new features that help learning in CNNs. For example, we derive the new Max-Min cross-entropy loss for training CNNs, which outperforms the traditional cross-entropy. Despite promising results in this paper, NRM can still be improved. For instance, an adversarial loss like in GANs can be incorporated into NRM so that the model can generate realistic images. Furthermore, more knowledge of image generation from graphics and physics can be integrated in NRM so that the model can employ more structures to help learning and generation.

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

# Supplementary Material

# Appendix A

This appendix contains several key tables with simulation studies and notations as well as figures for the connection between NRM and CNN in the main text.

### Semi-Supervised Learning Results on SVHN

Table 4: Error rate percentage on SVHN in comparison with other state-of-the-art methods. All results are averaged over 2 runs (except for NRM+RPN when using all labels, 1 run)

| | 250 labels 73257 images | 500 labels 73257 images | 1000 labels 73257 images | 73257 labels 73257 images |
|---|---|---|---|---|
| ALI Dumoulin et al. (2017) | | | $7.42 \pm 0.65$ | |
| Improved GAN (Salimans et al., 2016) | | $18.44 \pm 4.8$ | $8.11 \pm 1.3$ | |
| + Jacob.-reg + Tangents (Kumar et al., 2017) | | $4.87 \pm 1.60$ | $4.39 \pm 1.20$ | |
| $\Pi$ model (Laine & Aila, 2017) | $9.69 \pm 0.92$ | $6.83 \pm 0.66$ | $4.95 \pm 0.26$ | $2.50 \pm 0.07$ |
| Temporal Ensembling (Laine & Aila, 2017) | | $5.12 \pm 0.13$ | $4.42 \pm 0.16$ | $2.74 \pm 0.06$ |
| Mean Teacher (Tarvainen & Valpola, 2017) | $4.35 \pm 0.50$ | $4.18 \pm 0.27$ | $3.95 \pm 0.19$ | $\mathbf{2.50 \pm 0.05}$ |
| VAT+EntMin (Miyato et al., 2018) | | | $3.86$ | |
| DRM (Nguyen et al., 2016) | | $9.85$ | $6.78$ | |
| Supervised-only | $27.77 \pm 3.18$ | $16.88 \pm 1.30$ | $12.32 \pm 0.95$ | $2.75 \pm 0.10$ |
| NRM without RPN | $9.78 \pm 0.24$ | $7.42 \pm 0.61$ | $5.64 \pm 0.13$ | $3.46 \pm 0.04$ |
| NRM+RPN | $9.28 \pm 0.01$ | $6.56 \pm 0.88$ | $5.47 \pm 0.14$ | $3.57$ |
| NRM+RPN+Max-Min+MeanTeacher | $\mathbf{3.97 \pm 0.21}$ | $\mathbf{3.84 \pm 0.34}$ | $\mathbf{3.70 \pm 0.04}$ | $2.87 \pm 0.05$ |

### Rendering Process in NRM

---

**Algorithm 1** Rendering Process in NRM

---

**Input:** Object category $y$.
**Output:** Rendered image $x$ given the object category $y$.
**Parameters:** $\theta = \left( \{\mu(y)\}_{y=1}^{K}, \{\Gamma(\ell)\}_{\ell=1}^{L}, \{b(\ell)\}_{\ell=1}^{L} \right)$ where $\mu(y)$ is the class template, $\Gamma(\ell)$ is the rendering template at layer $\ell$, and $b(\ell)$ is the parameters of the conjugate prior $p(z|y, x)$ at layer $\ell$, which turn out to be the bias terms in the ReLU after convolutions at each layer in CNNs.

1. Use Markov chain Monte Carlo method to sample the latent variables $z(\ell)$ in NRM, $\ell = 1, 2, \ldots, L$ from $\pi_{z|y} = \text{Softmax}\left(\frac{1}{\sigma^2}\eta(y, z)\right)$, where $\eta(y, z) \triangleq \sum_{\ell=1}^{L} \langle b(t; \ell), s(\ell) \odot h(\ell) \rangle$.

2. Render $h(\ell)$, $\ell = 0, 1, \ldots, L - 1$ using the recursion $h(\ell - 1) = \sum_{p \in \mathcal{P}(\ell)} s(\ell, p) T(t; \ell, p) B(\ell, p) \Gamma(\ell, p) h(\ell, p)$ in Eqn. 3, in which $h(L) = \mu(y)$ and $T(t; \ell, p)$ and $B(\ell, p)$ are the local translation matrix and the zero-padding matrix at pixel location $p$ in layer $\ell$ as described above.

3. Add Gaussian pixel noise $\mathcal{N}(0, \sigma^2 \mathbf{1}_{D(0)})$ into $h(0)$ to achieve the final rendered image $x$, where $D(0)$ is the dimension, i.e. the number of pixels, of $h(0)$, as well as of $x$.

---

Table 5: Table of notations for NRM

Variables

| | | |
|---:|:---:|:---|
| $x$ | $\triangleq$ | input image of size $D(0)$ |
| $y$ | $\triangleq$ | object category |
| $z(\ell) = \{s(\ell), t(\ell)\}$ | $\triangleq$ | all latent variables of size $D(\ell)$ in level $\ell$ |
| $s(\ell, p)$ | $\triangleq$ | switching latent variable at pixel location $p$ in level $\ell$ |
| $t(\ell, p)$ | $\triangleq$ | local translation latent variable at pixel location $p$ in level $\ell$ |
| $h(y, z; \ell) = h(\ell)$ | $\triangleq$ | intermediate rendered image of size $D(\ell)$ in level $\ell$ |
| $h(y, z; 0) = h(0)$ | $\triangleq$ | rendered image of size $D(0)$ from NRM before adding noise |
| $\psi(\ell)$ | $\triangleq$ | corresponding feature maps in layer $\ell$ in CNNs. |

Parameters

| | | |
|---:|:---:|:---|
| $\mu(y) = h(y; L) = h(L)$ | $\triangleq$ | the template of class $y$, as well as the coarsest image of size $D(L)$ determined by the category $y$ at the top of NRM before adding any fine detail. $\mu(y)$ is learned from the data. |
| $\Lambda(\ell)$ | $\triangleq$ | rendering matrix of size $D(\ell-1) \times D(\ell)$ at layer $\ell$. |
| $\Gamma(\ell)$ | $\triangleq$ | dictionary of $D(\ell)$ rendering template $\Gamma(\ell, p)$ of size $F(\ell) \times 1$ at layer $\ell$. $\Gamma(\ell)$ is learned from the data. |
| $W(\ell) = \Gamma^\top(\ell, p)$ | $\triangleq$ | corresponding weight at the layer $\ell$ in CNNs |
| $B(\ell)$ | $\triangleq$ | set of zero-padding matrices $B(\ell, p) \in \mathbb{R}^{D(\ell-1) \times F(\ell)})$ at layer $\ell$ |
| $T(\ell)$ | $\triangleq$ | set of local translation matrices $T(\ell, p) \in \mathbb{R}^{D(\ell-1) \times D(\ell-1)})$ at layer $\ell$. $T(\ell, p)$ is chosen according to value of $t(\ell, p)$ |
| $b(t; \ell) = b(\ell)$ | $\triangleq$ | parameter of the conjugate prior $p(z|x, y)$ at layer $\ell$. This term is of size $D(\ell)$ and becomes the bias term after convolutions in CNNs. It can be made independent of $t$, which is equivalent to using the same bias in each feature map in CNNs. Here, $b(t; \ell)$ is learned from data. |
| $\pi_y$ | $\triangleq$ | probability of object category $y$. |
| $\sigma^2$ | $\triangleq$ | pixel noise variance |

Other Notations

| | | |
|---:|:---:|:---|
| $\eta(y, z)$ | $\triangleq$ | $\sum_{\ell=1}^{L} \frac{1}{\sigma^2} \langle b(t; \ell), s(\ell) \odot h(\ell) \rangle$. |
| Softmax $(\eta)$ | $\triangleq$ | $\frac{\exp(\eta)}{\sum_{\eta'} \exp(\eta')}$. |
| RPN | $\triangleq$ | $-\frac{1}{n} \sum_{i=1}^{n} \log p(z_i^*|y_i) = -\frac{1}{n} \sum_{i=1}^{n}$ Softmax $(\eta(y_i, z_i^*))$. |
| (y,z(L), …, z(1)) | $\triangleq$ | rendering configuration. |

# Appendix B

In this appendix, we give proofs for Theorem 2.3. We also give further connection of NRM to cross entropy as well as additional derivation of NRM to various models under both unsupervised and (semi)-supervised setting of data being mentioned in the main text. We also formally present our results on consistency and generalization bounds for NRM in supervised and semi-supervised learning settings. In addition, we explain how to extend NRM to derive ResNet and DenseNet. For the simplicity of the presentation, we denote $\theta = \left( \{\mu(y)\}_{y=1}^{K}, \{\Gamma(\ell)\}_{\ell=1}^{L}, \{\pi_y\}_{y=1}^{K}, \{b(\ell)\}_{\ell=1}^{L} \right)$ to represent all the parameters that we would like to estimate from NRM where $\mathcal{L}$ is the set of all possible values of latent (nuisance) variables $z = (t(\ell), s(\ell))_{\ell=1}^{L}$. Additionally, for each $(y, z) \in \mathcal{J} := \{1, \ldots, K\} \times \mathcal{L}$, we denote $\theta_{y,z} = \left( \mu(y), \{\Gamma(\ell)\}_{\ell=1}^{L}, \{b(\ell)\} \right)$, i.e., the subset of parameters corresponding to specific label $y$ and latent variable $z$. Furthermore, to stress the dependence of $\eta(y, z)$ on $\theta$, we define the following function

$$\tau(\theta_{y,z}) := \eta(y, z) = \sum_{\ell=1}^{L} b^{\top}(\ell) \left( s(\ell) \odot z(\ell) \right) = \sum_{\ell=1}^{L} b^{\top}(\ell) M(s; \ell) z(\ell)$$

for each $(y, z) \in \mathcal{J}$ where $M(s; \ell) = \text{diag}(s(\ell))$ is a masking matrix associated with $s(\ell)$. Throughout this supplement, we will use $\tau(\theta_{y,z})$ and $\eta(y, z)$ interchangeably as long as the context is clear. Furthermore, we assume that $\Gamma(\ell) \in \theta_{\ell}$, which is a subset of $\mathbb{R}^{F(\ell) \times D(\ell)}$ for any $1 \leq \ell \leq L$, $\mu(y) \in \Omega$, which is a subset of $\mathbb{R}^{D(\ell)}$, for $1 \leq y \leq K$, and $b(t; \ell) \in \Xi(\ell)$, which is a subset of $\mathbb{R}^{D(\ell)}$ for all choices of $t(\ell)$ and $1 \leq \ell \leq L$. Last but not least, we say that $\theta$ satisfies the *non-negativity assumption* if the intermediate rendered images $z(\ell)$ satisfy $z(\ell) \geq 0$ for all $1 \leq \ell \leq L$.

**Notation:** In this appendix, we use $A^{\top}$ to denote transpose of the matrix $A$.

## .1 CONNECTION BETWEEN NRM AND CROSS ENTROPY

As being established in part (a) of Theorem 2.3, the cross entropy is the upper bound of the NRM's negative conditional log likelihood. In the following full theorem, we will show both the upper bound and the lower bound of maximizing the conditional log likelihood in terms of the cross entropy.

**Theorem .1.** *Given any $\gamma > 0$, we denote $\mathcal{A}_{\gamma} = \{\theta : \|h(y, z; 0)\| = \gamma\}$. For any $n \geq 1$ and $\sigma > 0$, let $x_1, \ldots, x_n$ be i.i.d. samples from the NRM. Then, the following holds*
*(a) (Lower bound)*

$$\max_{(z_i)_{i=1}^{n}, \theta \in \mathcal{A}_{\gamma}} \frac{1}{n} \sum_{i=1}^{n} \log p(y_i | x_i, z_i; \theta)$$

$$\geq \max_{\theta \in \mathcal{A}_{\gamma}} \frac{1}{n} \sum_{i=1}^{n} \log \left( Softmax \left( \max_{z_i} \left( \frac{h^{\top}(y_i, z_i; 0)x_i + \eta(y_i, z_i)}{\sigma^2} \right) + b_{y_i} \right) \right)$$

$$= \max_{\theta \in \mathcal{A}_{\gamma}} \frac{1}{n} \sum_{i=1}^{n} \log q(y_i | x_i) = - \min_{\theta \in \mathcal{A}_{\Gamma}} H_{p,q}(y | x)$$

*where $b_y = \log \pi_y$ for all $1 \leq y \leq K$, $q(y | x) = Softmax \left( \max_{z} \left( h^{\top}(y, z; 0)x + \eta(y, z) \right) / \sigma^2 + b_y \right)$ for all $(x, y)$, and $H_{p,q}(y | x)$ is the cross-entropy between the estimated posterior $q(y | x)$ and the true posterior given by the ground-true labels $p(y | x)$.*
*(b) (Upper bound)*

$$\max_{(z_i)_{i=1}^{n}, \theta \in \mathcal{A}_{\gamma}} \frac{1}{n} \sum_{i=1}^{n} \log p(y_i | x_i, z_i; \theta)$$

$$\leq \max_{\theta \in \mathcal{A}_{\gamma}} \frac{1}{n} \sum_{i=1}^{n} \Bigg\{ \log q(y_i | x_i) + \max_{y} \Bigg( \max_{z_i} \Bigg( \frac{h^{\top}(y, z_i; 0)x_i + \eta(y, z_i)}{\sigma^2}$$

$$- \frac{h^{\top}(y, \overline{z}_i; 0)x_i + \eta(y, \overline{z}_i)}{\sigma^2} \Bigg) \Bigg) \Bigg\} + \log K$$

*where* $\overline{z}_i = \arg\max\limits_{z_i} p(y_i|x_i, z_i; \vec{\theta})$ *for* $1 \leq i \leq n$.

*Remark* .2. As being demonstrated in Theorem 2.2, $\max\limits_{z} \left(h^\top(y, z; 0)x + \eta(y, z)\right)/\sigma^2$, approximately, has the form of a CNN. If we further have the non-negativity assumption with $\theta$, then this is exact. Therefore, the cross entropy $H_{p,q}$ obtained in Theorem .1 has a strong connection with CNN.

*Remark* .3. The gap between the upper bound and the lower bound of maximizing the conditional log likelihood in terms of cross entropy function suggests how good the estimation in Theorem .1 is. In particular, this gap is given by:

$$\frac{1}{n}\sum_{i=1}^{n}\left\{\underbrace{\max_y\left(\max_{z_i}\left(\frac{h^\top(y, z_i; 0)x_i + \eta(y, z_i)}{\sigma^2}\right) - \frac{h^\top(y, \overline{z}_i; 0)x_i + \eta(y, \overline{z}_i)}{\sigma^2}\right)}_{\text{trade-off loss}}\right\} + \log K,$$

where $\overline{z}_i = \arg\max\limits_{z_i} p(y_i|x_i, z_i; \theta)$ for $1 \leq i \leq n$. As long as the number of labels is not too large and the trade-off loss is sufficiently small, the gap between the upper bound and the lower bound in Theorem .1 is small.

## .2 Learning in the NRM without noise for unsupervised setting under non-negativity assumption

To ease the presentation of the inference with NRM without noise, we first assume that rendered images $h(y, z; 0)$ satisfy the non-negativity assumption (Later in Section .3, we will discuss the relaxation of this assumption for the inference of the NRM). With this assumption, as being demonstrated in Theorem 2.2, we have:

$$\max_{y,z}\left\{h^\top(y, z; 0)x + \eta(y, z)\right\} = \max_y h^\top(y)\psi(L) \tag{10}$$

where we define

$$\psi(L) = \max_{z(L)} \Lambda^\top(z(L))\left(\max_{z(L-1)}\left(\Lambda^\top(z; L-1)\cdots\left(\max_{z(1)}\Lambda^\top(z; 1)x + b(1)\right)\cdots\right) + b(L-1)\right)$$
$$+ b(\ell).$$

Now, we will provide careful derivation of part (b) of Theorem 2.3 in the main text. Remind that, for the unsupervised setting, we have data $x_1, \ldots, x_n$ are i.i.d. samples from NRM. The complete-data log-likelihood of the NRM is given as follows:

$$\mathbb{E}_{y,z}\left[\log p(x, (y, z))\right] = \sum_{i=1}^{n}\sum_{(y,z)\in\mathcal{J}} P(y, z|x_i)\{\log \pi_{y,z} + \log \mathcal{N}(x_i|h(y, z; 0))\}$$

where we have

$$P(y, z|x_i) = \frac{\pi_{y,z}\mathcal{N}(x_i|h(y, z; 0))}{\sum\limits_{(y',z')\in\mathcal{J}}\pi_{y',z'}\mathcal{N}(x_i|h(y', z'; 0))}$$

$$= \frac{\pi_y\exp\left(-\dfrac{\|x_i - h(y, z; 0)\|^2 - 2\eta(y, z)}{2\sigma^2}\right)}{\sum\limits_{(y',z')\in\mathcal{J}}\pi_{y'}\exp\left(-\dfrac{\|x_i - h(y', z'; 0)\|^2 - 2\eta(y', z')}{2\sigma^2}\right)}.$$

At the zero-noise limit, i.e., $\sigma \to 0$, it is clear that $P(y, z|x_i) = 1$ as $(y, z) = \arg\min\limits_{(y',z')\in\mathcal{J}}\Big\{\|x_i - h(y', z'; 0)\|^2 - 2\eta(y', z')\Big\}$ and $P(y, z|x_i) = 0$ otherwise. Therefore, we can asymptotically view the complete log-likelihood of the NRM under the zero-noise limit as

$$\sum_{i=1}^{n}\sum_{(y,z)\in\mathcal{J}} r_{y,z}\left(\log \pi_{y,z} - \frac{1}{2}\|x_i - h(y, z; 0)\|^2\right)$$

$$= \underbrace{\sum_{i=1}^{n}\sum_{(y,z)\in\mathcal{J}} -\frac{1}{2}r_{y,z}\|x_n - h(y, z; 0)\|^2}_{\text{Reconstruction Loss}} + \underbrace{\sum_{i=1}^{n}\sum_{(y,z)\in\mathcal{J}} r_{y,z}\log \pi_{y,z}}_{\text{Path Normalization Regularizer}}$$

where

$$r_{y,z} \equiv \begin{cases} 1, & \text{if } (y,z) = \underset{(y',z') \in \mathcal{J}}{\arg\min} \left\{ \|x_i - h(y',z';0)\|^2 - 2\eta(y',z') \right\} \\ 0, & \text{otherwise} \end{cases}$$

With the above formulation, we have the following objective function

$$U_n = \min_\theta \frac{1}{n} \sum_{i=1}^n \sum_{(y,z) \in \mathcal{J}} r_{y,z} \left( \frac{1}{2} \|x_i - h(y,z;0)\|^2 - \log \pi_{y,z} \right) \tag{11}$$

where $\theta = \left( \{\mu(y)\}_{y=1}^K, \{\Gamma(\ell)\}_{\ell=1}^L, \{\pi_y\}_{y=1}^K, \{b(\ell)\}_{\ell=1}^L \right)$. We call the above objective function to be *unsupervised NRM without noise*.

**Relaxation of unsupervised NRM without noise**   Unfortunately, the inference with unsupervised NRM without noise is intractable in practice due to two elements: the involvement of $\|h(y',z';0)\|^2$ to determine the value of $r_{y,z}$ and the summation $\sum_{(y',z') \in \mathcal{J}} \exp(\eta(y',z') + \log \pi_{y'})$ in the denominator of $\pi_{y,z}$ for all $(y,z) \in \mathcal{J}$. Therefore, we need to develop a tractable version of this objective function.

**Theorem .4.** *(Relaxation of unsupervised NRM without noise) Assume that $\pi_y \geq \overline{\gamma}$ for all $1 \leq y \leq K$ for some given $\overline{\gamma} \in (0, 1/2)$. Denote*

$$V_n := \min_\theta \frac{1}{n} \sum_{i=1}^n \sum_{(y',z') \in \mathcal{J}} 1 \left\{ (y',z') = \underset{(y,z) \in \mathcal{J}}{\arg\max} \left( h^\top(y,z;0)x_i + \eta(y,z) \right) \right\} \left( \frac{\|x_i - h(y',z';0)\|^2}{2} - \log(\pi_{y',z'}) \right)$$

*where $p_{y',z'} = \exp\left( \eta(y',z') + \log \pi_{y'} \right) / \left( \sum_{y=1}^K \exp\left( \max_{z \in \mathcal{L}} \eta(y,z) + \log \pi_y \right) \right)$ for all $(y',z') \in \mathcal{J}$. For any $\theta$, we define*

$$(\overline{y}_i, \overline{z}_i) = \underset{(y,z) \in \mathcal{J}}{\arg\min} \left\{ \|x_i - h(y,z;0)\|^2 - 2\eta(y,z) \right\}$$

*and*

$$(\widetilde{y}_i, \widetilde{z}_i) = \underset{(y,z) \in \mathcal{J}}{\arg\max} \left( h^\top(y,z;0)x_i + \eta(y,z) \right)$$

*as $1 \leq i \leq n$. Then, the following holds*
*(a) Upper bound:*

$$U_n \leq \min_\theta \frac{1}{n} \sum_{i=1}^n \left\{ \left( \frac{\|x_i - h(\widetilde{y}_i, \widetilde{z}_i; 0)\|^2}{2} - \log(\pi_{\widetilde{y}_i, \widetilde{z}_i}) \right) + \underbrace{\left( \log \pi_{\widetilde{y}_i} - \log \pi_{\overline{y}_i} \right)}_{prior\ loss} \right\} + \log |\mathcal{L}|$$

$$\leq V_n + \log\left( \frac{1}{\overline{\gamma}} - 1 \right) + \log |\mathcal{L}|$$

*(b) Lower bound:*

$$U_n \geq \min_\theta \frac{1}{n} \sum_{i=1}^n \left\{ \left( \frac{\|x_i - h(\widetilde{y}_i, \widetilde{z}_i; 0)\|^2}{2} - \log(\pi_{\widetilde{y}_i, \widetilde{z}_i}) \right) + \underbrace{\left( \log \pi_{\overline{y}_i} - \log \pi_{\widetilde{y}_i} \right)}_{prior\ loss} \right.$$

$$\left. + \frac{1}{2} \underbrace{\left( \|h(\overline{y}_i, \overline{z}_i; 0)\|^2 - \|h(\widetilde{y}_i, \widetilde{z}_i; 0)\|^2 \right)}_{norm\ loss} \right\}$$

$$\geq V_n + \log\left( \frac{\overline{\gamma}}{1 - \overline{\gamma}} \right) + \min_\theta \frac{1}{n} \sum_{i=1}^n \frac{1}{2} \left( \|h(\overline{y}_i, \overline{z}_i; 0)\|^2 - \|h(\widetilde{y}_i, \widetilde{z}_i; 0)\|^2 \right)$$

Unlike $U_n$, the inference with objective function of $V_n$ is tractable. According to the upper bound and lower bound of $U_n$ in terms of $V_n$, we can use $V_n$ as a tractable approximation of $U_n$ for the inference purpose with unsupervised setting of data when the noise is treated to be 0. Therefore, we achieve the conclusion of part (b) of Theorem 2.3 in the main text. The algorithm for determined (local) minima of $V_n$ is summarized in Algorithm 2.

### .3 Relaxation of non-negativity assumption with rendered images

It is clear that the inference with $V_n$ relies on the non-negativity assumption such that equation equation 10 holds. Now, we will argue that when the non-negativity assumption with rendered images $h(y, z; 0)$ does not hold, we can relax $V_n$ to a more tractable version under that setting.

**Theorem .5.** *(Relaxation of objective function $V_n$ when non-negativity assumption does not hold)* *Assume that $\pi_y \geq \overline{\gamma}$ for all $1 \leq y \leq K$ for some given $\overline{\gamma} \in (0, 1/2)$. Denote*

$$W_n := \min_\theta \frac{1}{n} \sum_{i=1}^n \sum_{y'=1}^K 1_{\left\{ y' = \arg\max_{y \in \mathcal{J}} g(y, x_i) \right\}} \left( \frac{\|x_i - g(y', \overline{z}_i)\|^2}{2} - \log(\pi_{y', \overline{z}_i}) \right)$$

*where*

$$g(y, x) = h^\top(y) \, \text{MaxPool}\Bigg( \text{ReLu}\bigg( \text{Conv}\Big( \Gamma(\ell), \cdots \text{MaxPool}\Big( \text{ReLu}\Big( \text{Conv}\Big( \Gamma(1), I \Big) + b(1) \Big) \Big) \Big)$$
$$\cdots + b(\ell) \bigg) \Bigg)$$

*for all $(x, y)$. Additionally, $\overline{z}_i$ is the maximal value of $z$ in the CNN structure of $g(y, x_i)$ for $1 \leq i \leq n$. For any $\theta$, we define*

$$(\widetilde{y}_i, \widetilde{z}_i) = \arg\max_{(y,z) \in \mathcal{J}} \left( h^\top(y, z; 0) x_i + \eta(y, z) \right)$$

*and $\overline{y}_i = \arg\max_y g(y, x_i)$ as $1 \leq i \leq n$. Then, the following holds*
*(a) Upper bound:*

$$V_n \leq \min_\theta \frac{1}{n} \sum_{i=1}^n \Bigg\{ \left( \frac{\|x_i - h(\overline{y}_i, \overline{z}_i; 0)\|^2}{2} - \log(\pi_{\overline{y}_i, \overline{z}_i}) \right) + \underbrace{\left( \log \pi_{\overline{y}_i} - \log \pi_{\widetilde{y}_i} \right)}_{prior\ loss}$$
$$+ \underbrace{\frac{1}{2} \left( \|h(\widetilde{y}_i, \widetilde{z}_i; 0)\|^2 - \|h(\overline{y}_i, \overline{z}_i; 0)\|^2 \right)}_{norm\ loss} \Bigg\}$$

$$\leq W_n + \log\left( \frac{1}{\overline{\gamma}} - 1 \right) + \min_\theta \frac{1}{n} \sum_{i=1}^n \frac{1}{2} \left( \|h(\widetilde{y}_i, \widetilde{z}_i; 0)\|^2 - \|h(\overline{y}_i, \overline{z}_i; 0)\|^2 \right)$$

*(b) Lower bound:*

$$V_n \geq \min_\theta \frac{1}{n} \sum_{i=1}^n \Bigg\{ \left( \frac{\|x_i - h(\overline{y}_i, \overline{z}_i; 0)\|^2}{2} - \log(\pi_{\overline{y}_i, \overline{z}_i}) \right) + \underbrace{\left( \log \pi_{\overline{y}_i} - \log \pi_{\widetilde{y}_i} \right)}_{prior\ loss}$$
$$+ \underbrace{\frac{1}{2} \left( \|h(\widetilde{y}_i, \widetilde{z}_i; 0)\|^2 - \|h(\overline{y}_i, \overline{z}_i; 0)\|^2 \right)}_{norm\ loss} + \underbrace{\left( g(\overline{y}_i, x_i) - \left\{ h(\widetilde{y}_i, \widetilde{z}_i; 0)^\top x_i + \eta(\widetilde{y}_i, \widetilde{z}_i) \right\} \right)}_{CNN\ loss} \Bigg\}$$

$$\geq W_n + \log\left( \frac{\overline{\gamma}}{1 - \overline{\gamma}} \right) + \min_\theta \frac{1}{n} \sum_{i=1}^n \frac{1}{2} \left( \|h(\widetilde{y}_i, \widetilde{z}_i; 0)\|^2 - \|h(\overline{y}_i, \overline{z}_i; 0)\|^2 \right)$$
$$+ \min_\theta \frac{1}{n} \sum_{i=1}^n \left( g(\overline{y}_i, x_i) - \left\{ h(\widetilde{y}_i, \widetilde{z}_i; 0)^\top x_i + \eta(\widetilde{y}_i, \widetilde{z}_i) \right\} \right)$$

---

**Algorithm 2** Relaxation of unsupervised NRM without noise

---

**Input:** Data $x_i$, translation matrices $T(t; \ell)$, zero padding matrices $B(\ell)$, number of labels $K$, number of layers $L$.
**Output:** Parameters $\theta$.
Initialize $\theta = \left( \{\mu(y)\}_{y=1}^{K}, \{\Gamma(\ell)\}_{\ell=1}^{L}, \{\pi_y\}_{y=1}^{K}, \{b(\ell)\}_{\ell=1}^{L} \right)$.
**while** $\theta$ has not converged **do**
    1. E-Step: Update labels $(y, z)$ of each data
    **for** $i = 1$ **to** $n$ **do**
        $(\widehat{y}_i, \widehat{z}_i) = \arg\max\limits_{y,z} \left( h^{\top}(y, z)x_i + \log(\pi_{y,z}) \right).$
    **end for**
    2. M-Step: By using Stochastic Gradient Descent (SGD), update $\theta$ that minimizes
    $\sum\limits_{i=1}^{n} \left( \dfrac{\|x_i - h(\widehat{y}_i, \widehat{z}_i)\|^2}{2} - \log(\pi_{\widehat{y}_i, \widehat{z}_i}) \right).$
**end while**

---

The proof argument of the above theorem is similar to that of Theorem .4; therefore, it is omitted. The upper bound and lower bound of $V_n$ in terms of $W_n$ in Theorem .5 implies that we can use $W_n$ as a relaxation of $V_n$ when the non-negativity assumption with rendered images $h(y, z; 0)$ does not hold. The algorithm for achieving the (local) minima of $W_n$ is similar to Algorithm 2.

### .4 NRM WITH (SEMI)-SUPERVISED SETTING

In this section, we consider the application of NRM to the (semi)-supervised setting of the data. Under that setting, only a (full) portion of labels of data $x_1, \ldots, x_n$ is available. Without loss of generality, we also assume that the rendering path $h(y, z; 0)$ satisfies the non-negativity assumption. For the case that $h(y, z; 0)$ does not satisfy this assumption, we can argue in the same fashion as that of Theorem .5. Now, we assume that only the labels $(y_{n_1+1}, \ldots, y_n)$ are unknown for some $n_1 \geq 0$. When $n_1 = 0$, we have the supervised setting of data while we have the semi-supervised setting of data when $n - n_1$ is small. Our goal is to build a semi-supervised model based on NRM such that the clustering information from data $x_1, \ldots, x_{n_1}$ can be used efficiently to increase the accuracy of classifying the labels of data $x_{n_1+1}, \ldots, x_n$. For the sake of simple inference with that purpose, we only consider the setting of NRM when the noise goes to 0. Our idea of constructing the semi-supervised model based on NRM is inspired by an approximation of the upper bound of maximizing the conditional log likelihood of NRM in terms of the cross entropy and reconstruction loss in part (b) of Theorem 2.3. In particular, we combine the tractable version of reconstruction loss from the unsupervised setting in Theorem 2.3b and the cross entropy of approximate posterior in Theorem 2.3a, which can be formulated as follows

$$\min_{\theta} \frac{\alpha_{\text{RC}}}{n} \Bigg\{ \sum_{i=1}^{n_1} \sum_{(y',z') \in \mathcal{J}} 1 \left\{ (y',z') = \arg\max_{(y,z) \in \mathcal{J}} \left( h^{\top}(y,z;0)x_i + \tau(\theta_{y,z}) \right) \right\} \left( \frac{\|x_i - h(y',z';0)\|^2}{2} - \log(\pi_{y',z'}) \right)$$

$$+ \left( \sum_{i=n_1+1}^{n} \sum_{z' \in \mathcal{L}} 1 \left\{ z' = \arg\max_{z \in \mathcal{L}} \left( h^{\top}(y_i,z;0)x_i + \tau(\theta_{y_i,z}) \right) \right\} \left( \frac{\|x_i - h(y_i,z';0)\|^2}{2} - \log(\pi_{y_i,z'}) \right) \right\}$$

$$- \frac{\alpha_{\text{CE}}}{n - n_1} \sum_{i=n_1+1}^{n} \log q_{\theta}(y_i | x_i)$$

where $\alpha_{\text{RC}}$ and $\alpha_{\text{CE}}$ are non-negative weights associated with reconstruction loss and cross entropy respectively. Additionally, the approximate posterior $q_{\theta}(y|x_i)$ is chosen as

$$q_{\theta}(y|x_i) = \frac{\exp\left( \max\limits_{z \in \mathcal{L}} \left\{ h^{\top}(y,z;0)x_i + \tau(\theta_{y,z}) \right\} + \log \pi_y \right)}{\sum\limits_{y'=1}^{K} \exp\left( \max\limits_{z \in \mathcal{L}} \left\{ h^{\top}(y',z;0)x_i + \tau(\theta_{y',z}) \right\} + \log \pi_{y'} \right)}.$$

Note that, since the labels $(y_{n_1+1}, \ldots, y_n)$ are known, the reconstruction loss for clustering data $x_{n_1+1}, \ldots, x_n$ in the above objective function indeeds incorporate these information to improve the accuracy of estimating the parameters. We call the above objective function to be *(semi)-supervised NRM without noise*.

**Boosting the accuracy of (semi)-supervised NRM without noise**   In practice, it may happen that the accuracy of classifying data by using the parameters from (semi)-supervised NRM without noise is not very high. To account for that problem, we consider the following general version of (semi)-supervised NRM without noise that includes the variational inference term and the moment matching term

$$
\min_\theta \frac{\alpha_{\text{RC}}}{n} \Bigg\{ \sum_{i=1}^{n_1} \sum_{(y',z') \in \mathcal{J}} \mathbb{1}_{\left\{ (y',z') = \underset{(y,z) \in \mathcal{J}}{\arg\max} \left( h^\top(y,z;0)x_i + \tau(\theta_{y,z}) \right) \right\}} \left( \frac{\|x_i - h(y',z';0)\|^2}{2} - \log(\pi_{y',z'}) \right)
$$
$$
+ \left( \sum_{i=n_1+1}^{n} \sum_{z' \in \mathcal{L}} \mathbb{1}_{\left\{ z' = \underset{z \in \mathcal{L}}{\arg\max} \left( h^\top(y_i,z;0)x_i + \tau(\theta_{y_i,z}) \right) \right\}} \left( \frac{\|x_i - h(y_i,z';0)\|^2}{2} - \log(\pi_{y_i,z'}) \right) \right) \Bigg\}
$$
$$
- \frac{\alpha_{\text{CE}}}{n - n_1} \sum_{i=n_1+1}^{n} \log q_\theta(y_i|x_i) + \frac{\alpha_{\text{KL}}}{n} \sum_{i=1}^{n} \sum_{y=1}^{K} q_\theta(y|x_i) \log\left( \frac{q_\theta(y|x_i)}{\pi_y} \right)
$$
$$
+ \alpha_{\text{MM}} \sum_{\ell=1}^{L} \text{D}_{\text{KL}} \left( \mathcal{N}(\mu_{h(\ell)}, \sigma_{h(\ell)}^2) || \mathcal{N}(\mu_{\psi(\ell)}, \sigma_{\psi(\ell)}^2) \right). \tag{12}
$$

Here, $\alpha_{\text{KL}}$ and $\alpha_{MM}$ are non-negative weights associated with the variational inference loss and moment matching loss respectively. Additionally, $\mu_{h(\ell)}, \sigma_{h(\ell)}^2, \mu_{\psi(\ell)}, \sigma_{\psi(\ell)}^2$ in the moment matching loss are defined as follows:

$$
\mu_{h(\ell)} = \frac{1}{n} \sum_{i=1}^{n} \hat{h}(\ell)_i, \qquad\qquad \sigma_{h(\ell)}^2 = \frac{1}{n} \sum_{i=1}^{n} (\hat{h}(\ell)_i - \mu_{h(\ell)})^2
$$
$$
\mu_{\psi(\ell)} = \frac{1}{n} \sum_{i=1}^{n} \psi(\ell)_i, \qquad\qquad \sigma_{\psi(\ell)}^2 = \frac{1}{n} \sum_{i=1}^{n} (\psi(\ell)_i - \mu_{\psi(\ell)})^2 \tag{13}
$$

where $\hat{h}(\ell)_i$ is the estimated value of $h(\ell)$ given the optimal latent variables $\hat{s}^{(\ell)}$ and $\hat{t}^{(\ell)}$ inferred from the image $x_i$ for $1 \leq i \leq n$. It is clear that when $\alpha_{\text{KL}} = \alpha_{MM} = 0$, we return to (semi)-supervised NRM without noise. In Appendix C, we provide careful theoretical analyses regarding statistical guarantees of model equation 12.

Now, we will provide heuristic explanations about the improvement in terms of performance of model equation 12 based on the variational inference term and the moment matching term.

**Regarding the variational term:** The DRMM inference algorithm developed thus far ignores uncertainty in the latent nuisance posterior $p(y, z|x)$ due to the max-marginalization over $(y, z)$ in the E-step bottom-up inference. We would like to properly account for this uncertainty for two main reasons: (i) our fundamental hypothesis is that the brain performs probabilistic inference and (ii) uncertainty accounting is very important for good generalization in the semi-supervised setting since we have very little labeled data.

One approach attempts to approximate the true class posterior $p(y|x)$ for the DRMM. We employ *variational inference*, a technique that enables the approximate inference of the latent posterior. Mathematically, for the DRMM this means we would like to approximate the true class posterior $p(y|x) \approx q(y|x)$, where the approximate posterior $q$ is restricted to some tractable family of distributions (e.g. Gaussian or Categorical). We strategically choose the tractable family to be $q(y|x) \equiv p(y|\hat{z}, x)$, where $\hat{z} \equiv \underset{z}{\arg\max}\, p(y, z|x)$. In other words, we choose $q$ to be restricted to the DRMM family of nuisance max-marginalized class posteriors. Note that this is indeed an approximation, since the true DRMM class posterior has nuisances that are *sum*-marginalized out $p(y|x) = \sum_z p(y, z|x)$, whereas the approximating variational family has nuisances that are *max*-marginalized out.

Given our choice of variational family $q$, we derive the variational term for the loss function, starting from the principled goal of minimizing the KL-distance $D_{KL}[q(y|x)||p(y|x)]$ between the true and approximate posteriors with respect to the parameters of $q$. As a result, such an optimized $q$ will tilt towards better approximating $p(y|x)$, which in turn means that it will account for *some* of the uncertainty in $p(z|x)$. The variational terms in the loss are defined as Blei et al. (2017):

$$\mathcal{L}_{VI} \equiv \mathcal{L}_{RC} + \beta_{KL}\mathcal{L}_{KL}$$
$$\equiv -\mathbb{E}_q\left[\ln p(x|y)\right] + \beta_{KL}D_{KL}[q(y|x)||p(y)]. \tag{14}$$

This term is quite similar to that used in variational autoencoders (VAE) Kingma & Welling (2013), except for two key differences: (i) here the latent variable $y$ is discrete categorical rather than continuous Gaussian and (ii) we have employed a slight relaxation of the VAE by allowing for a penalty parameter $\beta_{KL} \neq 1$. The latter is motivated by recent experimental results showing that such freedom enables optimal disentangling of the true intrinsic latent variables from the data.

**Regarding the moment matching term:** Batch Normalization can potentially be derived by normalizing the intermediate rendered images $h(\ell)$, $\ell = 1, 2, \ldots, L$ in the NRM by subtracting their means and dividing by their standard deviations under the assumption that the means and standard derivations of $h(\ell)$ are close to those of the activation $\psi(\ell)$ in the CNNs. From this intuition, in Section .4 of Appendix A, we introduce the moment-matching loss to improve the performance of the NRM/CNNs trained for semi-supervised learning tasks.

### .5 Statistical Guarantees for (Semi)-Supervised Setting

For the sake of simplicity with proof argument, we only provide detail theoretical analysis for statistical guarantee with the setting of equation 12 when the moment matching term is skipped. In particular, we are interested in the following (semi)-supervised model

$$Y_n := \min_{\theta} \frac{\alpha_{\mathrm{RC}}}{n} \Bigg\{ \sum_{i=1}^{n_1} \sum_{(y',z')\in\mathcal{J}} \mathbf{1}_{\left\{(y',z')=\arg\max_{(y,z)\in\mathcal{J}}\left(h^\top(y,z;0)x_i+\tau(\theta_{y,z})\right)\right\}}$$
$$\times \left(\frac{\|x_i - h(y',z';0)\|^2}{2} - \log(\pi_{y',z'})\right)$$
$$+ \Bigg(\sum_{i=n_1+1}^{n} \sum_{z'\in\mathcal{L}} \mathbf{1}_{\left\{z'=\arg\max_{z\in\mathcal{L}}\left(h^\top(y_i,z;0)x_i+\tau(\theta_{y_i,z})\right)\right\}} \left(\frac{\|x_i - h(y_i,z';0)\|^2}{2} - \log(\pi_{y_i,z'})\right)\Bigg)\Bigg\}$$
$$- \frac{\alpha_{\mathrm{CE}}}{n-n_1} \sum_{i=n_1+1}^{n} \log q_\theta(y_i|x_i) + \frac{\alpha_{\mathrm{KL}}}{n} \sum_{i=1}^{n} \sum_{y=1}^{K} q_\theta(y|x_i)\log\left(\frac{q_\theta(y|x_i)}{\pi_y}\right) \tag{15}$$

where the approximate posterior $q_\theta(y|x_i)$ is chosen as

$$q_\theta(y|x_i) := \frac{\exp\left(\max_{z\in\mathcal{L}}\left\{h^\top(y,z;0)x_i+\tau(\theta_{y,z})\right\} + \log\pi_y\right)}{\sum_{y'=1}^{K} \exp\left(\max_{z\in\mathcal{L}}\left\{h^\top(y',z;0)x_i+\tau(\theta_{y',z})\right\} + \log\pi_{y'}\right)}.$$

Here, $\alpha_{\mathrm{RC}}$, $\alpha_{\mathrm{CE}}$, and $\alpha_{\mathrm{KL}}$ are non-negative weights associated with reconstruction loss, cross entropy, and variational inference respectively. As being indicated in the formulation of objection function $Y_n$, the only difference between $Y_n$ and equation 12 is the weight $\alpha_{\mathrm{MM}}$ regarding moment matching loss in equation 12 is set to be 0. To ease the presentation with theoretical analyses later, we call the objective function with $Y_n$ to be *partially labeled latent dependence regularized cross entropy (partially labeled LDCE).*

**Consistency of partially labeled LDCE**   Firstly, we demonstrate that the objective function of partially labeled LDCE enjoys the consistency guarantee.

**Theorem .6.** *(Consistency of objective function of partially labeled LDCE) Assume that $n_1$ is a function of $n$ such that $n_1/n \to \bar{\lambda}$ as $n \to \infty$. Furthermore, $\mathbb{P}(\|x\| \leq R) = 1$ as $x \sim P$ for some*

*given $R > 0$. We denote the population version of partially labeled LDCE as follows*

$$\overline{Y} := \min_{\theta} \alpha_{RC} \Bigg\{ \overline{\lambda} \bigg( \int \sum_{(y',z') \in \mathcal{J}} 1\left\{ (y',z') = \arg\max_{(y,z) \in \mathcal{J}} \left( h^{\top}(y,z;0)x + \tau(\theta_{y,z}) \right) \right\} \left( \frac{\|x - h(y',z';0)\|^2}{2} \right)$$

$$- \log(\pi_{y',z'}) \bigg) dP(x) \bigg) + (1 - \overline{\lambda}) \bigg( \int \sum_{z' \in \mathcal{L}} 1\left\{ z' = \arg\max_{z \in \mathcal{L}} \left( h^{\top}(y,z;0)x + \tau(\theta_{y,z}) \right) \right\} \left( \frac{\|x - h(y,z';0)\|^2}{2} \right)$$

$$- \log(\pi_{y,z'}) \bigg) dQ(x,c) \bigg) \Bigg\} - \alpha_{CE} \int \log q_{\theta}(y|x) dQ(x,c) + \alpha_{KL} \int \sum_{y=1}^{K} q_{\theta}(y|x) \log\left( \frac{q_{\theta}(y|x)}{\pi_y} \right) dP(x).$$

*Then, we obtain that $Y_n \to \overline{Y}$ almost surely as $n \to \infty$.*

The detail proof of Theorem .6 is deferred to Appendix C. Now, we denote $\widetilde{\theta} := \left( \{\widetilde{\mu}(y)\}_{y=1}^{K}, \left\{\widetilde{\Gamma}(\ell)\right\}_{\ell=1}^{L}, \{\widetilde{\pi}_y\}_{y=1}^{K}, \left\{\widetilde{b}(\ell)\right\}_{\ell=1}^{L} \right)$ the optimal solutions of objective function equation 15. Note that, the existence of these optimal solutions is guaranteed due to the compactness assumption of the parameter spaces $\Theta_\ell$, $\Omega$, and $\Xi_l$ for $1 \le \ell \le L$. The optimal solutions $\{\widetilde{\mu}(y)\}_{y=1}^{K}$ and $\left\{\widetilde{\Gamma}(\ell)\right\}_{\ell=1}^{L}$ lead to corresponding set of optimal rendered images $\widetilde{S}_n$. Similar to the case of SPLD regularized K-means, our goal is to guarantee the consistency of $\widetilde{S}_n$ as well as $\{\widetilde{\pi}_y\}_{y=1}^{K}$, $\left\{\widetilde{b}(\ell)\right\}_{\ell=1}^{L}$.

In particular, we denote $\widetilde{\mathcal{F}}_0$ the set of all optimal solutions $\widetilde{\theta}^0$ of population partially labeled LDCE where $\widetilde{\theta}^0 := \left( \{\widetilde{\mu}^0(y)\}_{y=1}^{K}, \left\{\widetilde{\Gamma}_0(\ell)\right\}_{\ell=1}^{L}, \{\widetilde{\pi}_y^0\}_{y=1}^{K}, \left\{\widetilde{b}_0(\ell)\right\}_{\ell=1}^{L} \right)$. For each $\widetilde{\theta}^0 \in \widetilde{\mathcal{F}}_0$, we define $\widetilde{S}_0$ the set of optimal rendered images associated with $\widetilde{\theta}^0$. We denote $\mathcal{G}(\widetilde{\mathcal{F}}_0)$ the corresponding set of all optimal rendered images $\widetilde{S}_0$, optimal prior probabilities $\{\widetilde{\pi}_y^0\}_{y=1}^{K}$, and optimal biases $\left\{\widetilde{b}_0(\ell)\right\}_{\ell=1}^{L}$.

**Theorem .7.** *(Consistency of optimal rendering paths and optimal solutions of partially labeled LDCE) Assume that $\mathbb{P}(\|x\| \le R) = 1$ as $x \sim P$ for some given $R > 0$. Then, we obtain that*

$$\inf_{(\widetilde{S}_0, \{\widetilde{\pi}_y^0\}, \{\widetilde{b}_0(\ell)\}) \in \mathcal{G}(\widetilde{\mathcal{F}}_0)} \left\{ H(\widetilde{S}_n, \widetilde{S}_0) + \sum_{y=1}^{K} |\widetilde{\pi}_y - \widetilde{\pi}_y^0| + \sum_{\ell=1}^{L} \|\widetilde{b}(\ell) - \widetilde{b}_0(\ell)\| \right\} \to 0$$

*almost surely as $n \to \infty$.*

The detail proof of Theorem .7 is postponed to Appendix C.

### .6 GENERALIZATION BOUND FOR CLASSIFICATION FRAMEWORK WITH (SEMI)-SUPERVISED SETTING

In this section, we provide a simple generalization bound for certain classification function with the optimal solutions $\widetilde{\theta} = \left( \{\widetilde{\mu}(y)\}_{y=1}^{K}, \left\{\widetilde{\Gamma}(\ell)\right\}_{\ell=1}^{L}, \{\widetilde{\pi}_y\}_{y=1}^{K}, \left\{\widetilde{b}(\ell)\right\}_{\ell=1}^{L} \right)$ of equation 12. In particular, we denote the following function $f : \mathbb{R}^{D^{(0)}} \times \{1, \ldots, K\} \to \mathbb{R}$ as

$$f(x,y) = \max_{z \in \mathcal{L}} \left\{ \widetilde{h}^{\top}(y,z;0)x + \tau(\widetilde{\theta}_{y,z}) \right\} + \log \widetilde{\pi}_y$$

for all $(x,y) \in \mathbb{R}^{D^{(0)}} \times \{1, \ldots, K\}$ where

$$\widetilde{h}(y,z;0) := \widetilde{\Lambda}(z;1) \ldots \widetilde{\Lambda}(z;L)\widetilde{\mu}(y),$$

$$\widetilde{\Lambda}(z;\ell) := \sum_{p \in \mathcal{P}(\ell)} s(\ell,p)T(t;\ell,p)B(\ell,p)\widetilde{\Gamma}(\ell,p)$$

for all $(y,z)$ and $1 \le \ell \le L$. To achieve the generalization bound regarding that classification function, we rely on the study of generalization bound with margin loss. For the simplicity of

argument, we assume that the true labels of $x_1, \ldots, x_n$ are $y_1, \ldots, y_n$ while $y_1, \ldots, y_{n_1}$ are not available to train. The margin of a labeled example $(x, y)$ based on $f$ can be defined as

$$\rho(f, x, y) = f(x, y) - \max_{l \neq y} f(x, l).$$

Therefore, the classification function $f$ misspecifies the labeled example $(x, y)$ as long as $\rho(f, x, y) \leq 0$. The empirical margin error of $f$ at margin coefficient $\Gamma \geq 0$ is

$$R_{n, \gamma}(f) = \frac{1}{n} \sum_{i=1}^{n} 1_{\{\rho(f, x_i, y_i) \leq \gamma\}}.$$

It is clear that $R_{n,0}(f)$ is the empirical risk of 0-1 loss, i.e., we have

$$R_{n,0}(f) = \frac{1}{n} \sum_{i=1}^{n} 1_{\left\{ \underset{1 \leq y \leq K}{\arg\max} f(x_i, y) \neq y_i \right\}}.$$

Similar to the argument in the case of SPLD regularized K-means, the optimal solutions $\widetilde{\theta}$ of partially labeled LDCE lead to a set of rendered images $\widetilde{h}(y, z; 0)$ for all $(y, z) \in \mathcal{J}$. However, only a small fraction of rendering paths are indeed active in the following sense. There exists a subset $\mathcal{L}_n$ of $\mathcal{L}$ such that $|\mathcal{L}_n| \leq \overline{\tau}_n |\mathcal{L}|$ where $\overline{\tau}_n \in (0, 1]$, which is independent of data $(x_1, y_1), \ldots, (x_n, y_n)$, and the following holds

$$\max_{z \in \mathcal{L}} \left\{ \widetilde{h}^{\top}(y, z; 0) x + \tau(\widetilde{\theta}_{y,z}) \right\} + \log \widetilde{\pi}_y = \max_{z \in \mathcal{L}_n} \left\{ \widetilde{h}^{\top}(y, z; 0) x + \tau(\widetilde{\theta}_{y,z}) \right\} + \log \widetilde{\pi}_y$$

for all $1 \leq y \leq K$. The above equation implies that

$$R_{n, \Gamma}(f) = R_{n, \gamma}(f_{\overline{\gamma}_n})$$

for all $\Gamma \geq 0$ and $n \geq 1$ where $f_{\overline{\tau}_n}(x, y) = \max_{z \in \mathcal{L}_n} \left\{ \widetilde{h}^{\top}(y, z; 0) x + \tau(\widetilde{\theta}_{y,z}) \right\} + \log \widetilde{\pi}_y$ for all $(x, y)$.

With that connection, we denote the expected margin error of classification function $f_{\overline{\tau}_n}$ at margin coefficient $\Gamma \geq 0$ is

$$R_{\gamma}(f_{\overline{\tau}_n}) = \mathbb{E} 1_{\{\rho(f_{\overline{\tau}_n}, x, y) \leq \gamma\}}.$$

The generalization bound that we establish in this section will base on the gap between the expected margin error $R_0(f_{\overline{\tau}_n})$ and its corresponding empirical version $R_{n, \Gamma}(f_{\overline{\tau}_n})$, which is also $R_{n, \Gamma}(f)$.

**Theorem .8.** *(Generalization bound for margin-based classification) Assume that $P(\|x\| \leq R) = 1$ for some given $R > 0$ and $x \sim P$. Additionally, the parameter spaces $\Theta_\ell$ and $\Omega$ are chosen such that $\|h(y, z; 0)\| \leq R$ for all $(y, z) \in \mathcal{J}$. For any $\delta > 0$, with probability at least $1 - \delta$, we have*

$$R_0(f_{\overline{\tau}_n}) \leq \inf_{\gamma \in (0,1]} \left\{ R_{n, \gamma}(f_{\overline{\tau}_n}) + \frac{8K(2K-1)}{\Gamma \sqrt{n}} \left( 2\overline{\tau}_n |\mathcal{L}| (R^2 + 1) + |\log \overline{\gamma}| \right) \right.$$
$$\left. + \left( \frac{\log \log_2(2\gamma^{-1})}{n} \right)^{1/2} + \sqrt{\frac{\log(2\delta^{-1})}{2n}} \right\}$$

*where $\overline{\gamma}$ is the lower bound of prior probability $\pi_y$ for all $y$.*

*Remark .9.* The result of Theorem .8 gives a simple characterization for the generalization bound of classification setup from optimal solutions of partially labeled LDCE based on the number of active rendering paths, which is inherent to the structure of NRM. Such dependence of generalization bound on the number of active rendering configurations $\overline{\tau}_n |\mathcal{L}|$ is rather interesting and may provide a new perspective on understanding the generalization bound. Nevertheless, there are certain limitations regarding the current generalization gap: (1) the active ratio $\overline{\tau}_n$ may change with the sample size unless we put certain constraints on the sparsity of switching variables $a$ to reduce the number of active optimal rendering configurations; (2) the generalization bound is depth- dependent due to the involvement of the number of rendering configurations $|\mathcal{L}|$. This is mainly because we have not fully taken into account all the structures of CNNs for the studying of generalization bound. Given some current progress on depth-independent generalization bound (Bartlett et al., 2017; Golowich et al., 2018), it is an interesting direction to explore whether the techniques in these work can be employed to improve $|\mathcal{L}|$ in the generalization bound in Theorem .8.

## .7 NEURAL RENDERING MODEL IS THE UNIFYING FRAMEWORK FOR NETWORKS IN THE CONVNET FAMILY

The structure of the rendering matrices $\Lambda(\ell)$ gives rise to MaxPooling, ReLU, and convolution operators in the CNNs. By modifying the structure of $\Lambda(\ell)$, we can derive different types of networks in the convolutional neural network family. In this section, we define and explore several other interesting variants of NRM: the Residual NRM (ResNRM) and the Dense NRM (DenseNRM). Inference algorithms in these NRMs yield ResNet He et al. (2016) and DenseNet Huang et al. (2017), respectively. Proofs for these correspondence are given in Appendix C. Both ResNet and DenseNet are among state-of-the-art neural networks for object recognition and popularly used for other visual perceptual inference tasks. These two architectures employ skip connections (a.k.a., shortcuts) to create short paths from early layers to later layers. During training, the short paths help avoid the vanishing-gradient problem and allow the network to propagate and reuse features.

### .7.1 RESIDUAL NEURAL RENDERING MODEL YIELDS RESNET

In a ResNet, layers learn residual functions with reference to the layer inputs. In particular, as illustrated in Fig. 5, layers in a ResNet are reformulated to represent the mapping $F(\psi) + W_{\text{skip}}\psi$ and the layers try to fit the residual mapping $F(\psi)$ where $\psi$ is the input feature. The term $W_{\text{skip}}\psi$ accounts for the skip connections/shortcuts He et al. (2016). In order to derive the ResNet, we rewrite the rendering matrix $\Lambda(\ell)$ as the sum of a shortcut matrix $\Lambda_{\text{skip}}(\ell)$ and a rendering matrix, both of which can be updated during the training. The shortcut matrices yields skip connections in He et al. (2016). Note that $\Lambda_{\text{skip}}(\ell)$ depends on the template selecting latent variables $s(\ell)$. In the rest of this section, for clarity, we will refer to $\Lambda(\ell)$ and $\Lambda_{\text{skip}}(\ell)$ as $\Lambda(t, s; \ell)$ and $\Lambda_{\text{skip}}(s; \ell)$, respectively, to show their dependency on latent variables in NRM. We define the Residual Neural Rendering Model as follows:

**Definition .10.** The Residual Neural Rendering Model (ResNRM) is the Neural Rendering Model whose rendering process from layer $\ell$ to layer $\ell - 1$, for some $\ell \in \{1, 2, \cdots, L\}$, has the residual form as follows:

$$h(\ell - 1) := (\Lambda(t, s; \ell) + \Lambda_{\text{skip}}(s; \ell)) \, h(\ell), \tag{16}$$

where $\Lambda_{\text{skip}}(t, s; \ell)$ is the shorcut matrices that results in skip connections in the corresponding ResNet. In particular, $\Lambda_{\text{skip}}(t, s; \ell)$ has the following form:

$$\Lambda_{\text{skip}}(s; \ell) = \tilde{\Lambda}_{\text{skip}}(\ell) M(s; \ell), \tag{17}$$

where $M(s; \ell) \equiv \text{diag}\,(s(\ell)) \in \mathbb{R}^{D(\ell) \times D(\ell)}$ is a diagonal matrix whose diagonal is the vector $s(\ell)$. This matrix selects the templates for rendering. Furthermore, $\tilde{\Lambda}_{\text{skip}}(\ell)$ is a rendering matrix that is independent of latent variables $t$ and $s$.

The following theorem show that similar to how CNNs can be derived from NRM, ResNet can be derived as a bottom-up inference in ResNRM.

**Theorem .11.** Inference in ResNRM yields skip connetions. In particular, if the rendering process at layer $\ell$ has the residual form as in Definition .10, the inference at this layer takes the following form:

$$\psi(\ell) \equiv \max_{t(\ell),s(\ell)} \left\{ \left( \Lambda^\top(t, s; \ell) + \Lambda_{\text{skip}}^\top(s; \ell) \right) \psi(\ell - 1) + b(\ell) \right\}$$

$$= \text{MaxPool} \left( \text{ReLu} \left( \text{Conv}(\Gamma^\top(\ell), \psi(\ell - 1)) + b(\ell) + \underbrace{\tilde{\Lambda}_{\text{skip}}^\top(\ell)\psi(\ell - 1)}_{\text{skip connection}} \right) \right)$$

$$\stackrel{d}{=} \text{MaxPool} \left( \text{ReLu} \left( \text{Conv}(W(\ell), \psi(\ell - 1)) + b(\ell) + \underbrace{W_{\text{skip}}(\ell)\psi(\ell - 1)}_{\text{skip connection}} \right) \right). \tag{18}$$

Here, when $\psi(\ell - 1)$ and $\psi(\ell)$ have the same dimensions, $\tilde{\Lambda}_{\text{skip}}(\ell)$ is chosen to be an constant identity matrix in order to derive the parameter-free, identity shortcut among layers of the same size in the ResNet. When $\psi(\ell - 1)$ and $\psi(\ell)$ have the different dimensions, $\tilde{\Lambda}_{\text{skip}}(\ell)$ is chosen to be a learnable shortcut matrix which yields the projection shortcut $W_{\text{skip}}(\ell)$ among layers of different sizes in

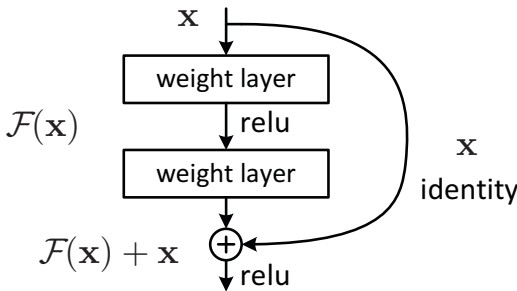

Figure 5: ResNet building block as in He et al. (2016)

the ResNet. As mentioned above, identity shortcuts and projection shortcuts are two types of skip connections in the ResNet. The operator $\stackrel{d}{=}$ implies that discriminative relaxation is applied.

In practice, the skip connections are usually across two layers or three layers. This is indeed a straightforward extension from the ResNRM. In particular, the ResNRM building block that corresponds to the ResNet building block in He et al. (2016) (see Fig. 5) takes the following form:

$$h(\ell - 2) := \left( \Lambda(t, s; \ell - 1)\Lambda(t, s; \ell) + \Lambda_{\text{skip}}(s; \ell) \right) h(\ell).$$

In inference, this ResNRM building block yields the ResNet building block in Fig. 5:

$$
\begin{aligned}
\psi(\ell) = \text{ReLu}\bigg( &\text{Conv}\Big( \Gamma^\top(\ell), \text{ReLu}\Big( \text{Conv}\left( \Gamma^\top(\ell - 1), \psi(\ell - 2) \right) + b(\ell - 1) \Big)\Big) \\
&+ b(\ell) + \tilde{\Lambda}_{\text{skip}}^\top(\ell)\psi(\ell - 2) \bigg) \\
\stackrel{d}{=} \text{ReLu}\bigg( &\text{Conv}\Big( W(\ell), \text{ReLu}\Big( \text{Conv}\Big( W(\ell - 1), \psi(\ell - 2) \Big) + b(\ell - 1) \Big)\Big) \\
&+ b(\ell) + W_{\text{skip}}(\ell)\psi(\ell - 2) \bigg).
\end{aligned}
\tag{19}
$$

### .7.2 Dense Neural Rendering Model Yields DenseNet

In a DenseNet Huang et al. (2017), instead of combining features through summation, the skip connections concatenate features. In addition, within a building block, all layers are connected to each other (see Fig. 6). Similar to how ResNet can be derived from ResNRM, DenseNet can also be derived from a variant of NRM, which we call the Dense Neural Rendering Model (DenseNRM). In DenseNRM, the rendering matrix $\Lambda(\ell)$ is concatenated by an identity matrix. This extra identity matrix, in inference, yields the skip connections that concatenate features at different layers in a DenseNet. We define DenseNRM as follows.

**Definition .12.** The Dense Neural Rendering Model (DenseNRM) is the Neural Rendering Model whose rendering process from layer $\ell$ to layer $\ell - 1$, for some $\ell \in \{1, 2, \cdots, L\}$, has the residual form as follows:

$$h(\ell - 1) := \left[ \Lambda(t, s; \ell)h(\ell), \, \mathbf{1}_{D(\ell)}h(\ell) \right]. \tag{20}$$

We again denote $\Lambda(\ell)$ as $\Lambda(t, s; \ell)$ to show the dependency of $\Lambda(\ell)$ on the latent variables $t(\ell)$ and $s(\ell)$. The following theorem establishes the connections between DenseNet and DenseNRM.

**Theorem .13.** Inference in DenseNRM yields DenseNet building blocks. In particular, if the rendering process at layer $\ell$ has the dense form as in Definition .12, the inference at this layer takes

the following form:

$$\psi(\ell) \equiv \left[ \begin{array}{c} \max_{t(\ell),s(\ell)} \left\{ \Lambda^\top(t,s;\ell)\psi(\ell-1) + b(\ell) \right\} \\ \psi(\ell-1) \end{array} \right]$$

$$= \left[ \begin{array}{c} \mathrm{MaxPool}(\mathrm{ReLu}(\mathrm{Conv}(\Gamma^\top(\ell), \psi(\ell-1)) + b(\ell))) \\ \psi(\ell-1) \end{array} \right]$$

$$\stackrel{d}{=} \left[ \begin{array}{c} \mathrm{MaxPool}(\mathrm{ReLu}(\mathrm{Conv}(W(\ell), \psi(\ell-1)) + b(\ell))) \\ \psi(\ell-1) \end{array} \right]. \qquad (21)$$

In Eqn. 21, we concatenate the output $\mathrm{MaxPool}\,\mathrm{ReLu}(\mathrm{Conv}(W(\ell), \psi(\ell-1)) + b(\ell)))$ at layer $\ell$ with the input feature $\psi(\ell-1)$ at layer $\ell-1$ to generate the input to the next layer $\psi(\ell)$, just like in the DenseNet. Proofs for Theorem .11 and .13 can be found in Appendix B. The approach to proving Theorem .11 can be used to prove the result in Eqn. 19.

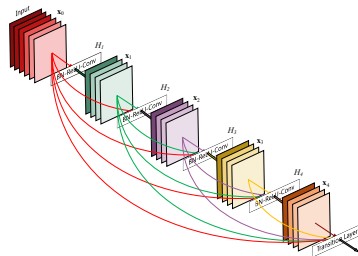

Figure 6: DenseNet building block as in Huang et al. (2017)

# Appendix C

In this appendix, we provide the proofs for key results in the paper.

## .8 Proof for Theorem 2.2: Deriving Convolutional Neural Networks from the Neural Rendering Model

To ease the clarity of the proof presentation, we ignore the normalizing factor $\frac{1}{\sigma^2}$ and only consider the proof for two layers, i.e., $L = 2$. The argument for $L \geq 3$ is similar and can be derived recursively from the proof for $L = 2$. Similar proof holds when the normalizing factor $\frac{1}{\sigma^2}$ is considered. Now, we obtain that

$$
\max_{z} \left\{ h^\top(y, z; 0)x + \eta(y, z) \right\}
$$

$$
= \max_{\substack{t(1), s(1) \\ t(2), s(2)}} \left\{ \sum_{p_1 \in \mathcal{P}(1)} h(1, p_1)s(1, p_1)\Gamma^\top(1, p_1)B^\top(1, p_1)T^\top(t; 1, p_1)x \right.
$$

$$
\left. + \sum_{p_1 \in \mathcal{P}(1)} h(1, p_1)s(1, p_1)b(t; 1, p_1) + \sum_{p_2 \in \mathcal{P}(2)} \mu(y; p_2)s(2, p_2)b(t; 2, p_2) \right\}
$$

$$
= \max_{\substack{t(1), s(1) \\ t(2), s(2)}} \left\{ \sum_{p_1 \in \mathcal{P}(1)} h(1, p_1)s(1, p_1) \left( \Gamma^\top(1, p_1)B^\top(1, p_1)T^\top(t; 1, p_1)x + b(t; 1, p_1) \right) \right.
$$

$$
\left. + \sum_{p_2 \in \mathcal{P}(2)} \mu(y; p_2)s(2, p_2)b(t; 2, p_2) \right\}
$$

$$
\overset{(a)}{=} \max_{t(2), s(2)} \left\{ \sum_{p_1 \in \mathcal{P}(1)} h(1, p_1) \max_{t(1, p_1), s(1, p_1)} s(1, p_1) \left( \Gamma^\top(1, p_1)B^\top(1, p_1)T^\top(t; 1, p_1)x + b(t; 1, p_1) \right) \right.
$$

$$
\left. + \sum_{p_2 \in \mathcal{P}(2)} \mu(y; p_2)s(2, p_2)b(t; 2, p_2) \right\} = A,
$$

where equation in (a) is due to the non-negativity assumption that $h(1, p_1) \geq 0$, just as in max-sum and max-product message passing. We define $\psi(1, p_1)$ as follows:

$$
\psi(1, p_1) = \max_{t(1, p_1), s(1, p_1)} s(1, p_1) \left( \Gamma^\top(1, p_1)B^\top(1, p_1)T^\top(t; 1, p_1)x + b(t; 1, p_1) \right),
$$

and let $\psi(1) = (\psi(1, p_1))_{p_1 \in \mathcal{P}(1)}$ be the vector of $\psi(1, p_1)$. The following holds:

$$
A = \max_{t(2), s(2)} \left\{ h^\top(1)\psi(1) + \sum_{p_2 \in \mathcal{P}(2)} \mu(y; p_2)s(2, p_2)b(t; 2, p_2) \right\}
$$

$$
\overset{(b)}{=} \max_{t(2), s(2)} \left\{ \sum_{p_2 \in \mathcal{P}(2)} \mu(y; p_2)s(2, p_2)\Gamma^\top(2, p_2)B^\top(2, p_2)T^\top(t; 2, p_2)\psi(1) \right.
$$

$$
\left. + \sum_{p_2 \in \mathcal{P}(2)} \mu(y; p_2)s(2, p_2)b(t; 2, p_2) \right\}
$$

$$
\overset{(c)}{=} \sum_{p_2 \in \mathcal{P}(2)} \mu(y; p_2) \underbrace{\max_{t(2), s(2)} s(2, p_2) \left( \Gamma^\top(2, p_2)B^\top(2, p_2)T^\top(t; 2, p_2)\psi(1) + b(t; 2, p_2) \right)}_{\psi(2, p_2)}
$$

$$
= \mu^\top(y)\psi(2).
$$

Here $\psi(2) = (\psi(2, p_2))_{p_2 \in \mathcal{P}(2)}$ is the vector of $\psi(2, p_2)$, and in line (b) we substitute $h(1)$ by:

$$
h(1) = \sum_{p \in \mathcal{P}(2)} s(2, p_2)T(t; 2, p_2)B(2, p_2)\Gamma(2, p_2)\mu(y; p_2).
$$

Notice that line (a) and (c) form a recursion. Therefore, we finish proving that the feedforward step in CNNs is the latent variable inference in NRM if $\psi(1)$ has the structure of the building block of

CNNs, i.e. $\text{MaxPool}\left(\text{ReLu}\left(\text{Conv}()\right)\right)$. Indeed,

$$
\begin{aligned}
\psi(1) &= \left(\psi(1, p(1))\right)_{p(1) \in \mathcal{P}(1)} \\
&= \left(\max_{t(1,p_1), s(1,p_1)} s(1, p_1) \left(\Gamma^\top(1, p_1) B^\top(1, p_1) T^\top(t; 1, p_1) x + b(t; 1, p_1)\right)\right)_{p(1) \in \mathcal{P}(1)} \\
&= \left(\max\left(\left(\text{ReLu}\left(\Gamma^\top(1, p_1) B^\top(1, p_1) T^\top(t; 1, p_1) x + b(t; 1, p_1)\right)\right)_{t(1,p_1)=0,1,2,3}\right)\right)_{p(1) \in \mathcal{P}(1)} \\
&\qquad\qquad\qquad\qquad\qquad\qquad\qquad\qquad\qquad\qquad\qquad\qquad\qquad\qquad\qquad (22) \\
&= \text{MaxPool}\left(\text{ReLu}\left(\text{Conv}\left(\Gamma^\top(1), x\right) + b(t; 1)\right)\right) \\
&\overset{d}{=} \text{MaxPool}\left(\text{ReLu}\left(\text{Conv}\left(W(1), x\right) + b(t; 1)\right)\right),
\end{aligned}
$$

where $W(1) = \Gamma^\top(1)$ corresponds to the weights at layer 1 of the CNN, and $\overset{d}{=}$ implies that discriminative relaxation is applied. In Eqn. 22, since $s(1, p_1) \in \{0, 1\}$, $\max_{s(1,p_1)} \left(s(1, p_1) \times .\right)$ is equivalent to $\max(\,.\,, 0)$ and, therefore, yields the ReLU function. Also, in order to compute $\max_{t(1,p_1)} ()$, we take the brute force approach, computing the function inside the parentheses for all possible values of $t(1, p_1)$ and pick the maximum one. This procedure is equivalent to the MaxPool operator in CNNs. Here we can make the bias term $b(t; 1)$ independent of $t$ and $b(t; 1, p_1)$ are the same for all pixels $p_1$ in the same feature map of $h(1)$ as in CNNs. Similarly, $\psi(2) = \text{MaxPool}\left(\text{ReLu}\left(\text{Conv}\left(W(2), \psi(1)\right) + b(2)\right)\right)$. Thus, we obtain the conclusion of the theorem for $L = 2$.

### .9 PROOF FOR THEOREM .11: DERIVING THE RESIDUAL NETWORKS FROM THE RESIDUAL NEURAL RENDERING MODEL

Similar to the proof in Section .8 above, when the rendering process at each layer in ResNRM is as in Eqn. 16, the activations $\psi(\ell)$ is given by:

$$
\begin{aligned}
\psi(\ell) &= \left(\max_{t(\ell,p), s(\ell,p)} s(\ell, p) \left(\left(\Gamma^\top(\ell, p) B^\top(\ell, p) T^\top(t; \ell, p) + \tilde{\Lambda}_{\text{skip}}^\top(\ell, p)\right) \psi(\ell - 1) + b(t; \ell, p)\right)\right)_{p \in \mathcal{P}(\ell)} \\
&= \left(\max\left(\left(\text{ReLu}\left(\Gamma^\top(\ell, p) B^\top(\ell, p) T^\top(t; \ell, p) \psi(\ell - 1) + b(t; \ell, p)\right.\right.\right.\right. \\
&\qquad\qquad\qquad\qquad \left.\left.\left.\left. + \tilde{\Lambda}_{\text{skip}}^\top(\ell, p) \psi(\ell - 1)\right)\right)_{t(\ell,p)=0,1,2,3}\right)\right)_{p \in \mathcal{P}(\ell)} \\
&= \text{MaxPool}\left(\text{ReLu}\left(\text{Conv}\left(\Gamma^\top(\ell, p), \psi(\ell - 1)\right) + b(t; \ell) + \tilde{\Lambda}_{\text{skip}}^\top(\ell) \psi(\ell - 1)\right)\right) \\
&\overset{d}{=} \text{MaxPool}\left(\text{ReLu}\left(\text{Conv}(W^{(\ell)}, \psi(\ell - 1)) + b(t; \ell) + \underbrace{W_{\text{skip}}(\ell) \psi(\ell - 1)}_{\text{skip connection}}\right)\right). \qquad (23)
\end{aligned}
$$

Here we can again make the bias term $b(t; \ell)$ independent of $t$ and $b(t; \ell, p)$ are the same for all pixels $p_\ell$ in the same feature map of $h(\ell)$ as in CNNs. We obtain the conclusion of the Theorem .11.

### .10 PROOF FOR THEOREM .13: DERIVING THE DENSELY CONNECTED NETWORKS FROM THE DENSE NEURAL RENDERING MODEL

Similar to the proof for Theorem .11 above, when the rendering process at each layer in ResNRM is as in Eqn. 20, the activations $\psi(\ell)$ is given by:

$$
\psi(\ell) \equiv \begin{bmatrix} \left( \max_{s(\ell,p),t(\ell,p)} \left( s(\ell,p) \left( \Gamma^\top(\ell,p) B^\top(\ell,p) T^\top(t;\ell,p) \psi(\ell-1) + b(t;\ell,p) \right) \right) \right)_{p \in \mathcal{P}(\ell)} \\ \psi(\ell-1) \end{bmatrix}
$$

$$
= \begin{bmatrix} \left( \max \left( \left( \mathrm{ReLu}\left( \Gamma^\top(\ell,p) B^\top(\ell,p) T^\top(t;\ell,p) \psi(\ell-1) + b(t;\ell,p) \right) \right)_{t(\ell,p)=0,1,2,3} \right) \right)_{p \in \mathcal{P}(\ell)} \\ \psi(\ell-1) \end{bmatrix}
$$

$$
= \begin{bmatrix} \mathrm{MaxPool}(\mathrm{ReLu}(\mathrm{Conv}(\Gamma^\top(\ell,p), \psi(\ell-1)) + b(t;\ell))) \\ \psi(\ell-1) \end{bmatrix}
$$

$$
\overset{d}{=} \begin{bmatrix} \mathrm{MaxPool}(\mathrm{ReLu}(\mathrm{Conv}(W^{(\ell)}, \psi(\ell-1)) + b(t;\ell))) \\ \psi(\ell-1) \end{bmatrix}. \tag{24}
$$

Again we can make the bias term $b(t;\ell)$ independent of $t$ and $b(t;\ell,p)$ are the same for all pixels $p_\ell$ in the same feature map of $h(\ell)$ as in CNNs. We obtain the conclusion of the Theorem .13.

### .11 PROVING THAT THE PARAMETRIZED JOINT PRIOR P(Y,Z) IS A CONJUGATE PRIOR

Again, for simplicity, we only consider the proof for two layers. The argument for $L \geq 3$ is similar and can be derived recursively from the proof for $L = 2$. In the derivation below, $h(2)$ is $\mu(y)$. As in Eqn.1 in the definition of the NRM, the joint prior of $y$ and $z$ is given by:

$$
p(y,z) \propto \exp\left( \frac{1}{\sigma^2} b^\top(t;2)(s(2) \odot h(2)) + \frac{1}{\sigma^2} b^\top(t;1)(s(1) \odot h(1)) + \ln \pi_y \right)
$$

$$
= \exp\left( \frac{1}{\sigma^2} h^\top(2)(b(t;2) \odot s(2)) + \frac{1}{\sigma^2} h^\top(1)(b(t;1) \odot s(1)) + \ln \pi_y \right)
$$

$$
= \exp\left( \frac{1}{\sigma^2} h^\top(2)(b(t;2) \odot s(2)) + \frac{1}{\sigma^2} h^\top(2)\Lambda^\top(2)(b(t;1) \odot s(1)) + \ln \pi_y \right)
$$

$$
= \exp\left( \frac{1}{\sigma^2} h^\top(2) \left[ \Lambda^\top(2)(b(t;1) \odot s(1)) + b(t;2) \odot s(2) \right] + \ln \pi_y \right) \tag{25}
$$

Furthermore, as explained in Section .14 of Appendix D, due to the constant norm assumption with $h(y,z;0)$, the likelihood $p(x|y,z)$ is estimated as follows:

$$
p(x|y,z) \propto \exp\left( \frac{1}{\sigma^2} h^\top(y,z;0) x \right)
$$

$$
= \exp\left( \frac{1}{\sigma^2} h^\top(2)\Lambda^\top(2)\Lambda^\top(1) x \right)
$$

The posterior $p(y,z|x)$ is given by:

$$
p(y,z|x) \propto \exp\left( \frac{1}{\sigma^2} h^\top(2) \left[ \Lambda^\top(2)(b(t;1) \odot s(1)) + \Lambda^\top(2)\Lambda^\top(1) x + b(t;2) \odot s(2) \right] + \ln \pi_y \right)
$$

$$
= \exp\left( \frac{1}{\sigma^2} h^\top(2) \left[ \Lambda^\top(2)(b(t;1) \odot s(1) + \Lambda^\top(1) x) + b(t;2) \odot s(2) \right] + \ln \pi_y \right) \tag{26}
$$

Comparing Eqn. 25 and Eqn. 26, we see that the prior and the posterior have the same functional form. This completes the proof.

### .12 DERIVING OTHER COMPONENTS IN THE CONVOLUTIONAL NEURAL NETWORKS

#### .12.1 DERIVING THE LEAKY RECTIFIED LINEAR UNIT (LEAKY RELU)

The Leaky ReLU can be derived from the NRM in the same way as we derive the ReLU in Section .8, but instead of $s(\ell) \in \{0,1\}$, we now let $s(\ell) \in \{\alpha, 1\}$, where $\alpha$ is a small positive constant. Then, in

Eqn. 22, $\max\limits_{s(\ell,p)}(s(\ell,p)\times.)$ is equivalent to $[.<0](\alpha\times.)+[.>=0](.)$, which is the LeakyReLU function. Note that compared to Eqn. 22, here we replace the layer number $(1)$ by $(\ell)$ since the same derivation can be applied at any layer.

### .13 Proofs for Batch Normalization

In this section we will derive the batch normalization from a 2-layer DRM. The same derivation can be generalized to the case of K-layer DRM.

In order to derive the Batch Normalization, we normalize the intermediate rendering image $h(\ell)$.

$$h(y,z;0)$$

$$= \sum_{p_1\in\mathcal{P}(1)} s(1,p_1)T(t;1,p_1)B(1,p_1)\Gamma(1,p_1)\frac{1}{\sigma_h(1,p_1)}\left(h(1,p_1)-\mathbb{E}_h(1,p_1)\right)-\mathbb{E}_{h(y,z;0)}$$

$$= \sum_{p_1\in\mathcal{P}(1)} s(1,p_1)T(t;1,p_1)B(1,p_1)\Gamma(1,p_1)\frac{1}{\sigma_h(1,p_1)}$$

$$\times\left(\left(\sum_{p_2\in\mathcal{P}(2)} s(1,p_2)T(t;2,p_2)B(2,p_2)\Gamma(2,p_2)\frac{1}{\sigma_h(y,z;0)}h(y,z;0)\right)(1,p_1)-\mathbb{E}_h(1,p_1)\right)$$

$$-\mathbb{E}_{h(y,z;0)} \tag{27}$$

During inference, we de-mean the input image and find $z^* = \arg\max\limits_z h^\top(y,z;0)(x-\mathbb{E}_{h(y,z;0)})$. In particular, the inference can be derived as follows:

$$\max_z h^\top(y,z;0)(x-\mathbb{E}_{h(y,z;0)})$$

$$=\max_z\left(\sum_{p_1\in\mathcal{P}(1)} s(1,p_1)T(t;1,p_1)B(1,p_1)\Gamma(1,p_1)\frac{1}{\sigma_h(1,p_1)}\left(h(1,p_1)-\mathbb{E}_h(1,p_1)\right)\right)^\top(x-\mathbb{E}_{h(y,z;0)})$$

$$-\underbrace{\mathbb{E}_{h(y,z;0)}^\top(x-\mathbb{E}_{h(y,z;0)})}_{\text{const w.r.t } y,z}$$

$$\overset{a}{\approx}\max_z\sum_{p_1\in\mathcal{P}(1)}\left(\underbrace{s(1,p_1)\left(h(1,p_1)-\mathbb{E}_h(1,p_1)\right)\frac{1}{\sigma_h(1,p_1)}}_{\text{scalar}}\underbrace{\underbrace{\Gamma^\top(1,p_1)B^\top(1,p_1)T^\top(t;1,p_1)}_{\text{row vector}}\underbrace{(x-\mathbb{E}_i)}_{\text{column vector}}}_{\text{scalar}}\right)$$

$$\underbrace{\phantom{\sum_{p_1\in\mathcal{P}(1)}}}_{\text{dot product}}$$

$$+\text{ const} := A + \text{const}.$$

Direct computation leads to

$$A=\max_z\sum_{p_1\in\mathcal{P}(1)}\left(s(1,p_1)\left(h(1,p_1)-\mathbb{E}_h(1,p_1)\right)\underbrace{\frac{\sigma_\psi(1,p_1)}{\sigma_h(1,p_1)}}_{\alpha(1,p_1)}\underbrace{\frac{1}{\sigma_\psi(1,p_1)}}_{\text{normalize}}\right.$$

$$\times\underbrace{\left(\Gamma^\top(1,p_1)B^\top(1,p_1)T^\top(t;1,p_1)x-\mathbb{E}[\Gamma^\top(1,p_1)B^\top(1,p_1)T^\top(t;1,p_1)x]\right)}_{\text{de-mean}}\Big)$$

$$=\max_{z(2)}\sum_{p_1\in\mathcal{P}(1)}\max_{s(1,p_1),t(1,p_1)}\left(s(1,p_1)\left(h(1,p_1)-\mathbb{E}_h(1,p_1)\right)\right.$$

$$\times\text{BatchNorm}(\Gamma^\top(1,p_1)B^\top(1,p_1)T^\top(t;1,p_1)x;\alpha(1,p_1))\Big)$$

$$=\max_{z(2)}\left(h(1)-\mathbb{E}_h(1)\right)^\top\left(\text{MaxPool}(\text{ReLu}(\text{BatchNorm}(\text{Conv}(\Gamma(1),x);\alpha(1),0)))\right) \tag{28}$$

In line (a), we approximate $\mathbb{E}_{h(y,z;0)}$ by its empirical value $\mathbb{E}_i$. The de-mean and normalize operators with the scale parameter $\alpha(1)$ and the shift parameter $\beta(1) = 0$ in the equations above already have the form of batch normalization. Note that when the model is trained with Stochastic Gradient Descent (SGD), the scale and shift parameters at each layer also account for the error in evaluating statistics of the activations using the mini-batches. Thus, $\beta(1)$ is not 0 any more, but a parameter learned during the training. Also, in the equations above, $\psi(1)$ is the activations at layer 1 in CNNs and given by:

$$\psi(1) = \text{MaxPool}(\text{ReLu}(\text{Normalize}(\text{Demean}(\text{Conv}(\Gamma(1), x))))), \tag{29}$$

where $\sigma_\psi(1)$ is the standard deviation of the $\psi(1)$. Eqn.28 can be expressed in term of $\psi(1)$ as follows:

$$\max_{z(2)} \left( h(1) - \mathbb{E}_h(1) \right)^\top \psi(1) + \text{ const}$$

$$= \max_{z(2)} h^\top(1)(\psi(1) - \mathbb{E}_\psi(1)) + (h^\top(1)\mathbb{E}_\psi(1) - \mathbb{E}_h^\top(1)\psi(1)) + \text{ const}$$

$$= \max_{z(2)} \sum_{p_2 \in \mathcal{P}(2)} \left( s(2, p_2)\mu(y, p_2) \underbrace{\frac{\sigma_\psi(2, p_2)}{\sigma_{\mu(y)}(p_2)}}_{\alpha(2, p_2)} \underbrace{\frac{1}{\sigma_\psi(2, p_2)}}_{\text{normalize}} \right.$$

$$\times \left( \underbrace{\Gamma^\top(2, p_2)B^\top(2, p_2)T^\top(t; 2, p_2)\psi(1) - \mathbb{E}[\Gamma^\top(2, p_2)B^\top(2, p_2)T^\top(t; 2, p_2)\psi(1)]}_{\text{de-mean}} \right.$$

$$\left. \left. + \underbrace{h(1, p_1)\mathbb{E}_\psi(1, p_1) - \mathbb{E}_h(1, p_1)\psi(1, p_1)}_{\beta(2, p_2)} \right) \right) + \text{ const}$$

$$= \max_{z(2)} \mu^\top(y) \left( \text{MaxPool}(\text{ReLu}(\text{BatchNorm}(\text{Conv}(\Gamma(2), \psi(1)); \alpha(2), \alpha(2) \odot \beta(2)))) \right) + \text{ const} \tag{30}$$

The batch normalization at this layer of the CNN has the scale parameter $\alpha(2)$ and the shift parameter is the element-wise product of $\alpha(2)$ and $\beta(2)$.

### .14 Proofs for connection between NRM and cross entropy

**PROOF OF THEOREM .1**  (a) To ease the presentation of proof, we denote the following key notation

$$A := \max_{(z_i)_{i=1}^n, \theta \in \mathcal{A}_\gamma} \frac{1}{n} \sum_{i=1}^n \ln p(y_i | x_i, z_i; \theta).$$

From the definition of $\mathcal{A}_\gamma = \{\theta : \|h(y, z; 0)\| = \gamma\}$, we achieve the following equations

$$A = \max_{\theta \in \mathcal{A}_\gamma} \frac{1}{n} \sum_{i=1}^n \max_{z_i} \log p(y_i | x_i, z_i; \theta)$$

$$= \max_{\theta \in \mathcal{A}_\gamma} \frac{1}{n} \sum_{i=1}^n \max_{z_i} \log \frac{p(x_i | y_i, z_i; \theta)p(y_i | z_i; \theta)}{\sum_{y=1}^K p(x_i | y, z_i; \theta)p(y | z_i; \theta)}$$

$$= \max_{\theta \in \mathcal{A}_\gamma} \frac{1}{n} \sum_{i=1}^n \max_{z_i} \left\{ \log p(x_i | y_i, z_i; \theta) + \log p(y_i, z_i | \theta) - \log \left( \sum_{y=1}^K p(x_i | y, z_i; \theta)p(y, z_i | \theta) \right) \right\}.$$

From the formulation of NRM, we have the following formulation of prior probabilities $p(y, z | \theta)$

$$p(y, z | \theta) = \frac{\exp\left( \dfrac{\eta(y, z)}{\sigma^2} \right) \pi_y}{\sum_{y', z'} \exp\left( \dfrac{\eta(y', z')}{\sigma^2} \right) \pi_{y'}}.$$

for all $(y, z)$. By means of the previous equations, we eventually obtain that

$$
\begin{aligned}
A \;=\; & \max_{\theta \in \mathcal{A}_\gamma} \frac{1}{n} \sum_{i=1}^{n} \max_{z_i} \Bigg\{ \log p(x_i|y_i, z_i; \theta) + \log \left( \frac{\exp\left(\dfrac{\eta(y_i, z_i)}{\sigma^2}\right) \pi_{y_i}}{\sum\limits_{y,z} \exp\left(\dfrac{\eta(y, z)}{\sigma^2}\right) \pi_y} \right) \\
& \qquad\qquad - \log \sum_{y=1}^{K} \left( p(x_i|y, z_i; \theta) \frac{\exp\left(\dfrac{\eta(y, z_i)}{\sigma^2}\right) \pi_y}{\sum\limits_{y',z'} \exp\left(\dfrac{\eta(y', z')}{\sigma^2}\right) \pi_{y'}} \right) \Bigg\} \\
\;=\; & \max_{\theta \in \mathcal{A}_\gamma} \frac{1}{n} \sum_{i=1}^{n} \max_{z_i} \left( \log p(x_i|y_i, z_i; \theta) + \log \left( \exp\left(\frac{\eta(y_i, z_i)}{\sigma^2}\right) \pi_{y_i} \right) \right) \\
& \qquad\qquad - \log \left( \sum_{y=1}^{K} \left( p(x_i|y, z_i; \theta) \exp\left(\frac{\eta(y, z_i)}{\sigma^2}\right) \pi_y \right) \right).
\end{aligned}
$$

By defining $\psi_i(y, z_i) := \log p(x_i|y, z_i; \theta) + \log \left( \exp\left(\dfrac{\eta(y, z_i)}{\sigma^2}\right) \pi_y \right)$ for all $1 \leq i \leq n$, the above equation can be rewritten as

$$
\begin{aligned}
A \;=\; & \max_{\theta \in \mathcal{A}_\gamma} \frac{1}{n} \sum_{i=1}^{n} \max_{z_i} \left( \log \left( \exp(\psi_i(y_i, z_i)) \right) - \log \left( \sum_{y=1}^{K} \exp(\psi_i(y, z_i)) \right) \right) \qquad (31) \\
\;=\; & \max_{\theta \in \mathcal{A}_\gamma} \frac{1}{n} \sum_{i=1}^{n} \max_{z_i} \log \left( \mathrm{Softmax}(\psi_i(y_i, z_i)) \right) \\
\;\geq\; & \max_{\theta \in \mathcal{A}_\gamma} \frac{1}{n} \sum_{i=1}^{n} \log \left( \mathrm{Softmax}\left( \max_{z_i} \psi_i(y_i, z_i) \right) \right) = B. \qquad (32)
\end{aligned}
$$

By means of direct computation, the following equations hold

$$
\begin{aligned}
B \;=\; & \max_{\theta \in \mathcal{A}_\gamma} \frac{1}{n} \sum_{i=1}^{n} \log \left( \mathrm{Softmax}\left( \max_{z_i} \log p(x_i|y_i, z_i; \theta) + \eta(y_i, z_i)/\sigma^2 + \log \pi_{y_i} \right) \right) \\
\;=\; & \max_{\theta \in \mathcal{A}_\gamma} \frac{1}{n} \sum_{i=1}^{n} \log \left( \mathrm{Softmax}\left( \max_{z_i} -\frac{\|x_i - \mu_{y_i, z_i}\|^2}{2\sigma^2} + \frac{\eta(y_i, z_i)}{\sigma^2} + \log \pi_{y_i} \right) \right) \\
\;\overset{(i)}{=}\; & \max_{\theta \in \mathcal{A}_\gamma} \frac{1}{n} \sum_{i=1}^{n} \log \left( \mathrm{Softmax}\left( \max_{z_i} \left( \frac{h^\top(y_i, z_i; 0)x_i + \eta(y_i, z_i)}{\sigma^2} \right) + b_{y_i} \right) \right) \\
\;=\; & -\min_{\theta \in \mathcal{A}_\gamma} \frac{1}{n} \sum_{i=1}^{n} -\log \left( \mathrm{Softmax}\left( \max_{z_i} \left( \frac{h^\top(y_i, z_i; 0)x_i + \eta(y_i, z_i)}{\sigma^2} \right) + b_{y_i} \right) \right) \\
\;=\; & -\min_{\theta \in \mathcal{A}_\gamma} \frac{1}{n} \sum_{i=1}^{n} -\log q(y_i|x_i) = -\min_{\theta \in \mathcal{A}_\Gamma} H_{p,q}(y|x)
\end{aligned}
$$

where equation (i) is due to the constant norm assumption with rendered images $h(y, z; 0)$. Therefore, we achieve the conclusion of part (a) of the theorem.

(b) Regarding the upper bound, from the definition of $\overline{z}_i$, we obtain that

$$\max_{z_i} \log\left(\text{Softmax}(\psi_i(y_i, z_i))\right) - \log\left(\text{Softmax}\left(\max_{z_i}\psi_i(y_i, z_i)\right)\right)$$

$$= \log\left(\text{Softmax}(\psi_i(y_i, \overline{z}_i))\right) - \log\left(\text{Softmax}\left(\max_{z_i}\psi_i(y_i, z_i)\right)\right)$$

$$\leq \log\frac{\max\limits_{z_i}\exp(\psi_i(y_i, z_i))}{\sum\limits_{y=1}^{K}\exp(\psi_i(y, \overline{z}_i))} - \log\left(\text{Softmax}\left(\max_{z_i}\psi_i(y_i, z_i)\right)\right)$$

$$= \log\left(\sum_{y=1}^{K}\exp(\max_{z_i}\psi_i(y, z_i))\right) - \log\left(\sum_{y=1}^{K}\exp(\psi_i(y, \overline{z}_i))\right)$$

$$\leq \log K + \max_{y}\max_{z_i}\psi_i(y, z_i) - \max_{y}\psi_i(y, \overline{z}_i)$$

$$\leq \log K + \max_{y}\left(\max_{z_i}\psi_i(y, z_i) - \psi_i(y, \overline{z}_i)\right)$$

for any $1 \leq i \leq n$. As a consequence, we obtain the conclusion of part (b) of the theorem.

**PROOF OF THEOREM .4**   (a) From the definitions of $(\overline{y}_i, \overline{z}_i)$ and $(\widetilde{y}_i, \widetilde{z}_i)$, we obtain that

$$U_n = \min_{\theta}\frac{1}{n}\sum_{i=1}^{n}\left(\frac{1}{2}\|x_i - h(\overline{y}_i, \overline{z}_i; 0)\|^2 - \log\pi_{\overline{y}_i, \overline{z}_i}\right)$$

$$= \min_{\theta}\left\{\frac{1}{n}\sum_{i=1}^{n}\left(\frac{1}{2}\|x_i - h(\overline{y}_i, \overline{z}_i; 0)\|^2 - \eta(\overline{y}_i, \overline{z}_i) - \log\pi_{\overline{y}_i}\right)\right.$$

$$\left. + \log\left(\sum_{(y', z')\in\mathcal{J}}\exp(\eta(y', z') + \log\pi_{y'})\right)\right\} = A$$

By means of direct computation, the following inequality holds

$$A \leq \min_{\theta}\left\{\frac{1}{n}\sum_{i=1}^{n}\left(\frac{1}{2}\|x_i - h(\widetilde{y}_i, \widetilde{z}_i; 0)\|^2 - \eta(\widetilde{y}_i, \widetilde{z}_i) - \log\pi_{\overline{y}_i}\right)\right.$$

$$\left. + \log\left(\sum_{(y', z')\in\mathcal{J}}\exp(\eta(y', z') + \log\pi_{y'})\right)\right\}.$$

It is clear that

$$\sum_{(y', z')\in\mathcal{J}}\exp(\eta(y', z') + \log\pi_{y'}) \leq |\mathcal{L}|\sum_{y'}\exp(\max_{z'\in\mathcal{L}}\eta(y', z') + \log\pi_{y'}).$$

Combining this inequality with the inequality of the term $A$ in the above display, we have

$$U_n \leq \min_{\theta}\left\{\frac{1}{n}\sum_{i=1}^{n}\left(\frac{1}{2}\|x_i - h(\widetilde{y}_i, \widetilde{z}_i; 0)\|^2 - \eta(\widetilde{y}_i, \widetilde{z}_i) - \log\pi_{\widetilde{y}_i}\right) + (\log\pi_{\widetilde{y}_i} - \log\pi_{\overline{y}_i})\right.$$

$$\left. + \log\left(\sum_{y'}\exp(\max_{z'\in\mathcal{L}}\eta(y', z') + \log\pi_{y'})\right) + \log|\mathcal{L}|\right\}$$

$$= \min_{\theta}\frac{1}{n}\sum_{i=1}^{n}\left\{\left(\frac{\|x_i - h(\widetilde{y}_i, \widetilde{z}_i; 0)\|^2}{2} - \log(\pi_{\widetilde{y}_i, \widetilde{z}_i})\right) + \left(\log\pi_{\widetilde{y}_i} - \log\pi_{\overline{y}_i}\right)\right\} + \log|\mathcal{L}|$$

$$\leq \min_{\theta}\frac{1}{n}\sum_{i=1}^{n}\left(\frac{\|x_i - h(\widetilde{y}_i, \widetilde{z}_i; 0)\|^2}{2} - \log(\pi_{\widetilde{y}_i, \widetilde{z}_i})\right) + \log\left(\frac{1}{\gamma} - 1\right) + \log|\mathcal{L}|$$

$$= V_n + \log\left(\frac{1}{\gamma} - 1\right) + \log|\mathcal{L}|$$

where the final inequality is due to the fact that $\pi_{\widetilde{y}_i}/\pi_{\overline{y}_i} \leq (1 - \overline{\gamma})/\overline{\gamma}$ for all $1 \leq i \leq n$. Therefore, we achieve the conclusion of part (a) of the theorem.

(b) Similar to the proof argument with part (a), we have

$$
\begin{aligned}
U_n &= \min_\theta \left\{ \frac{1}{n} \sum_{i=1}^n \left( \frac{1}{2} \|x_i - h(\overline{y}_i, \overline{z}_i; 0)\|^2 - \eta(\overline{y}_i, \overline{z}_i) - \log \pi_{\overline{y}_i} \right) \right. \\
&\qquad\qquad\qquad\qquad\qquad \left. + \log \left( \sum_{(y', z') \in \mathcal{J}} \exp(\eta(y', z') + \log \pi_{y'}) \right) \right\} \\
&\geq \min_\theta \left\{ \frac{1}{n} \sum_{i=1}^n \left( \frac{1}{2} \|x_i\|^2 - h(\widetilde{y}_i, \widetilde{z}_i; 0)^\top x_i - \eta(\widetilde{y}_i, \widetilde{z}_i) + \|h(\overline{y}_i, \overline{z}_i; 0)\|^2/2 - \log \pi_{\overline{y}_i} \right) \right. \\
&\qquad\qquad\qquad\qquad \left. + \log \left( \sum_{y'} \exp(\max_{z' \in \mathcal{L}} \eta(y', z') + \log \pi_{y'}) \right) \right) = B.
\end{aligned}
$$

Direct computation with $B$ leads to

$$
\begin{aligned}
B &= \min_\theta \frac{1}{n} \sum_{i=1}^n \left\{ \left( \frac{\|x_i - h(\widetilde{y}_i, \widetilde{z}_i; 0)\|^2}{2} - \log(\pi_{\widetilde{y}_i, \widetilde{z}_i}) \right) + \left( \log \pi_{\overline{y}_i} - \log \pi_{\widetilde{y}_i} \right) \right. \\
&\qquad\qquad\qquad\qquad\qquad\qquad\qquad \left. + \|h(\overline{y}_i, \overline{z}_i; 0)\|^2 - \|h(\widetilde{y}_i, \widetilde{z}_i; 0)\|^2 \right\} \\
&\geq \min_\theta \frac{1}{n} \sum_{i=1}^n \left( \frac{\|x_i - h(\widetilde{y}_i, \widetilde{z}_i; 0)\|^2}{2} - \log(\pi_{\widetilde{y}_i, \widetilde{z}_i}) \right) + \min_\theta \frac{1}{n} \sum_{i=1}^n \left( \log \pi_{\overline{y}_i} - \log \pi_{\widetilde{y}_i} \right) \\
&\qquad\qquad\qquad\qquad\qquad\qquad + \min_\theta \frac{1}{n} \sum_{i=1}^n \left( \|h(\overline{y}_i, \overline{z}_i; 0)\|^2 - \|h(\widetilde{y}_i, \widetilde{z}_i; 0)\|^2 \right) \\
&\geq V_n + \log \left( \frac{\overline{\gamma}}{1 - \overline{\gamma}} \right) + \min_\theta \frac{1}{n} \sum_{i=1}^n \left( \|h(\overline{y}_i, \overline{z}_i; 0)\|^2 - \|h(\widetilde{y}_i, \widetilde{z}_i; 0)\|^2 \right)
\end{aligned}
$$

where the final inequality is due to the fact that $\pi_{\widetilde{y}_i}/\pi_{\overline{y}_i} \geq \overline{\gamma}/(1 - \overline{\gamma})$ for all $1 \leq i \leq n$. As a consequence, we achieve the conclusion of part (b) of the theorem.

.15 PROOFS FOR STATISTICAL GUARANTEE AND GENERALIZATION BOUND FOR (SEMI)-SUPERVISED LEARNING

**PROOF OF THEOREM .6**   The proof of this theorem relies on several results with uniform laws of large numbers. In particular, we will need to demonstrate the following results

$$\sup_\theta \left| \frac{1}{n_1} \sum_{i=1}^{n_1} \max_{(y,z)\in\mathcal{J}} \left( h^\top(y,z;0)x_i + \tau(\theta_{y,z}) \right) \right.$$
$$\left. - \int \max_{(y,z)\in\mathcal{J}} \left( h^\top(y,z;0)x + \tau(\theta_{y,z}) \right) dP(x) \right| \to 0, \tag{33}$$

$$L_{n_1} = \sup_\theta \left| \frac{1}{n_1} \sum_{i=1}^{n_1} \sum_{(y',z')\in\mathcal{J}} 1_{\left\{ (y',z')=\arg\max_{(y,z)\in\mathcal{J}} \left( h^\top(y,z;0)x_i+\tau(\theta_{y,z}) \right) \right\}} \left( \frac{\|h(y',z';0)\|^2}{2} - \log \pi_{y'} \right) \right.$$
$$\left. - \int \left( \sum_{(y',z')\in\mathcal{J}} 1_{\left\{ (y',z')=\arg\max_{(y,z)\in\mathcal{J}} \left( h^\top(y,z;0)x+\tau(\theta_{y,z}) \right) \right\}} \left( \frac{\|h(y',z';0)\|^2}{2} - \log \pi_{y'} \right) \right) dP(x) \right| \to 0, \tag{34}$$

$$\sup_\theta \left| \frac{1}{n-n_1} \sum_{i=n_1+1}^{n} \max_{z\in\mathcal{L}} \left( h^\top(y_i,z;0)x_i + \tau(\theta_{y_i,z}) \right) \right.$$
$$\left. - \int \max_{z\in\mathcal{L}} \left( h^\top(y,z;0)x + \tau(\theta_{y,z}) \right) dQ(x,c) \right| \to 0, \tag{35}$$

$$E_n^{(1)} = \sup_\theta \left| \frac{1}{n-n_1} \sum_{i=n_1+1}^{n} \sum_{z'\in\mathcal{L}} 1_{\left\{ z'=\arg\max_{z\in\mathcal{L}} \left( h^\top(y_i,z;0)x_i+\tau(\theta_{y_i,z}) \right) \right\}} \left( \frac{\|h(y_i,z';0)\|^2}{2} - \log \pi_{y_i} \right) \right.$$
$$\left. - \int \left( \sum_{z'\in\mathcal{L}} 1_{\left\{ z'=\arg\max_{g\in\mathcal{J}} \left( h^\top(y,z;0)x+\tau(\theta_{y,z}) \right) \right\}} \left( \frac{\|h(y,z';0)\|^2}{2} - \log \pi_y \right) \right) dQ(x,c) \right| \to 0, \tag{36}$$

$$E_n^{(2)} = \sup_\theta \left| \frac{1}{n-n_1} \sum_{i=n_1+1}^{n} \log q_\theta(y_i|x_i) - \int \log q_\theta(y|x) dQ(x,c) \right| \to 0, \tag{37}$$

$$E_n^{(3)} = \sup_\theta \left| \frac{1}{n} \sum_{i=1}^{n} \sum_{y=1}^{K} q_\theta(y|x_i) \log \left( \frac{q_\theta(y|x_i)}{\pi_y} \right) \right.$$
$$\left. - \int \sum_{y=1}^{K} q_\theta(y|x) dQ(x,c) \log \left( \frac{q_\theta(y|x)}{\pi_y} \right) dP(x) \right| \to 0, \tag{38}$$

almost surely as $n \to \infty$. The proof for equation 35 is similar to that of equation 33; therefore, it is omitted.

**Proof of equation 33:**   It is clear that

$$\sup_\theta \left| \frac{1}{n_1} \sum_{i=1}^{n_1} \max_{(y,z)\in\mathcal{J}} \left( h^\top(y,z;0)x_i + \tau(\theta_{y,z}) \right) - \int \max_{(y,z)\in\mathcal{J}} \left( h^\top(y,z;0)x + \tau(\theta_{y,z}) \right) dP(x) \right|$$
$$\leq \sup_{|S'|\leq|\mathcal{J}|} \left| \frac{1}{n_1} \sum_{i=1}^{n_1} \max_{s\in S'} s^\top[x_i,1] - \int \max_{s\in S'} s^\top[x,1] dP(x) \right|$$

where $[x,1] \in \mathbb{R}^{D^{(0)}+1}$ denotes the vector forms by concatenating 1 to $x \in \mathbb{R}^{D^{(0)}}$ and $S'$ in the above supremum is the set of finite elements in $\mathbb{R}^{D^{(0)}+1}$. Therefore, to achieve the result of equation 33, it is sufficient to show that

$$\sup_{|S'|\leq|\mathcal{J}|} \left| \frac{1}{n_1} \sum_{i=1}^{n_1} \max_{s\in S'} s^\top[x_i,1] - \int \max_{s\in S'} s^\top[x,1] dP(x) \right| \to 0. \tag{39}$$

To obtain the conclusion of equation 39, we utilize the classical result with bracketing entropy to establish the uniform laws of large number (cf. Lemma 3.1 in van de Geer (2000)). In particular, we denote $\mathcal{G}$ to be the family of function on $\mathbb{R}^{D^{(0)}}$ with the form $f_{S'}(x) = \max\limits_{s \in S'} s^\top [x,1]$ where $S' \in \mathcal{O}_{|\mathcal{J}|}$, which contains all sets that have at most $|\mathcal{J}|$ elements in $\mathbb{R}^{D^{(0)}+1}$. Due to the assumption with distribution $P$, we can restrict $\mathcal{O}_{|\mathcal{J}|}$ to contain only set $S'$ with elements in $\mathbb{B}(R)$, which is a closed ball of radius $R$ on $\mathbb{R}^{D^{(0)}+1}$. By means of Lemma 2.5 in (van de Geer, 2000), we can find a finite set $E_\delta$ such that each element in $\mathbb{B}(R)$ is within distance $\delta$ to some element of $E_\delta$ for all $\delta > 0$. We denote $\mathcal{O}_{|\mathcal{J}|}(\delta)$ to be the subset of $\mathcal{O}_{|\mathcal{J}|}$ such that it only contains sets with elements in $E_\delta$ for all $\delta > 0$. Now, for each set $S' = \{s_1, \ldots, s_k\} \in \mathcal{O}_{|\mathcal{J}|}$, we can choose corresponding set $\overline{S} = \{s'_1, \ldots, s'_k\} \in \mathcal{O}_{|\mathcal{J}|}(\delta)$ such that $\|s_i - s'_i\| \leq \delta$ for all $1 \leq i \leq k$. Now, we denote

$$\overline{f}_{\overline{S}}(x) = \max_{s \in \overline{S}} s^\top [x,1] + \delta \|[x,1]\|,$$

$$\underline{f}_{\overline{S}}(x) = \max_{s \in \overline{S}} s^\top [x,1] - \delta \|[x,1]\|$$

for any $\overline{S} \in \mathcal{O}_{|\mathcal{J}|}(\delta)$. It is clear that $\underline{f}_{\overline{S}}(x) \leq f_S(x) \leq \overline{f}_{\overline{S}}(x)$ for all $x \in \mathbb{R}^{D^{(0)}}$. Furthermore, we also have that

$$\int (\overline{f}_{\overline{S}}(x) - \underline{f}_{\overline{S}}(x))dP(x) = 2\delta \int \|[x,1]\| dP(x) \leq 2\delta \left( \int \|x\| dP(x) + 1 \right).$$

For any $\epsilon > 0$, by choosing $\delta < \dfrac{\epsilon}{2(\int \|x\| dP(x) + 1)}$ then we will have that $\int (\overline{f}_{\overline{S}}(x) - \underline{f}_{\overline{S}}(x))dP(x) < \epsilon$. It implies that the $\epsilon$-bracketing entropy of $\mathcal{G}$ is finite for the $\mathbb{L}_1$ norm with distribution $P$ (for the definition of bracketing entropy, see Definition 2.2 in van de Geer (2000)). According to Lemma 3.1 in van de Geer (2000), it implies that $\mathcal{G}$ satisfies the uniform law of large numbers, i.e., equation 39 holds.

**Proof of equation 34:** To achieve the conclusion of this claim, we will need to rely on the control of Rademacher complexity based on Vapnik-Chervonenkis (VC) dimension. In particular, we firstly demonstrate that

$$\sup_{|S'|=k} \left| \frac{1}{n_1} \sum_{i=1}^{n_1} 1_{\left\{ j = \arg\max\limits_{1 \leq l \leq |S'|} [x_i,1]^\top s'_l \right\}} - \int 1_{\left\{ j = \arg\max\limits_{1 \leq l \leq |S'|} [x,1]^\top s'_l \right\}} dP(x) \right| \to 0 \qquad (40)$$

almost surely as $n_1 \to \infty$ for each $1 \leq j \leq k$ and $k \geq 1$ where the supremum is taken with respect to $S' = \{s'_1, \ldots, s'_k\}$. For each $j$, we denote the Rademacher complexity as follows

$$R_{n_1} = \mathbb{E} \sup_{|S'|=k} \left| \frac{1}{n_1} \sum_{i=1}^{n_1} \sigma_i 1_{\left\{ j = \arg\max\limits_{1 \leq l \leq |S'|} [x_i,1]^\top s'_l \right\}} \right|$$

where $\sigma_1, \ldots, \sigma_{n_1}$ are i.i.d. Rademacher random variables, i.e., $\mathbb{P}(\sigma_i = -1) = \mathbb{P}(\sigma_i = 1) = 1/2$ for $1 \leq i \leq n_1$. Then, for any $n_1 \geq 1$ and $\delta \geq 0$, according to standard argument with Rademacher complexity (Vershynin, 2011),

$$\sup_{|S'|=k} \left| \frac{1}{n_1} \sum_{i=1}^{n_1} 1_{\left\{ j = \arg\max\limits_{1 \leq l \leq |S'|} [x_i,1]^\top s'_l \right\}} - \int 1_{\left\{ j = \arg\max\limits_{1 \leq l \leq |S'|} [x,1]^\top s'_l \right\}} dP(x) \right| \leq 2R_{n_1} + \delta$$

with probability at least $1 - 2\exp\left( -\dfrac{n_1 \delta^2}{8} \right)$. According to Borel-Cantelli's lemma, to achieve equation 40, it is sufficient to demonstrate that $R_{n_1} \to 0$ as $n_1 \to \infty$.

To achieve that result, we will utilize the study of VC dimension with partitions (cf. Section 21.5 in Devroye et al. (1996)). In particular, for each set $S' = (s'_1, \ldots, s'_k)$, it gives rise to the partition $A_i = \left\{ x \in \mathbb{R}^{D^{(0)}} : [x,1]^\top s'_i \geq [x,1]^\top s'_l \ \forall l \in \{1, \ldots, k\} \right\}$ as $1 \leq i \leq k$. For

our purpose with equation 40, it is sufficient to consider $\mathcal{P}_{n_1} = \left\{ A_j, \bigcup_{i \neq j} A_i \right\}$, which is a partition of $\mathbb{B}(R)$, for each set $S'$ with $k$ elements. We denote $\mathcal{F}$ to be the collection of all $\mathcal{P}_n$ for all $S'$ with $k$ elements and $\mathcal{B}(\mathcal{P}_n)$ the collection of all sets obtained from the unions of elements of $\mathcal{P}_n$. For each data $(x_1, \ldots, x_{n_1})$, we let $N_\mathcal{F}(x_1, \ldots, x_{n_1})$ the number of different sets in $\{(x_1, \ldots, x_{n_1}) \cap A : A \in \mathcal{B}(\mathcal{P}_{n_1}) \text{ for } \mathcal{P}_{n_1} \in \mathcal{F}\}$. The shatter coefficient of $\mathcal{F}$ is defined as

$$\Delta_{n_1}(\mathcal{F}) = s(\mathcal{F}, n_1) = \max_{(x_1, \ldots, x_{n_1})} N_\mathcal{F}(x_1, \ldots, x_{n_1}).$$

According to Lemma 21.1 in Devroye et al. (1996), $\Delta_{n_1}(\mathcal{F}) \leq 4\Delta_{n_1}^*(\mathcal{F})$ where $\Delta_{n_1}^*(\mathcal{F})$ is the maximal number of different ways that $n_1$ points can be partitioned by members of $\mathcal{F}$. Now, for each element $\mathcal{P}_{n_1} = \left\{ A_j, \bigcup_{i \neq j} A_i \right\}$ of $\mathcal{F}$, it is clear that the boundaries between $A_j$ and $\bigcup_{i \neq j} A_i$ are subsets of hyperplanes. From the formulation of $A_j$, we have at most $k - 1$ boundaries between $A_j$ and $\bigcup_{i \neq j} A_i$. From the classical result of Dudley (1978), each $n_1$ points in $\mathbb{B}(R)$ can be splitted by a hyperplane in at most $n_1^{D^{(0)}+1}$ different ways as the VC dimension of the hyperplane is at most $D^{(0)} + 1$. As a consequence, we would have $\Delta_{n_1}^*(\mathcal{F}) \leq n_1^{(D^{(0)}+1)(k-1)}$, which leads to $\Delta_{n_1}(\mathcal{F}) \leq 4 n_1^{(D^{(0)}+1)(k-1)}$.

Going back to our evaluation with Rademacher complexity $R_{n_1}$, by means of Massart's lemma, we have that

$$
\begin{aligned}
R_{n_1} &= \mathbb{E}\left( \mathbb{E}_\sigma \sup_{|S'|=k} \left| \frac{1}{n_1} \sum_{i=1}^{n_1} \sigma_i 1_{\left\{ j = \arg\max_{1 \leq l \leq |S'|} [x_i, 1]^\top s_l' \right\}} \right| \Big| x_1, \ldots, x_{n_1} \right) \\
&\leq \mathbb{E}\left( \sqrt{\frac{2 \log 2 N_\mathcal{F}(x_1, \ldots, x_{n_1})}{n_1}} \right) \leq \sqrt{\frac{2(\log 8 + (D^{(0)}+1)(k-1) \log n_1)}{n_1}} \to 0 \quad (41)
\end{aligned}
$$

as $n_1 \to \infty$. Therefore, equation 40 is proved.

**Proof of equation 36:** To achieve the conclusion of this claim, we firstly demonstrate that

$$
\begin{aligned}
\sup_{|S'|=k} &\left| \frac{1}{n - n_1} \sum_{i=n_1+1}^{n} 1_{\left\{ j = \arg\max_{1 \leq l \leq |S'|} [x_i, 1]^\top s_l' \right\}} 1_{\{y_i=l\}} \right. \\
&\left. - \int 1_{\left\{ j = \arg\max_{1 \leq l \leq |S'|} [x, 1]^\top s_l' \right\}} 1_{\{c=l\}} dQ(x, c) \right| \to 0 \quad (42)
\end{aligned}
$$

almost surely as $n \to \infty$ for each $1 \leq j \leq k$ and $1 \leq l \leq K$ where $k \geq 1$ and the supremum is taken with respect to $S' = \{s_1', \ldots, s_k'\}$. The proof of the above result will rely on VC dimension with Voronoi partitions being established in equation 34. In particular, according to the standard argument with Rademacher complexity, it is sufficient to demonstrate that

$$R_n' = \mathbb{E} \sup_{|S'|=k} \left| \frac{1}{n - n_1} \sum_{i=n_1+1}^{n} \sigma_i 1_{\left\{ j = \arg\max_{1 \leq l \leq |S'|} [x_i, 1]^\top s_l' \right\}} 1_{\{y_i=l\}} \right| \to 0.$$

By means of the inequality with Rademacher complexity in equation 41, we obtain that

$$
\begin{aligned}
R'_n &= \sum_{v=0}^{n-n_1} \sum_{\vec{c} \in A_v} \mathbb{E}\left( \mathbb{E}_\sigma \sup_{|S'|=k} \left| \frac{1}{n-n_1} \sum_{i=n_1+1}^{n} \sigma_i 1_{\left\{ j = \underset{1 \le l \le |S'|}{\arg\max} [x_i,1]^\top s'_l \right\}} 1_{\{y_i=l\}} \right| \middle| \vec{c} \in A_v \right) \mathbb{P}(\vec{c} \in A_v) \\
&= \sum_{v=0}^{n-n_1} \mathbb{E} \sup_{|S'|=k} \left| \frac{1}{n-n_1} \sum_{i=n_1+1}^{n_1+v} \sigma_i 1_{\left\{ j = \underset{1 \le l \le |S'|}{\arg\max} [x_i,1]^\top s'_l \right\}} \right| p_l^v (1-p_l)^{n-n_1-v} \binom{n-n_1}{v} \\
&\le \sum_{v=1}^{n-n_1} \frac{v}{n-n_1} \sqrt{\frac{2(\log 8 + (D^{(0)}+1)(k-1)\log v)}{v}} p_l^v (1-p_l)^{n-n_1-v} \binom{n-n_1}{v} \\
&\le \sqrt{\frac{2\log 8}{n-n_1}} \sum_{v=1}^{n-n_1} \sqrt{\frac{v}{n-n_1}} p_l^v (1-p_l)^{n-n_1-v} \binom{n-n_1}{v} \\
&\quad + \sqrt{\frac{2(D^{(0)}+1)(k-1)\log(n-n_1)}{n-n_1}} \sum_{v=1}^{n-n_1} \sqrt{\frac{v \log v}{(n-n_1)\log(n-n_1)}} p_l^v (1-p_l)^{n-n_1-v} \binom{n-n_1}{v} \\
&\le \left( \sqrt{\frac{2\log 8}{n-n_1}} + \sqrt{\frac{2(D^{(0)}+1)(k-1)\log(n-n_1)}{n-n_1}} \right) \sum_{v=1}^{n-n_1} p_l^v (1-p_l)^{n-n_1-v} \binom{n-n_1}{v} \\
&= \left( \sqrt{\frac{2\log 8}{n-n_1}} + \sqrt{\frac{2(D^{(0)}+1)(k-1)\log(n-n_1)}{n-n_1}} \right) (1 - (1-p_l)^{n-n_1}) \to 0
\end{aligned}
$$

as $n \to \infty$ where $\vec{c} = (c_{n-n_1+1}, \dots, c_n)$ and $A_v$ is the set of $\vec{c}$ such that there are exactly $v$ values of $y_i$ to be $l$ for $0 \le v \le n-n_1$. The final inequality is due to the fact that $v/(n-1) \le 1$ and $v \log v / \{(n-1)\log(n-1)\} \le 1$ for all $1 \le v \le n-n_1$. Therefore, we achieve the conclusion of equation 42.

Now, coming back to equation 36, by means of triangle inequality, we achieve that

$$
\begin{aligned}
E_n^{(1)} &\le \sum_{l=1}^{K} \sup_\theta \left| \frac{1}{n-n_1} \sum_{i=n_1+1}^{n} \sum_{z' \in \mathcal{L}} 1_{\left\{ z' = \underset{z \in \mathcal{L}}{\arg\max} \left( h^\top(y_i,z;0)x_i + \tau(\theta_{y_i,z}) \right) \right\}} 1_{\{y_i=l\}} \left( \frac{\|h(y_i,z';0)\|^2}{2} - \log \pi_{y_i} \right) \right. \\
&\qquad \left. - \int \left( \sum_{z' \in \mathcal{L}} 1_{\left\{ z' = \underset{g \in \mathcal{J}}{\arg\max} \left( h^\top(y,z;0)x + \tau(\theta_{y,z}) \right) \right\}} 1_{\{y=l\}} \left( \frac{\|h(y,z';0)\|^2}{2} - \log \pi_y \right) \right) dQ(x,y) \right| \\
&\le \sum_{l=1}^{K} \sup_\theta \sum_{z' \in \mathcal{L}} \left| \frac{\|h(l,z';0)\|^2}{2} - \log \pi_l \right| \left| \frac{1}{n-n_1} \sum_{i=n_1+1}^{n} 1_{\left\{ z' = \underset{z \in \mathcal{L}}{\arg\max} \left( h^\top(y_i,z;0)x_i + \tau(\theta_{y_i,z}) \right) \right\}} 1_{\{y_i=l\}} \right. \\
&\qquad \left. - \int 1_{\left\{ z' = \underset{g \in \mathcal{J}}{\arg\max} \left( h^\top(y,z;0)x + \tau(\theta_{y,z}) \right) \right\}} 1_{\{y=l\}} dQ(x,y) \right| \to 0
\end{aligned}
$$

where the last inequality is due to the results with uniform laws of large numbers from equation 42 and the fact that $\dfrac{\|h(l,z';0)\|^2}{2} - \log \pi_l$ is bounded for all $l$ and $z' \in \mathcal{L}$. Hence, we obtain the conclusion of equation 36.

**Proof of equation 37:**   For this claim, we have the following inequality

$$
\begin{aligned}
E_n^{(2)} \;\leq\; & \sup_{\{S'_y\},\{\pi_y\}} \left| \frac{1}{n-n_1} \sum_{i=n_1+1}^{n} \log \frac{\exp\left( \max\limits_{s \in S'_{y_i}} s^\top [x_i,1] + \log \pi_{y_i} \right)}{\sum\limits_{\ell=1}^{K} \exp\left( \max\limits_{s \in S'_l} s^\top [x_i,1] + \log \pi_l \right)} \right. \\
& \left. - \int \log \frac{\exp\left( \max\limits_{s \in S'_y} s^\top [x,1] + \log \pi_y \right)}{\sum\limits_{\ell=1}^{K} \exp\left( \max\limits_{s \in S'_l} s^\top [x,1] + \log \pi_l \right)} dQ(x,y) \right| \\
\leq\; & \sum_{\ell=1}^{K} \sup_{\{S'_y\},\{\pi_y\}} \left| \frac{1}{n-n_1} \sum_{i=n_1+1}^{n} \log \frac{\exp\left( \max\limits_{s \in S'_{y_i}} s^\top [x_i,1] + \log \pi_{y_i} \right)}{\sum\limits_{\ell=1}^{K} \exp\left( \max\limits_{s \in S'_l} s^\top [x_i,1] + \log \pi_l \right)} 1_{\{y_i=l\}} \right. \\
& \left. - \int \log \frac{\exp\left( \max\limits_{s \in S'_y} s^\top [x,1] + \log \pi_y \right)}{\sum\limits_{\ell=1}^{K} \exp\left( \max\limits_{s \in S'_l} s^\top [x,1] + \log \pi_l \right)} 1_{\{c=l\}} dQ(x,y) \right| = \sum_{\ell=1}^{K} F_{n,l}
\end{aligned}
$$

where $\left\{ S'_y \right\}$ in the above supremum stands for the collection of sets $S'_1, \ldots, S'_K$ such that $|S'_y| \leq |\mathcal{L}|$ and elements in $S'_y$ are in $\mathbb{R}^{D^{(0)}+1}$. Therefore, to achieve the conclusion of equation 37, it is sufficient to demonstrate that $F_{n,l} \to 0$ almost surely as $n \to \infty$ for each $1 \leq l \leq K$.

In fact, for each $1 \leq l \leq K$, we denote $\mathcal{G}$ to be the family of function on $\mathbb{R}^{D^{(0)}} \times \{1, \ldots, K\}$ of the form

$$
f_{\{S'_y\},\{\pi_y\}}(x,y) = \log \frac{\exp\left( \max\limits_{s \in S'_y} s^\top [x,1] + \log \pi_y \right)}{\sum\limits_{\ell=1}^{K} \exp\left( \max\limits_{s \in S'_l} s^\top [x,1] + \log \pi_l \right)} 1_{\{y=l\}}
$$

for all $(x,y)$ where $S'_1, \ldots, S'_K \in \mathcal{O}_{|\mathcal{L}|}$, which contains all sets that have at most $|\mathcal{L}|$ elements in $\mathbb{R}^{D^{(0)}+1}$, and $\{\pi_y\}$ satisfy that $\sum\limits_{y=1}^{K} \pi_y = 1$ and $\pi_y \geq \overline{\gamma}$ for all $1 \leq y \leq K$. Due to the assumption with distribution $P$, we can restrict $\mathcal{O}_{|\mathcal{L}|}$ to contain only set $S'$ with elements in the ball $\mathbb{B}(R)$ of radius $R$ on $\mathbb{R}^{D^{(0)}+1}$. By means of Lemma 2.5 in (van de Geer, 2000), we can find a finite set $E_\delta$ such that each element in $\mathbb{B}(R)$ is within distance $\delta$ to some element of $E_\delta$ for all $\delta > 0$. Additionally, there exists a set $\Delta(\delta)$ such that for each $(\pi_1, \ldots, \pi_K)$, we can find a corresponding element $(\pi'_1, \ldots, \pi'_K) \in \Delta(\delta)$ such that $\|(\pi_1, \ldots, \pi_K) - (\pi'_1, \ldots, \pi'_K)\| \leq \delta$. We denote

$$
\mathcal{F}(\delta) = \left\{ \left\{ S'_y \right\}, \{\pi_y\} : \text{ elements of } S'_y \text{ in } E_\delta, \text{ and}(\pi_1, \ldots, \pi_K) \in \Delta(\delta) \right\}
$$

for all $\delta > 0$.

For each element $\left\{ S'_y \right\}_{y=1}^{K}, \{\pi_y\}_{y=1}^{K}$, we can choose the corresponding element $\left\{ \overline{S}'_y \right\}_{y=1}^{K}, \{\overline{\pi}_y\}_{y=1}^{K} \in \mathcal{F}(\delta)$ such that $S'_y = \left\{ s_{y1}, \ldots, s_{yk_y} \right\}$, $\overline{S}'_y = \left\{ s'_{y1}, \ldots, s'_{yk_y} \right\}$ satisfy $\|s_{yj} - s'_{yj}\| \leq \delta$ for all $1 \leq y \leq K$ and $1 \leq j \leq k_y$. Additionally,

$\|(\pi_1,\ldots,\pi_K) - (\overline{\pi}_1,\ldots,\overline{\pi}_K)\| \leq \delta$. With these notations, we define

$$\overline{f}_{\{\overline{S}'_y\},\{\overline{\pi}_y\}}(x,y) = \log\left(\frac{\exp(\max_{s\in\overline{S}'_y} s^\top[x,1] + \log\overline{\pi}_y)}{\sum_{\tau=1}^K \exp(\max_{s\in\overline{S}'_\tau} s^\top[x,1] + \log\overline{\pi}_\tau)}\right)1_{\{y=l\}} + 2\delta\|[x,1]\| + 2\delta/\overline{\gamma},$$

$$\underline{f}_{\{\overline{S}'_y\},\{\overline{\pi}_y\}}(x,y) = \log\left(\frac{\exp(\max_{s\in\overline{S}'_y} s^\top[x,1] + \log\overline{\pi}_y)}{\sum_{\tau=1}^K \exp(\max_{s\in\overline{S}'_\tau} s^\top[x,1] + \log\overline{\pi}_\tau)}\right)1_{\{y=l\}} - 2\delta\|[x,1]\| - 2\delta/\overline{\gamma}$$

for any $\left(\{\overline{S}'_y\},\{\overline{\pi}_y\}\right) \in \mathcal{F}(\delta)$. By means of Cauchy-Schwarz's inequality, we have

$$s_{yi}^\top[x,1]1_{\{y=l\}} - (s'_{yi})^\top[x,1]1_{\{y=l\}} \quad \leq \quad \|s_{ci} - s'_{yi}\|\|[x,1]\|1_{\{y=l\}}$$
$$\leq \quad \delta\|[x,1]\|$$

for all $x$ and $1 \leq i \leq k$. Additionally, the following also holds

$$s_{yi}^\top[x,1]1_{\{y=l\}} - (s'_{yi})^\top[x,1]1_{\{y=l\}} \geq -\delta\|[x,1]\|.$$

Furthermore, $|\log\pi_y - \log\overline{\pi}_y| \leq \log(1 + \delta/\overline{\gamma}) \leq \delta/\overline{\gamma}$. Hence, we obtain that

$$\frac{\exp(\max_{s\in S'_y} s^\top[x,1] + \log\pi_y)}{\sum_{\tau=1}^K \exp(\max_{s\in S'_\tau} s^\top[x,1] + \log\pi_\tau)} \quad \leq \quad \frac{\exp(\max_{s\in\overline{S}'_y} s^\top[x,1] + \log\overline{\pi}_y + \delta\|[x,1]\| + \delta/\overline{\gamma})}{\sum_{\tau=1}^K \exp(\max_{s\in\overline{S}'_\tau} s^\top[x,1] + \log\overline{\pi}_\tau - \delta\|[x,1]\| - \delta/\overline{\gamma})}$$

$$\leq \quad \frac{\exp(\max_{s\in\overline{S}'_y} s^\top[x,1] + \log\overline{\pi}_y)}{\sum_{\tau=1}^K \exp(\max_{s\in\overline{S}'_\tau} s^\top[x,1] + \log\overline{\pi}_\tau)}\exp(2\delta\|[x,1]\| + 2\delta/\overline{\gamma}).$$

Similarly, we also have

$$\frac{\exp(\max_{s\in S'_y} s^\top[x,1] + \log\pi_y)}{\sum_{\tau=1}^K \exp(\max_{s\in S'_\tau} s^\top[x,1] + \log\pi_\tau)} \geq \frac{\exp(\max_{s\in\overline{S}'_y} s^\top[x,1] + \log\overline{\pi}_y)}{\sum_{\tau=1}^K \exp(\max_{s\in\overline{S}'_\tau} s^\top[x,1] + \log\overline{\pi}_\tau)}\exp(-2\delta\|[x,1]\| - 2\delta/\overline{\gamma}).$$

As a consequence, we achieve that

$$\underline{f}_{\{\overline{S}'_y\},\{\overline{\pi}_y\}}(x,y) \leq f_{\{S'_y\},\{\pi_y\}}(x,y) \leq \overline{f}_{\{\overline{S}'_y\},\{\overline{\pi}_y\}}(x,y)$$

for all $(x,y)$. With the formulations of $\underline{f}_{\{\overline{S}'_y\},\{\overline{\pi}_y\}}(x,c)$ and $\overline{f}_{\{\overline{S}'_y\},\{\overline{\pi}_y\}}(x,y)$, we have

$$\int\left(\overline{f}_{\{\overline{S}'_y\},\{\overline{\pi}_y\}}(x,y) - \underline{f}_{\{\overline{S}'_y\},\{\overline{\pi}_y\}}(x,y)\right)dQ(x,y)$$

$$= 4\delta\int\|[x,1]\|dQ(x,y) + 4\delta/\overline{\gamma} \leq 4\delta\left(\int\|x\|dQ(x,y) + 1\right) + 4\delta/\overline{\gamma}.$$

For any $\epsilon > 0$, by choosing $\delta < \frac{\epsilon}{4\int\|x\|dQ(x,y) + 4 + 4/\overline{\gamma}}$ then we will have $\int(\overline{f}_{\{\overline{S}'_y\},\{\overline{\pi}_y\}}(x,y) - \underline{f}_{\{\overline{S}'_y\},\{\overline{\pi}_y\}}(x,y))dQ(x,y) < \epsilon$. It implies that the $\epsilon$-bracketing entropy of $\mathcal{G}$ is finite for the $L_1$ norm with distribution $Q$. Therefore, it implies that $\mathcal{G}$ satisfies the uniform law of large numbers, i.e., $F_{n,l} \to 0$ almost surely as $n \to \infty$ for all $1 \leq l \leq K$. As a consequence, the uniform law of large number result equation 37 holds.

Going back to the original problem, denote $\widetilde{\theta}^0 = \left(\{\widetilde{\mu}^0(y)\}_{y=1}^K, \{\widetilde{\Gamma}_0(\ell)\}_{\ell=1}^L, \{\widetilde{\pi}_y^0\}_{y=1}^K, \{\widetilde{b}_0(\ell)\}_{\ell=1}^L\right)$ the optimal solutions of population partially labeled LDCE (Note that, the existence of these optimal

solutions is guaranteed due to the compact assumptions with the parameter spaces $\Theta_\ell$, $\Omega$, and $\Xi_\ell$ for all $1 \le \ell \le L$). Then, according to the formulation of partially labeled LDCE, we will have that

$$
\begin{aligned}
Y_n \;\le\; & \min_\theta \frac{\alpha_{\mathrm{RC}}}{n} \Bigg\{ \sum_{i=1}^{n_1} \sum_{(y',z')\in\mathcal{J}} \mathbb{1}_{\left\{(y',z')=\arg\max_{(y,z)\in\mathcal{J}}\left( x_i^\top \widetilde{h}^0(y,z;0)+\tau(\widetilde{\theta}^0_{y,z})\right)\right\}} \left( \frac{\|x_i - \widetilde{h}^0(y',z';0)\|^2}{2} \right. \\
& \left. - \log(\widetilde{p}^0(y',z')) \right) + \sum_{i=n_1+1}^{n} \sum_{z'\in\mathcal{L}} \mathbb{1}_{\left\{z'=\arg\max_{z\in\mathcal{L}}\left( x_i^\top \widetilde{h}^0(y,z;0)+\tau(\widetilde{\theta}^0_{y,z})\right)\right\}} \left( \frac{\|x_i - \widetilde{h}^0(y_i,z';0)\|^2}{2} \right. \\
& \left. - \log(\widetilde{p}^0(y_i,z')) \right) \Bigg\} - \frac{\alpha_{\mathrm{CE}}}{n-n_1} \sum_{i=n_1+1}^{n} \log q_{\widetilde{\theta}^0}(y_i|x_i) + \frac{\alpha_{\mathrm{KL}}}{n} \sum_{i=1}^{n} \sum_{y=1}^{K} q_{\widetilde{\theta}^0}(y|x_i) \log\left( \frac{q_{\widetilde{\theta}^0}(y|x_i)}{\pi_y} \right) = D_n
\end{aligned}
$$

for all $n \ge 1$ where we have the following formulations

$$
\begin{aligned}
\widetilde{h}^0(y,z;0) &= \widetilde{\Lambda}_0(z;1)\dots\widetilde{\Lambda}_0(z;L)\widetilde{\mu}^0(y), \\
\widetilde{\Lambda}_0(z;\ell) &= \sum_{p\in\mathcal{P}(\ell)} s(\ell,p)T(t;\ell,p)B(\ell,p)\widetilde{\Gamma}_0(\ell,p), \\
\widetilde{p}^0(y',z') &= \exp\left( \tau(\widetilde{\theta}^0_{y',z'}) + \log\widetilde{\pi}^0_{y'} \right) \Big/ \left( \sum_{y=1}^{K} \exp\left( \max_{z\in\mathcal{L}} \tau(\widetilde{\theta}^0_{y,z}) + \log\widetilde{\pi}^0_y \right) \right).
\end{aligned}
$$

From the results with uniform laws of large numbers in equation 33, equation 34, equation 35, equation 36, equation 37, and equation 38, we obtain that

$$
\begin{aligned}
& \frac{1}{n_1} \sum_{i=1}^{n_1} \sum_{(y',z')\in\mathcal{J}} \mathbb{1}_{\left\{(y',z')=\arg\max_{(y,z)\in\mathcal{J}}\left( x_i^\top \widetilde{h}^0(y,z;0)+\tau(\widetilde{\theta}^0_{y,z})\right)\right\}} \left( \frac{\|x_i - \widetilde{h}^0(y',z';0)\|^2}{2} - \log(\widetilde{p}^0(y',z')) \right) \\
& \to \int \sum_{(y',z')\in\mathcal{J}} \mathbb{1}_{\left\{(y',z')=\arg\max_{(y,z)\in\mathcal{J}}\left( x^\top \widetilde{h}^0(y,z;0)+\tau(\widetilde{\theta}^0_{y,z})\right)\right\}} \left( \frac{\|x - \widetilde{h}^0(y',z';0)\|^2}{2} - \log(\widetilde{p}^0(y',z')) \right) dP(x), \\
& \frac{1}{n-n_1} \sum_{i=n_1+1}^{n} \sum_{z'\in\mathcal{L}} \mathbb{1}_{\left\{z'=\arg\max_{z\in\mathcal{L}}\left( x_i^\top \widetilde{h}^0(y,z;0)+\tau(\widetilde{\theta}^0_{y,z})\right)\right\}} \left( \frac{\|x_i - \widetilde{h}^0(y_i,z';0)\|^2}{2} - \log(\widetilde{p}^0(y_i,z')) \right) \\
& \to \int \sum_{z'\in\mathcal{L}} \mathbb{1}_{\left\{z'=\arg\max_{z\in\mathcal{L}}\left( x^\top \widetilde{h}^0(y,z;0)+\tau(\widetilde{\theta}^0_{y,z})\right)\right\}} \left( \frac{\|x - \widetilde{h}^0(y,z';0)\|^2}{2} - \log(\widetilde{p}^0(y,z')) \right) dQ(x,y), \\
& \frac{1}{n-n_1} \sum_{i=n_1+1}^{n} \log q_{\widetilde{\theta}^0}(y_i|x_i) \to \int \log q_{\widetilde{\theta}^0}(y|x) dQ(x,y), \\
& \frac{1}{n} \sum_{i=1}^{n} \sum_{y=1}^{K} q_{\widetilde{\theta}^0}(y|x_i) \log\left( \frac{q_{\widetilde{\theta}^0}(y|x_i)}{\pi_y} \right) \to \int \sum_{y=1}^{K} q_{\widetilde{\theta}^0}(y|x) \log\left( \frac{q_{\widetilde{\theta}^0}(y|x)}{\pi_y} \right) dP(x)
\end{aligned}
$$

almost surely as $n \to \infty$. Combining with the fact that $n_1/n \to \overline{\lambda}$, the above results lead to $D_n \to \overline{Y}$ almost surely as $n \to \infty$. Therefore, we have $\lim_{n\to\infty} Y_n \le \overline{Y}$ almost surely as $n \to \infty$. On the other hand, with the results of uniform laws of large numbers in equation 33, equation 34, equation 35,

equation 36, equation 37, and equation 38, we also have that

$$\frac{1}{n_1}\sum_{i=1}^{n_1}\sum_{(y',z')\in\mathcal{J}}1\left\{(y',z')=\arg\max_{(y,z)\in\mathcal{J}}\left(x_i^\top\widetilde{h}(y,z;0)+\tau(\widetilde{\theta}_{y,z})\right)\right\}\left(\frac{\|x_i-\widetilde{h}(y',z';0)\|^2}{2}-\log(\widetilde{p}(y',z'))\right)$$

$$\to\int\sum_{(y',z')\in\mathcal{J}}1\left\{(y',z')=\arg\max_{(y,z)\in\mathcal{J}}\left(x^\top\widetilde{h}(y,z;0)+\tau(\widetilde{\theta}_{y,z})\right)\right\}\left(\frac{\|x-\widetilde{h}(y',z';0)\|^2}{2}-\log(\widetilde{p}(y',z'))\right)dP(x),$$

$$\frac{1}{n-n_1}\sum_{i=n_1+1}^{n}\sum_{z'\in\mathcal{L}}1\left\{z'=\arg\max_{z\in\mathcal{L}}\left(x_i^\top\widetilde{h}(y,z;0)+\tau(\widetilde{\theta}_{y,z})\right)\right\}\left(\frac{\|x_i-\widetilde{h}(y_i,z';0)\|^2}{2}-\log(\widetilde{p}(y_i,z'))\right)$$

$$\to\int\sum_{z'\in\mathcal{L}}1\left\{z'=\arg\max_{z\in\mathcal{L}}\left(x^\top\widetilde{h}^0(y,z;0)+\tau(\widetilde{\theta}_{y,z})\right)\right\}\left(\frac{\|x-\widetilde{h}(y,z';0)\|^2}{2}-\log(\widetilde{p}(y,z'))\right)dQ(x,y),$$

$$\frac{1}{n-n_1}\sum_{i=n_1+1}^{n}\log q_{\widetilde{\theta}}(y_i|x_i)\to\int\log q_{\widetilde{\theta}}(y|x)dQ(x,y),$$

$$\frac{1}{n}\sum_{i=1}^{n}\sum_{y=1}^{K}q_{\widetilde{\theta}}(y|x_i)\log\left(\frac{q_{\widetilde{\theta}}(y|x_i)}{\pi_y}\right)\to\int\sum_{y=1}^{K}q_{\widetilde{\theta}}(y|x)\log\left(\frac{q_{\widetilde{\theta}}(y|x)}{\pi_y}\right)dP(x)$$

almost surely as $n\to\infty$ where $\widetilde{\theta}=\left(\{\widetilde{\mu}(y)\}_{y=1}^{K},\left\{\widetilde{\Gamma}(\ell)\right\}_{\ell=1}^{L},\{\widetilde{\pi}_y\}_{y=1}^{K},\left\{\widetilde{b}(\ell)\right\}_{\ell=1}^{L}\right)$ is the optimal solution of partially labeled LDCE and

$$\widetilde{h}(y,z;0)=\widetilde{\Lambda}(z;1)\ldots\widetilde{\Lambda}(z;L)\widetilde{\mu}(y),$$

$$\widetilde{\Lambda}(z;\ell)=\sum_{p\in\mathcal{P}(\ell)}s(\ell,p)T(t;\ell,p)B(\ell,p)\widetilde{\Gamma}(\ell,p),$$

$$\widetilde{p}(y',z')=\exp\left(\tau(\widetilde{\theta}_{y',z'})+\log\widetilde{\pi}_{y'}\right)/\left(\sum_{y=1}^{K}\exp\left(\max_{z\in\mathcal{L}}\tau(\widetilde{\theta}_{y',z'})+\log\widetilde{\pi}_y\right)\right).$$

Hence, we eventually achieve that

$$Y_n\to\alpha_{\mathrm{RC}}\left\{\overline{\lambda}\left(\int\sum_{(y',z')\in\mathcal{J}}1\left\{(y',z')=\arg\max_{(y,z)\in\mathcal{J}}\left(x^\top\widetilde{h}(y,z;0)+\tau(\widetilde{\theta}_{y,z})\right)\right\}\left(\frac{\|x-\widetilde{h}(y',z';0)\|^2}{2}\right.\right.\right.$$

$$\left.-\log(\widetilde{p}(y',z'))\right)dP(x)\right)+(1-\overline{\lambda})\left(\int\sum_{z'\in\mathcal{L}}1\left\{z'=\arg\max_{z\in\mathcal{L}}\left(x^\top\widetilde{h}^0(y,z;0)+\tau(\widetilde{\theta}_{y,z})\right)\right\}\left(\frac{\|x-\widetilde{h}(y,z';0)\|^2}{2}\right.\right.$$

$$\left.\left.\left.-\log(\widetilde{p}(y,z'))\right)dQ(x,y)\right)\right\}-\alpha_{\mathrm{CE}}\int\log q_{\widetilde{\theta}}(y|x)dQ(x,y)$$

$$+\alpha_{\mathrm{KL}}\int\sum_{y=1}^{K}q_{\widetilde{\theta}}(y|x)\log\left(\frac{q_{\widetilde{\theta}}(y|x)}{\pi_y}\right)dP(x)\geq\overline{Y}$$

almost surely as $n\to\infty$. As a consequence, we achieve $Y_n\to\overline{Y}$ almost surely as $n\to\infty$. We reach the conclusion of the theorem.

**PROOF OF THEOREM .7** The proof argument of this theorem is a direct application of the results with uniform laws of large numbers in the proof of Theorem .6. In fact, we define

$$\widetilde{\mathcal{F}}_0(\epsilon)=\left\{(S,\{\pi_y\},\{b(\ell)\}):S=\left\{h(y,z;0):(y,z)\in\mathcal{J}\right\},\right.$$

$$\text{and}\inf_{(\widetilde{S}_0,\{\widetilde{\pi}_y^0\},\{\widetilde{b}_0(\ell)\})\in\mathcal{G}(\widetilde{\mathcal{F}}_0)}\left\{H(S,\widetilde{S}_0)+\sum_{y=1}^{K}|\pi_y-\widetilde{\pi}_y^0|+\sum_{z\in\mathcal{L}}\|b(\ell)-\widetilde{b}_0(\ell)\|\right\}\geq\epsilon\right\}$$

for any $\epsilon > 0$. Since the parameter spaces of $\theta$ are compact sets, the set $\widetilde{\mathcal{F}}_0(\epsilon)$ is also a compact set for all $\epsilon > 0$. Denote

$$
\begin{aligned}
&g\left(S, \{\pi_y\}, \{b(\ell)\}\right) \\
=\ & \alpha_{\mathrm{RC}}\Bigg\{\overline{\lambda}\Bigg(\int \sum_{(y',z')\in\mathcal{J}} 1_{\left\{(y',z')=\left(h^\top(y,z;0)x+\tau(\theta_{y,z})\right)\right\}}\left(\frac{\|x-h(y',z';0)\|^2}{2}-\log(\pi_{y',z'})\right)dP(x)\Bigg) \\
& +(1-\overline{\lambda})\Bigg(\int \sum_{z'\in\mathcal{L}} 1_{\left\{z'=\arg\max_{z\in\mathcal{L}}\left(h^\top(y,z;0)x+\tau(\theta_{y,z})\right)\right\}}\left(\frac{\|x-h(y,z';0)\|^2}{2}-\log(\pi_{y,z'})\right)dQ(x,y)\Bigg\} \\
& -\alpha_{\mathrm{CE}}\int \log q_\theta(y|x)dQ(x,y)+\alpha_{\mathrm{KL}}\int\sum_{y=1}^K q_\theta(y|x)dQ(x,c)\log\left(\frac{q_\theta(y|x)}{\pi_y}\right)dP(x)
\end{aligned}
$$

for all $(S, \{\pi_y\}, \{b(\ell)\})$. From the definition of $\widetilde{\mathcal{F}}_0(\epsilon)$, we have that

$$
g\left(S, \{\pi_y\}, \{b(\ell)\}\right) > g\left(\widetilde{S}_0, \{\widetilde{\pi}_y^0\}, \{\widetilde{b}_0(\ell)\}\right)
$$

for all $(S, \{\pi_y\}, \{b(\ell)\}) \in \widetilde{\mathcal{F}}_0(\epsilon)$ and $\left(\widetilde{S}_0, \{\widetilde{\pi}_y^0\}, \{\widetilde{b}_0(\ell)\}\right) \in \mathcal{G}(\widetilde{\mathcal{F}}_0)$. As $\widetilde{\mathcal{F}}_0(\epsilon)$ is a compact set, we further have that

$$
\inf_{(S,\{\pi_y\},\{b(\ell)\})\in\widetilde{\mathcal{F}}_0(\epsilon)} g\left(S, \{\pi_y\}, \{b(\ell)\}\right) > g\left(\widetilde{S}_0, \{\widetilde{\pi}_y^0\}, \{\widetilde{b}_0(\ell)\}\right)
$$

for all $\left(\widetilde{S}_0, \{\widetilde{\pi}_y^0\}, \{\widetilde{b}_0(\ell)\}\right) \in \mathcal{G}(\widetilde{\mathcal{F}}_0)$ and $\epsilon > 0$.

Now, according to the uniform laws of large numbers established in the proof of Theorem .6, we have that $Y_n \rightarrow g\left(\widetilde{S}_n, \{\widetilde{\pi}_y\}, \{\widetilde{b}(\ell)\}\right)$ almost surely as $n \rightarrow \infty$. According to the result of Theorem .6, it implies that $g\left(\widetilde{S}_n, \{\widetilde{\pi}_y\}, \{\widetilde{b}(\ell)\}\right) \rightarrow g\left(\widetilde{S}_0, \{\widetilde{\pi}_y^0\}, \{\widetilde{b}_0(\ell)\}\right)$ almost surely for all $\left(\widetilde{S}_0, \{\widetilde{\pi}_y^0\}, \{\widetilde{b}_0(\ell)\}\right) \in \mathcal{G}(\widetilde{\mathcal{F}}_0)$. Therefore, for each $\epsilon > 0$ we can find sufficiently large $N$ such that we have

$$
\inf_{\left(\widetilde{S}_0, \{\widetilde{\pi}_y^0\}, \{\widetilde{b}_0(\ell)\}\right)\in\mathcal{G}(\widetilde{\mathcal{F}}_0)}\left\{H(\widetilde{S}_n, \widetilde{S}_0)+\sum_{y=1}^K|\widetilde{\pi}_y-\widetilde{\pi}_y^0|+\sum_{z\in\mathcal{L}}\|\widetilde{b}(\ell)-\widetilde{b}_0(\ell)\|\right\} < \epsilon
$$

almost surely for all $n \geq N$. As a consequence, we achieve the conclusion of the theorem.

**PROOF OF THEOREM .8** The proof of the theorem is an application of Theorem 11 for generalization bound with margin from Koltchinskii & Panchenko (2002) based on an evaluation of Rademacher complexity. In particular, we denote

$$
\begin{aligned}
\mathcal{J}_n =\ & \Bigg\{h_{\overline{\tau}_n}(x,y):\mathbb{R}^{D^{(0)}}\times\{1,\dots,K\}\rightarrow\mathbb{R}| \\
& h_{\overline{\tau}_n}(x,y)=\max_{z\in\mathcal{L}(\overline{\tau}_n)}\left\{h^\top(y,z;0)x+\tau(\theta_{y,z})\right\}+\log\pi_y \ \forall\,(x,y)\ \text{for some}\ |\mathcal{L}(\overline{\tau}_n)|\leq\overline{\tau}_n|\mathcal{L}|\Bigg\}.
\end{aligned}
$$

Now, we denote $\widetilde{\mathcal{J}}_n = \{h_{\overline{\tau}_n}(.,y):\ y \in \{1,\dots,K\},\ h_{\overline{\tau}_n} \in \mathcal{J}_n\}$. For any $\delta > 0$, using the same argument as that of the proof of Theorem 11 in (Koltchinskii & Panchenko, 2002), with probability at least $1 - \delta$, we have

$$
\begin{aligned}
R_0(f_{\overline{\tau}_n}) \leq\ & \inf_{\Gamma\in(0,1]}\Bigg\{R_{n,\Gamma}(f_{\overline{\tau}_n})+\frac{8K(2K-1)}{\Gamma}\Re_n(\widetilde{\mathcal{J}}_n) \\
& +\left(\frac{\log\log_2(2\Gamma^{-1})}{n}\right)^{1/2}+\sqrt{\frac{\log(2\delta^{-1})}{2n}}\Bigg\}
\end{aligned}
\tag{43}
$$

where $\Re_n(\widetilde{\mathscr{J}}_n)$ is Rademacher complexity of $\widetilde{\mathscr{J}}_n$, which in our case is defined as

$$\Re_n(\widetilde{\mathscr{J}}_n) = \mathbb{E} \sup_\theta \sup_{|\mathcal{L}(\overline{\tau}_n)| \leq \overline{\tau}_n |\mathcal{L}|} \left| \frac{1}{n} \sum_{i=1}^{n} \sigma_i \left( \max_{g \in \mathcal{L}(\overline{\tau}_n)} \left\{ h^\top(y, z; 0) x_i + \tau(\theta_{y,z}) \right\} + \log \pi_y \right) \right|$$

where $\sigma_1, \ldots, \sigma_n$ are i.i.d. Rademacher random variables. Since $\overline{\gamma}$ is the lower bound of $\pi_y$ for all $1 \leq y \leq K$, we obtain that

$$\begin{aligned}
\Re_n(\widetilde{\mathscr{J}}) &\leq \mathbb{E} \sup_\theta \sup_{|\mathcal{L}(\overline{\tau}_n)| \leq \overline{\tau}_n |\mathcal{L}|} \left| \frac{1}{n} \sum_{i=1}^{n} \sigma_i \left( \max_{g \in \mathcal{L}(\overline{\tau}_n)} \left\{ h^\top(y, z; 0) x_i + \tau(\theta_{y,z}) \right\} \right) \right| \\
&\quad + \mathbb{E} \sup_\theta \sup_{|\mathcal{L}(\overline{\tau}_n)| \leq \overline{\tau}_n |\mathcal{L}|} \left| \frac{1}{n} \sum_{i=1}^{n} \sigma_i \log \pi_y \right| \\
&\leq \mathbb{E} \sup_\theta \sup_{|\mathcal{L}(\overline{\tau}_n)| \leq \overline{\tau}_n |\mathcal{L}|} \left| \frac{1}{n} \sum_{i=1}^{n} \sigma_i \left( \max_{g \in \mathcal{L}(\overline{\tau}_n)} \left\{ h^\top(y, z; 0) x_i + \tau(\theta_{y,z}) \right\} \right) \right| + \frac{|\log \overline{\gamma}|}{\sqrt{n}}
\end{aligned}$$

Furthermore, we have the following inequalities

$$\mathbb{E} \sup_\theta \sup_{|\mathcal{L}(\overline{\tau}_n)| \leq \overline{\tau}_n |\mathcal{L}|} \left| \frac{1}{n} \sum_{i=1}^{n} \sigma_i \left( \max_{g \in \mathcal{L}(\overline{\tau}_n)} \left\{ h^\top(y, z; 0) x_i + \tau(\theta_{y,z}) \right\} \right) \right|$$

$$\leq \mathbb{E} \sup_{|S'| \leq \overline{c}_n |\mathcal{L}|} \left| \frac{1}{n} \sum_{i=1}^{n} \sigma_i \max_{s \in S'} s^\top [x_i, 1] \right|$$

$$\leq 2\overline{c}_n |\mathcal{L}| \mathbb{E} \sup_{s \in \mathbb{B}(R)} \left| \frac{1}{n} \sum_{i=1}^{n} \sigma_i s^\top [x_i, 1] \right|$$

$$\leq 2\overline{c}_n |\mathcal{L}| R \mathbb{E} \left\| \frac{1}{n} \sum_{i=1}^{n} \sigma_i [x_i, 1] \right\|$$

$$\leq \frac{2\overline{c}_n |\mathcal{L}| (R^2 + 1)}{\sqrt{n}}$$

where the final inequality is due to Cauchy-Schwartz's inequality. Combining the above results with equation 43, we achieve the conclusion of the theorem.

# Appendix D

In this appendix, we provide detail descriptions for several simulation studies in the main text.

## .16 ARCHITECTURE OF THE NETWORK USED IN OUR SEMI-SUPERVISED EXPERIMENTS

Table 6: The network architecture used in all of semi-supervised experiments on CIFA10, CIFAR100 and SVHN.

| NAME | DESCRIPTION |
|------|-------------|
| input | $32 \times 32$ RGB image |
| conv1a | 128 filters, $3 \times 3$, pad = 'same', LReLU ($\alpha = 0.1$) |
| conv1b | 128 filters, $3 \times 3$, pad = 'same', LReLU ($\alpha = 0.1$) |
| conv1c | 128 filters, $3 \times 3$, pad = 'same', LReLU ($\alpha = 0.1$) |
| pool1 | Maxpool $2 \times 2$ pixels |
| drop1 | Dropout, $p = 0.5$ |
| conv2a | 256 filters, $3 \times 3$, pad = 'same', LReLU ($\alpha = 0.1$) |
| conv2b | 256 filters, $3 \times 3$, pad = 'same', LReLU ($\alpha = 0.1$) |
| conv2c | 256 filters, $3 \times 3$, pad = 'same', LReLU ($\alpha = 0.1$) |
| pool2 | Maxpool $2 \times 2$ pixels |
| drop2 | Dropout, $p = 0.5$ |
| conv3a | 512 filters, $3 \times 3$, pad = 'valid', LReLU ($\alpha = 0.1$) |
| conv3b | 256 filters, $1 \times 1$, LReLU ($\alpha = 0.1$) |
| conv3c | 128 filters, $1 \times 1$, LReLU ($\alpha = 0.1$) |
| pool3 | Global average pool ($6 \times 6 \to 1 \times 1$ pixels) |
| dense | Fully connected $128 \to 10$ |
| output | Softmax |

## .17 TRAINING DETAILS

### .17.1 SEMI-SUPERVISED LEARNING EXPERIMENTS ON CIFAR10, CIFAR100, AND SVHN

The training losses are discussed in Section 2.3. In addition to the cross-entropy loss, the reconstruction loss, and the RPN regularization, in order to further improve the performance of NRM, we introduce two new losses for training the model. Those two new losses are from our derivation of batch normalization using the NRM framework and from applying variational inference on the NRM. More details on these new training losses can be found in Appendix A.4. We compare NRM with state-of-the-art methods on semi-supervised object classification tasks which use consistency regularization, such as the $\Pi$ model (Laine & Aila, 2017), the Temporal Ensembling (Laine & Aila, 2017), the Mean Teacher (Tarvainen & Valpola, 2017), the Virtual Adversarial Training (VAT), and the Ladder Network (Rasmus et al., 2015). We also compare NRM with methods that do not use consistency regularization including the improved GAN (Salimans et al., 2016) and the Adversarially Learned Inference (ALI) (Dumoulin et al., 2017).

All networks were trained using Adam with learning rate of 0.001 for the first 20 epochs. Adam momentum parameters were set to beta1 = 0.9 and beta2 = 0.999. Then we used SGD with decayed learning rate to train the networks for another 380 epochs. The starting learning rate for SGD is 0.15 and the end learning rate at epoch 400 is 0.0001. We use batch size 128. Let the weights for the cross-entropy loss, the reconsruction loss, the KL divergence loss, the moment matching loss, and the RPN regularization be $\alpha_{CE}$, $\alpha_{RC}$, $\alpha_{KL}$, $\alpha_{MM}$, and $\alpha_{PN}$, respectively. In our training, $\alpha_{CE} = 1.0$, $\alpha_{RC} = 0.5$, $\alpha_{KL} = 0.5$, $\alpha_{MM} = 0.5$, and $\alpha_{PN} = 1.0$. For Max-Min cross-entropy, we use $\alpha^{max} = \alpha^{min} = 0.5$.

### .17.2 SUPERVISED LEARNING EXPERIMENTS WITH MAX-MIN CROSS ENTROPY

**Training on CIFAR10** We use the 26 2 x 96d "Shake-Shake-Image" ResNet in Gastaldi (2017) with the Cutout data augmentation suggested in DeVries & Taylor (2017) as our baseline. We implement the Max-Min cross-entropy on top of this baseline and turn it into a Max-Min network. In addition to Cutout data augmentation, standard translation and flipping data augmentation is applied on the 32 x 32 x 3 input image. Training procedure are the same as in (Gastaldi, 2017). In particular, the models were trained for 1800 epochs. The learning rate is initialized at 0.2 and is annealed using a Cosine

function without restart (see (Loshchilov & Hutter, 2016)). We train our models on 1 GPU with a mini-batch size of 128.

**Training on CIFAR10** We use the Squeeze-and-Excitation ResNeXt-50 as in Hu et al. (2018) as our baseline. As with CIFAR10, we implement the Max-Min cross-entropy for the baseline and turn it into a Max-Min network. During training, we follow standard practice and perform data augmentation with random-size cropping (Szegedy et al., 2015) to 224 x 224 x 3 pixels. We train the network with the Nesterov accelerated SGD for 125 epochs. The intial learning rate is 0.1 with momentum 0.9. We divide the learning rate by 10 at epoch 30, 60, 90, 95, 110, 120. Our network is trained on 8 GPUs with batch size of 32.

### .17.3 Semi-Supervised Training on MNIST with 50K Labeled to Get the Trained Model for Generating Reconstructed Image in Figure 2

The architecture of the baseline CNN we use is given in the Table 7. We use the same training procedure as in Section .17.1

Table 7: The network architecture used in our MNIST semi-supervised training.

| Name | Description |
|---|---|
| input | $28 \times 28$ image |
| conv1 | 32 filters, $5 \times 5$, pad = 'full', ReLU |
| pool1 | Maxpool $2 \times 2$ pixels |
| conv2a | 64 filters, $3 \times 3$, pad = 'valid', ReLU |
| conv2b | 64 filters, $3 \times 3$, pad = 'full', ReLU |
| pool2 | Maxpool $2 \times 2$ pixels |
| conv3 | 128 filters, $3 \times 3$, pad = 'valid', ReLU |
| pool3 | Global average pool ($6 \times 6 \rightarrow 1 \times 1$ pixels) |
| dense | Fully connected $128 \rightarrow 10$ |
| output | Softmax |

