# OpenReview forum: "Neural Rendering Model: Joint Generation and Prediction for Semi-Supervised Learning"
_ICLR.cc/2019/Conference_

### Official Review · AnonReviewer1 · 2018-11-02
**intersting model and claims, but incomprehensible paper**

**Rating:** 3
**Confidence:** 4

**Review:**

The paper claims to propose a novel generative probabilistic neural network model such that its encoder (classifying an image) can be approximated by a convolutional neural network with ReLU activations and MaxPooling layers. Besides the standard parameters of the units (weights and biases), the model has two additional latent variables per unit, which decide whether and where to put the template (represented by the weights of the neuron) in the subsequent layer, when generating an image from the class. Furthermore, the authors claim to derive new learning criteria for semi-supervised learning of the model including a novel regulariser and claim to prove its consistency.

Unfortunately, the paper is written in a way that is completely incomprehensible (for me). The accumulating ambiguities, together with its sheer length (44 pages with all supplementary appendices!), make it impossible for me to verify the model and the proofs of the claimed theorems. This begins already with definition of the model. The authors consider the latent variables as dependent and model them by a joint distribution. Its definition remains obscure, let alone the question how to marginalise over these variables when making inference. Without a thorough understanding of the model definition, it becomes impossible (for me) to follow the presentation of the learning approaches and the proofs for the theorem claims.

In my view, the presented material exceeds the limits of a single conference paper. A clear and concise definition of the proposed model accompanied by a concise derivation of the basic inference and learning algorithm would already make a highly interesting paper.

Considering the present state of the paper, I can't, unfortunately, recommend to accept it for ICLR.

---

> ### Author Response · Authors · 2018-11-22
> **An example of a 2-layer Neural Rendering Model (NRM) to clarify the model's definition**
>
> We would like to thank the reviewer for his/her comments. In what follows we shall give an example of a 2-layer Neural Rendering Model (NRM) to clarify the definition of our model. Also, in order to simplify the notations, we will define the generation process in the vectorized form. A L-layer NRM can be generalized from this example.
>
> The 2-layer NRM is a generative model, which generates images from the class templates $\mu$ via a linear transformations $\Lambda$. Here $\mu$ is a vector depending on the class label y and $\Lambda$ is a matrix depending on a set of latent variables z. Let further assume that $\mu$ of size 2 x 1, and $\Lambda$ is of size D x 2. Here, we assume the generated image X is of size D x 1. Images are generated from this 2-layer NRM as follows:
>
> 1) First, we sample the class label y from a categorical distribution $Cat(\pi_y)$. Given the value of y, we will select the corresponding template $\mu$ from a set of predefined templates, which will be learned during the training of the model by stochastic gradient descent (SGD).
>
> 2) Second, given y, we sample the latent variables z from the prior p(z|y), which is also a categorical distribution $Cat(\pi(z|y))$. Note that z contains s and t, which are the template selecting variable and the local translation variable defined in our paper. s and t are vectors of size 2 x 1. Element in s and t correspond to pixels in $\mu$. Element s(1) and s(2) in s take one of the two possible values - 0 and 1 - which selects render or not render. Element t(1) and t(2) in t take one of the four possible values - UPPER LEFT, UPPER RIGHT, LOWER LEFT, and LOWER RIGHT. If s(1) is 0, then the first column of $\Lambda$, i.e. $\Lambda(:,1)$, is a vector of 0’s. If s(1) is 1, then $\Lambda(:,1)$ takes one of the four possible predefined values depending on the value of t(1). These four vectors are locally translated versions of each other. The same process is applied for s(2), t(2), and $\Lambda(:,2)$. The generated image X is then given by:
>
> 	X = $\mu(1)$ x $\Lambda(:,1)$ + $\mu(2)$ x $\Lambda(:,2)$ + pixel noise
>
> Note that similar to the class template $\mu$, $\Lambda$ will be learned during training by SGD.
>
> The process above is captured by equation (1), (2), and (3) and illustrated by Figure 3 in our paper.

---

### Official Review · AnonReviewer2 · 2018-11-02
**The paper is not self-contained. Otherwise it provides an interesting probabilistic interpretation of CNNs which allows to design semi-supervised learning algorithm leading to promising empirical results.**

**Rating:** 5
**Confidence:** 3

**Review:**

pros:
- Interesting probabilistic interpretation of the CNNs improving work of [Patel 2016].
- State-of-the-art results following from the proposed probabilistic model.

cons:
- The regular 10 pages of the paper are not self-contained.

The paper is written in overly condensed way. I found it impossible to clearly understand major claims of the paper without reading the accompanied 34 pages long appendix. Many concepts/notations used in the paper are introduced in the appendix. My assessment is done solely based on reading the 10 regular pages.

- The probabilistic model NMR (equ (1) and (2)) defines distribution of inputs given latent variables and the outputs, $p(x|z,y)$, as well as it defines a distribution $p(z|y)$. Hence, in principle, one could maximize $p(x)=\sum_{i} E_{z} p(x_i|z,y)p(z|y)p(y)$ when learning from unsupervised data. Instead, the authors propose to learn by MINIMIZING the expectation (not clear w.r.t which variables) of $\log p(x,z|y)$ (equ (7)). Although it leads to empirically nice results, I do not see a clear motivation for such objective function.

- The motivation for using the MIN-MAX entropy as a loss function (sec 3) is also not clear. Why it should be better than the standard cross-entropy in the statistical sense?

- The proposed probabilistic model NMR differs form the previous work of [Patel 2016] by introducing the prior (1) on the latent variable. Unfortunately, pros and cons of this modifications are not fully discussed. E.g. how using dependent latent variables impact complexity of the inference.

---

> ### Author Response · Authors · 2018-11-22
> **Justify why we discuss many ideas, including the Max-Min network and the statistical guarantees, in our paper**
>
> We would like to thank the reviewer for his/her comments. We agree with the reviewer that we include many ideas in our paper. The Neural Rendering Model (NRM) without the Max-Min network can be itself a separate paper. However, given the NRM, it would be great to demonstrate how the model can be employed to improve the CNNs. In the paper, we show how to use NRM to design losses for training CNNs with both unlabeled and labeled data. We would also like to show how to utilize NRM to modify the architecture of CNNs. This is why we try to incorporate the Max-Min network into our paper. Furthermore, given our probabilistic setting, we believe that it is important to provide statistical guarantees for the model to establish that NRM is well defined statistically.

---

> ### Author Response · Authors · 2018-11-22
> **Summarize the major claims of our paper**
>
> We shall try to summarize the major claims of our paper below.
>
> Contribution 1: We develop a new generative model, the Neural Rendering Model (NRM), whose inference matches the architecture of a CNN. Different from the Deep Rendering Model (DRM) work of [Patel 2016], latent variables in our NRM are dependent.
>
> Contribution 2: We develop losses for training the CNNs with labeled and unlabeled data from maximizing the conditional log-likelihood and the expected complete-data log-likelihood of the NRM. Deriving losses for training the CNNs with labeled data is missing in the DRM work of [Patel 2016]. Given these losses, we provide consistency guarantees and generalization bounds for supervised and semi-supervised learning task in the NRM. Using the NRM, we also develop a new CNN architecture, which we term the Max-Min network. We show that the Max-Min network outperforms CNNs on supervised and semi-supervised learning tasks on popular benchmarks.
>
> Contribution 3: We show that NRM + Max-Min network achieves state-of-the-art empirical results for semi-supervised and supervised learning on benchmarks including CIFAR10, CIFAR100, and SVHN.

---

> ### Author Response · Authors · 2018-11-22
> **Explain why we condition the expected complete-data likelihood on the class label $y$**
>
> The reviewer is right that one way to do unsupervised learning is to maximize the expected complete data likelihood $\sum_{i} E_{z, y}  p(x_i, z, y|\theta)$ in which $\theta$ is the parameters of the model. This is indeed what we do to learn from unlabeled data using the NRM. However, in order to simplify the notations and formulas in the main text of our paper, we condition our model on $y$ so that we can ignore the term $ln p(y)$ in our equations. In order to convert this conditional model $p(x, z|y; \theta)$ into a marginal model $p(x, z, y| \theta)$, we only need to add the term $ln p(y)$ back into the equations in our paper. We will amend the text to reflect this.

---

> ### Author Response · Authors · 2018-11-22
> **Motivation for using the MIN-MAX entropy**
>
> We agree with the reviewer that we didn’t explain the Max-Min cross-entropy loss very clearly in the main text of our paper due to the page constraint. Let us try to explain the motivation for our Max-Min cross-entropy loss here. In the Appendix C.14 of our paper, we give a proof for Theorem 2.3(a) in the main text, which establishes the connection between the cross-entropy loss for training with labeled data and the conditional log-likelihood of the NRM. The equation (31) and (32) in that appendix (page 30) show how we derive the cross-entropy loss from the conditional log-likelihood of the NRM. Notice that, from equation (31) to equation (32), we lower bound the conditional log-likelihood of the NRM by maximizing over $z_{i}$ of the term $log(exp(\psi_{i}(y_{i}, z_{i})))$ without considering the log sum term. This corresponds to maximizing the likelihood of the correct labels without considering the likelihood of the incorrect labels. Notice that without normalization, maximizing the likelihood of the correct labels might also increase the likelihood of the incorrect labels. Alternatively, we can lower bound the conditional log-likelihood of the NRM by minimizing over $z_{i}$ of the log sum term. This is equivalent to minimizing the likelihood of the incorrect labels and yields the MIN cross-entropy loss. Combining both cross-entropy losses from these two lower bounds yields the MAX-MIN cross-entropy loss, which tries to maximize the likelihood of the correct labels and minimize the likelihood of the incorrect labels at the same time.

---

> ### Author Response · Authors · 2018-11-22
> **Compare the NRM with the Deep Rendering Model of [Patel 2016] and show the advantage of our parametrized prior on the latent variables**
>
> Let us take this opportunity to clarify our contributions in the Neural Rendering Model (NRM) in comparison with the Deep Rendering Model (DRM) work of [Patel 2016]. In our response below, we will also address your question regarding the advantage of our parametrized prior on the latent variables in the NRM.
>
> 1) We introduce the dependency between latent variables in our NRM. This dependency is implicitly enforced by the joint prior of all latent variables $p(z|y)$ in the NRM (see in equation (1) in our paper). Particularly, we parametrize this joint prior such that it cannot be factorized into the product of priors of individual latent variables. As a result, the latent variables are dependent. Such dependency is missing in the DRM, and this limits the DRM’s performance on semi-supervised learning tasks. As shown in our paper, the NRM, which captures the dependency between latent variables, significantly outperforms the DRM in semi-supervised learning tasks on popular benchmarks. Furthermore, the parametric form we choose for the joint prior $p(z|y)$ yields a conjugate prior for the NRM. Thus, inference and learning in the NRM is still computationally efficient. In particular, as shown in Theorem 2.2 in our paper, during inference, due to its conjugate form, the joint prior $p(z|y)$ only adds an additional bias term b(l) into each convolutional layer l in the CNNs. During learning, compared to the DRM, the NRM only needs to learn that extra bias term b(l) at each layer.
>
> 2) We derive the cross-entropy loss used for training CNNs with labeled data in conjunction with the architecture of the CNNs, all from maximizing the conditional log-likelihood of the NRM as shown in Theorem 2.3(a). This derivation is missing in the DRM of [Patel 2016]. In their paper, they only derive the reconstruction loss for unsupervised learning without addressing supervised learning, which CNNs are good at. From our derivation of the cross-entropy loss, we are able to provide statistical guarantees and generalization bounds of NRM and CNNs for supervised and semi-supervised learning tasks. These statistical guarantees and generalization bounds are missing in [Patel 2016] due to the fact that they cannot explain supervised learning in CNNs from their DRM.

---

### Official Review · AnonReviewer4 · 2018-11-13
**Interesting direction for probabilistic inference with CNNs and good semi-supervised learning results, but writing is difficult to follow**

**Rating:** 5
**Confidence:** 3

**Review:**

Summary: This paper introduces the Neural Rendering Model (NRM), a generative model in which the computations involved in inference correspond to those of a CNN forward pass. The NRM’s supervised learning objective is lower bounded by a variant of the cross-entropy objective. This objective is used to formulate a max-min network, which has a particular type of weight sharing between a standard branch with max pooling / ReLUs and a second branch with min pooling / NReLUs. The max-min objective and network show strong performance on semi-supervised learning tasks.

Posing a CNN as inference in a generative model is an interesting direction, and could be very useful for probabilistic inference in the context of neural nets. However, the paper is rather difficult to follow and requires frequent reference to the appendix to understand the main body. Some important components (like those relating to rendering paths and RPNs) are given good intuitive explanations early on but remain a bit ambiguous throughout the paper. I would recommend improving the presentation before publication.

Question: “we can modify NRM to incorporate our knowledge of the tasks and datasets into the model and perform JMAP inference to achieve a new CNN architecture.“
I appreciate the CNN / NRM correspondence in Table 1, and see how the NRM may be modified to produce modified CNN architectures. That being said, I am not sure I understand what sorts of task-specific knowledge are being referred to here. Could you give an example of a type of knowledge that the NRM would allow you to bake into a CNN architecture, but would otherwise be difficult to incorporate?

Minor:
“As been shown later in Section 2.2…”

“…is part of the optimization in equation equation 6.”

---

> ### Author Response · Authors · 2018-11-22
> **Explanation for the rendering paths and the RPN terms**
>
> We would like to thank the reviewer for pointing out that the rendering paths and the RPN are not clearly explained throughout our paper. In what follows we shall try to elucidate these two terms.
>
> A rendering path is a set of latent variables (s_{l}, t_{l}, y), l = 1,2,...,L, in the NRM, where s_{l} decides to render or not at particular locations in layer l, t_{l} decides how to translate the rendered image locally in layer l, and y is the class label. This corresponds to a set of the ON-OFF states of the ReLUs, the argmax values from the Max Pooling layers, and the class label y in the CNN. For example, let consider a 2 x 2 image patch at layer 1 in the CNN, taking the following values [2, -3; -4, 5]. After applying Max Pooling, our patch yields one scalar, which is 5. The argmax values from the Max Pooling is then 4 or Lower Right, which implies that the Lower Right location has the highest pixel value in our patch. 4 or Lower Right is then the value for t_{1} at the corresponding pixel in the layer 1 of the NRM. After applying ReLU on 5 yield the same value 5. The ON-OFF states of the ReLU will be 1 instead of 0. 1 is then the value for s_{1} at the corresponding pixel locations in the layer 1 of the NRM. A particular value for (s_{l}, t_{l}, y), l = 1,2,...,L, defines a rendering path in NRM.
>
> The Rendering Path Normalization (RPN) term is proportional to the negative of the log prior of the most probable rendering path (see equation 8). When minimizing the negative log-likelihood of the NRM, this RPN term encourages that the rendering path estimated by the CNN during inference has higher prior compared to other rendering paths.
>
> We will amend the text to include the explanation above.

---

> ### Author Response · Authors · 2018-11-22
> **Examples of how to incorporate the task-specific knowledge into the NRM**
>
> One example of task-specific knowledge discussed in the paper is learning with given labels (supervised learning) or learning without labels (unsupervised learning). As mentioned in the paper, when labeled data are available, we can design the objective loss for CNNs from the conditional log-likelihood of the NRM. Similarly, objective loss for training CNNs with unlabeled data can be derived from the expected complete-data log-likelihood of the NRM. This unsupervised learning loss can only be computed using the NRM since it is the reconstruction loss between the input images and the images reconstructed from the NRM.
>
> Another example of task-specific knowledge that can be incorporated into the NRM is learning when there are outliers in the training set. Instead of using Gaussian pixel noise in equation (2) in our paper, we can allow the noise to be from a Students’ t-distribution. Then the NRM is equivalent to a mixture of Students t-distributions. It has been shown in (Bishop, 2006) that a mixture of Students t-distributions is robust to outliers. However, this idea is out of the scope of our paper, and we leave it for future work.
>
> C. M. Bishop. Pattern Recognition and Machine Learning. Springer, 2006. ISBN 978-0387-31073-2. URL http://research.microsoft.com/en-us/um/people/cmbishop/prml/.

---

### Meta-Review · Area_Chair1 · 2018-12-10
**Very interesting direction but requiring major revision for readability**

**Confidence:** 4
**Recommendation:** Reject

**Metareview:**

This paper introduced a Neural Rendering Model, whose inference calculation corresponded to those in a CNN. It derived losses for both supervised and unsupervised learning settings. Furthermore, the paper introduced Max-Min network derived from the proposed loss, and showed strong performance on semi-supervised learning tasks.

All reviewers agreed this paper introduces a highly interesting research direction and could be very useful for probabilistic inference. However, all reviewers found this paper hard to follow. It was written in an overly condensed way and tried to explain several concepts within the page limit such as NRM, rendering path, max-min network. In the end, it was not able to explain key concepts sufficiently.

I suggest the authors take a major revision on the paper writing and give a better explanation about main components of the proposed method. The reviewer also suggested splitting the paper into two conference submissions in order to explain the main ideas sufficiently under a conference page limit.